



# Coastal HF radars in the Mediterranean: Applications in support of science priorities and  societal needs

Emma Reyes[1], Eva Aguiar[1], Michele Bendoni[2], Maristella Berta[3], Carlo Brandini[2, 4], Alejandro Cáceres-Euse[5], Fulvio Capodici [6], Vanessa Cardin[7], Daniela Cianelli[8,9], Giuseppe Ciraolo[6], Lorenzo Corgnati[3], Vlado Dadić[10], Bartolomeo Doronzo[2,4], Aldo Drago[11], Dylan Dumas[5], Pierpaolo Falco[9,12], Maria Fattorini[2,4], Maria J. Fernandes[13], Adam Gauci[11], Roberto Gómez[14], Annalisa Griffa[3], Charles-Antoine Guérin[5], Ismael Hernández-Carrasco[15], Jaime Hernández-Lasheras[1], Matjaž Ličer[16,17], Pablo Lorente, [18,19], Marcello Magaldi[3], Carlo Mantovani[3], Hrvoje Mihanović[10], Anne Molcard[5], Baptiste Mourre[1], Adèle Révelard[1], Catalina Reyes-Suárez[7], Simona Saviano[8,9], Roberta Sciascia[3], Stefano Taddei[2], Joaquín Tintoré[1,15], Yaron Toledo[20], Marco Uttieri[8,9], Ivica Vilibić[10,21], Enrico Zambianchi [9,22], Alejandro Orfila[15]

[1] SOCIB -Balearic Islands Coastal Ocean Observing and Forecasting System-, Palma, 07122, Spain.
[2] Consorzio LaMMA, Livorno, 57126, Italy.
[3] Consiglio Nazionale delle Ricerche (CNR), Istituto di Scienze Marine (ISMAR), Lerici, 19032, Italy
[4] Consiglio Nazionale delle Ricerche (CNR), Istituto per la Bioeconomia (IBE), Sesto Fiorentino, 50019, Italy
[5] Mediterranean Institute of Oceanography, Université de Toulon, Aix Marseille Univ., CNRS, IRD, MIO, Toulon, CS 60584 83041, France
[6] Università degli Studi di Palermo, Palermo, 90218, Italy.
[7] Instituto Nazionale di Oceanografia e di Geofisica Sperimentale (OGS), Sgonico, 34010, Italy.
[8] Stazione Zoologica Anton Dohrn, Naples, 80121, Italy.
[9] Consorzio Nazionale Interuniversitario per le Scienze del Mare (CoNISMa), Rome, 00196 , Italy
[10] Institute of Oceanography and Fisheries, Split, 21000, Croatia
[11] Physical Oceanography Research Group, University of Malta, Msida, MSD 2080, Malta.
[12] Universita' Politecnica delle Marche, DISVA, Ancona, 60121, Italy
[13] Qualitas Instruments Lda., Caparica, 2825-182, Portugal
[14] Helzel Messtechnik GmbH, Kaltenkirchen, 24568, Germany
[15] Mediterranean Institute for Advanced Studies – IMEDEA- (CSIC-UIB), Esporles, 07190, Spain.
[16] National Institute of Biology, Marine Biology Station, Piran, 6330, Slovenia
[17] Slovenian Environment Agency, Ljubljana, SI-1000, Slovenia
[18] NOLOGIN CONSULTING SL, Zaragoza, 50018, Spain.
[19] Puertos del Estado, Madrid, 28042, Spain.
[20] School of Mechanical Engineering, Tel-Aviv University, Tel-Aviv, 6905904, Israel.
[21] Ruđer Bošković Institute, Division for Marine and Environmental Research, Zagreb, 10000, Croatia
[22] Dipartimento di Scienze e Tecnologie (DiST), Parthenope University of Naples, Naples, 80133, Italy.

*Correspondence to: Emma Reyes (ereyes@socib.es)*

**Abstract.** The Mediterranean Sea is a prominent climate change hot spot, being their socio-economically vital coastal areas the most vulnerable targets for maritime safety, diverse met-ocean hazards and marine pollution. Providing an unprecedented spatial and temporal resolution at wide coastal areas, High-frequency radars (HFRs) have been steadily gaining recognition as an effective land-based remote sensing technology for a continuous monitoring of the surface circulation, increasingly waves and occasionally winds. HFR measurements have boosted the thorough scientific knowledge of coastal processes, also



fostering a broad range of applications, which has promoted their integration in the Coastal Ocean Observing Systems worldwide, with more than half of the European sites located in the Mediterranean coastal areas. In this work, we present a review of existing HFR data multidisciplinary science-based applications in the Mediterranean Sea, primarily focused on meeting end-users and science-driven requirements, addressing regional challenges in three main topics: i) maritime safety; ii) extreme hazards; iii) environmental transport process. Additionally, the HFR observing and monitoring regional capabilities

in the Mediterranean region required to underpin the underlying science and the further development of applications are also analyzed. The outcome of this assessment has allowed us to finally provide a set of recommendations for the future improvement prospects to maximize the contribution in extending the science-based HFR products into societal relevant downstream services to support the blue growth in the Mediterranean coastal areas, helping to meet the UN's Decade of Ocean Science for Sustainable Development and the EU's Green Deal goals.

**1 Coastal monitoring to support Blue Growth in the Mediterranean Sea**

The coastal and ocean economy has been since ancient times and is nowadays, more than ever, the backbone of the Mediterranean countries' blue economy. In 2017, the Mediterranean was the third largest sea basin in terms of Gross Value Added (GVA) and the first in terms of employment  (European Commission, 2020). The key sector is clearly coastal tourism, from which the Mediterranean is the world's leading destination, followed by maritime transport, living resources and port

activities. Furthermore, coastal tourism and fisheries benefit from the location of the Marine Protected Areas (MPAs) which are currently covering the 7% of the northern Mediterranean Sea surface (Meola et al., 2019), and are expected to increase as one of the goals of the United Nations (UN) Decade of Ocean Science for Sustainable Development.

Both, Mediterranean coastal areas and communities are, however, negatively impacted by all human activities related to these traditional sectors. Regarding the sector of maritime transport, it is worth highlighting that the Mediterranean Sea is one of the

world's busiest shipping lanes of oil and gas tankers, container vessels and ships, involving a higher risk of marine oil and marine litter (ML) pollution. Although the extent of the latter is not fully understood yet, first estimations provided from Cózar et al., (2015) identify the Mediterranean Sea as a great accumulation zone of plastic debris, being comparable to the accumulation zones described for the five subtropical ocean gyres. Additionally, Soto-Navarro et al., (2021) have recently found that the hot-spots for the ML risk concentrate in the coastal regions, highly impacting on the Mediterranean biodiversity,

specially, in the MPAs and, particularly, in those near ML sources.

Given the strategic role of ports in the globalized trading system, it is important to underline that four ports from the Mediterranean are included in the top-five European ports when looking at different categories according to the Eurostat statistics from 2020. Moreover, the Mediterranean Sea hosts the three main migratory routes to Europe, representing a huge humanitarian, political and security challenge for the bordering countries. In this context, we cannot ignore the more than

99400 migrants who arrived in Europe in 2020, mainly by sea and, particularly, to Spain, Greece and Italy by crossing the



Mediterranean Sea, according to the data from the International Organization for Migration (IOM). This complex migratory hub contributes to the increased risk to life and maritime safety in the Mediterranean.

Last but not least, as recently reviewed by Tintoré et al., (2019), the Mediterranean is one of the most vulnerable regions in the world due to the impact of climate change. As a result of large-scale warming, among many other impacts reported by the
authors, an increase in frequency and/or intensity of extreme events (Mitchell et al., 2006) is expected. In this context, De Alfonso et al., (2021) points out an average of eight storms per year registered in the Spanish Mediterranean coast with particular severe events registered in November 2001 (Gómez et al., 2002), October 2007 (Cohuet et al., 2011), December 2008 (Sánchez-Arcilla et al., 2014), January 2017 and January 2020 (Amores et al., 2020; de Alfonso et al., 2021, Lorente et al., 2021, Sotillo et al., 2021). Aiming to monitor and understand this regional and sub-regional ocean state and variability,
from daily to interannual scales, a set of indicators for the Mediterranean Sea and the Balearic Islands (Juza and Tintoré, 2021) are made available through a user-friendly visualization tool by SOCIB (Tintoré et al., 2013, 2019).

The increased capability to address the above-mentioned regional challenges at the required spatio-temporal scales has directly benefited, *inter alia*, from the unprecedented high spatio-temporal resolution offered by the High Frequency Radar (HFR hereinafter)  technology in wide coastal areas. Providing continuous monitoring of the surface circulation (Lipa, Barrick and
Maresca, 1981; Paduan and Graber, 1997; Headrick and Thomason, 1998; Molcard et al., 2009; Paduan and Washburn, 2013; Wyatt, 2014; Roarty et al., 2019; Dumas and Guérin, 2020) and, increasingly, wave parameters (Lipa et al., 1990, 2005, 2006; Gurgel et al., 2006; Wyatt et al.,  2006; Orasi et al., 2018; Wyatt and Green, 2009; Long et al., 2011; Wyatt, 2011; Falco et al., 2016; Saviano et al., 2019; 2020; Basáñez et al., 2020; Bué et al., 2020) and, occasionally, wind field (Long and Trizna, 1972; Heron, 2002; Huang et al., 2004; Shen et al., 2012; Kirincich et al., 2016a; Zeng et al., 2016, 2018; Shen and Gurgel, 2018;
Saviano et al., 2021), HFR is a land-based remote sensing technology that give us a unique insight to coastal ocean state and variability with relative ease in terms of technical effort, manpower and costs (i.e. for the same amount of information and compared to other conventional observing platforms), allowing us to improve our understanding of sub-mesoscale and mesoscale coastal processes.

Moreover, coastal ocean surface current and wave real-time information, which are the primary and the secondary basic
products of HFRs, respectively, are being used extensively by: search and rescue (Ullman et al., 2006; Ličer et al., 2020; Révelard et al., 2021), environmental agencies for pollutant monitoring of oil spill (Abascal et al., 2009), marine litter tracking (Declerck et al., 2019), recreational activities, navigational safety, ports & shipping, ship detection and tracking (Ponsford et al., 2001; Dzvonkovskaya et al., 2007; Maresca et al., 2013; Laws et al., 2016), coastal and offshore engineering applications, aquaculture, marine renewables (Wyatt, 2012; Basáñez and Pérez-Muñunzuri, 2021; Mundaca-Moraga et al., 2021) and early
warning detection systems for natural hazards (Lipa et al., 2006; Gurgel et al., 2011; Grilli et al., 2015; Guérin et al., 2018), among others. Furthermore, the mapping of surface currents at high spatio-temporal resolution provided by the HFRs in the coastal strip allow us to use them as a ground truth for coastal model real-time assessment (Wilkin and Hunter, 2013; Lorente et al., 2016, 2019b; Mourre et al., 2018; Aguiar et al., 2020) and improvement, through HFR data assimilation (Breivik and Saetra, 2001; Paduan and Shulman, 2004; Barth et al., 2008; Iermano et al., 2016; Hernández-Lasheras et al., 2021), as well



as for the evaluation of coastal remote sensing products (Manso-Narvarte et al., 2018; Caballero et al., 2020; Gommenginger et al., 2021). The development of advanced HFR data products as the gap-filled nowcasts and Lagrangian trajectories, allow us to satisfactory estimate the transport making the HFR data a key asset in the assessment and protection of the coastal marine environment including: dispersal/retention of particles (Cianelli et al., 2017; Hernández-Carrasco et al., 2018a; Davila et al., 2021), cross-shelf exchanges and transport (Sciascia et al., 2018), eddy tracking (Nencioli et al., 2010; Bagaglini et al., 2020)

and 3D eddy characterization (Manso-Narvarte et al., 2021).

In addition to this, many strong coordinated efforts to significantly increase the prompt distribution, availability, easy access and accuracy of HFR data have been made in recent years at the global (Roarty et al., 2019), European (Rubio et al., 2017) and regional levels (Lorente et al., submitted to this Special Issue), also leveraged by national initiatives and specific projects. These joint efforts have enhanced the creation of a community at the HFR operator level, accelerating therefore the speed of

the take up of the data, also underpinning the growth of HFR multidisciplinary applications worldwide, as described by several authors (Fujii et al., 2013; Paduan and Washburn, 2013; Wyatt et al., 2014; Rubio et al. 2017 and Roarty et al., 2019).

These wide range of applications have also boosted the positive trend on the HFR implementation all around the world. Consistently, HFRs are also nowadays playing a crucial role as one of the backbones of the Coastal Ocean Observing Systems -COOSs- of the Mediterranean Sea, which are currently encompassing more than the half of the existing HFR systems installed

in Europe (Lorente et al., submitted to this Special Issue), constituting therefore an important focus of HFR activity.

Demonstrating the potential of the HFR observing and monitoring regional capabilities, this work reviews the existing advanced and emerging scientific and societal applications using HFR data, developed to address the major challenges identified in the Mediterranean coastal waters, organized around three main topics: i) maritime safety; ii) extreme hazards and iii) environmental transport processes. Recognizing also the added value of networking, it is worth to highlight that this review

encompasses the main outcomes of multidisciplinary, international and intersectoral regional coordinated efforts in the frame of the Mediterranean Operational Network for the Global Ocean Observing System (MONGOOS). These endeavors are primarily focused on meeting end-users and science-driven requirements, aiming to unlock HFR data potential, delivering greater uptake, use and value from the data, for the benefit of the ecosystems, services and human activities of the coastal areas of the Mediterranean Sea.

This manuscript constitutes the second part of two complementary contributions, providing the first one a detailed overview of the main achievements, ongoing activities, future challenges and the roadmap towards an integrated, mature, HFR network in the Mediterranean Sea (Lorente et al., submitted to this Special Issue). The sections of this paper are as follows: Sect. 2 presents several HFR applications addressing science priorities and societal needs, classified in the above-mentioned three topics. Sect. 3 includes the discussion and a preliminary assessment of the capabilities of the existing HFR applications. Based

on this assessment, the Sect. 4 provides a set of recommendations towards setting out future prospects, aiming to maximize the contribution in the provision of the required data products and benefits needed by the socio-economically vital and often environmentally stressed coastal risk-prone areas of the Mediterranean Sea. This contribution will help to overcome the



challenges of the United Nations (UN's) Decade of Ocean Science for Sustainable Development and to address the transitional changes required towards the European Green Deal. Finally, a summary and the main conclusions are provided in Sect. 5.

## 2 High-Frequency radar applications in the Mediterranean

### 2.1 Maritime Safety

Around 200.000 large vessels operate annually in the Mediterranean Sea, including ferries, cargo and commercial vessels, of which around 300 tankers transport oil-based products every day accounting for more than 350 million tons per year (more than 25% of the world's oil tonnage) as highlighted by Di Muccio et al., (2020). This intense maritime traffic makes the basin a susceptible area in terms of oil spills, search and rescue (SAR) operations and other maritime emergencies. Over the past half century spills over the sea from tankers have shown a downward trend, reaching its lowest in early 2020 due to the global health and economic crisis triggered by the COVID-19 pandemic (March et al., 2021). However, oil spills as well as chemical spills and other hazardous substance releases are still present, putting marine health at risk. For instance, in the second half of February 2021, around 170 km of coastline from Israel to southern Lebanon suffered from a large oil spill, one of the worst ecological disasters in decades. In this context, it has once again been demonstrated how accurate forecasting of oil spill modeling (for this particular event, the model MEDSLIK-II was used, as described in De Dominicis et al., 2013a, 2013b) and Lagrangian trajectory analysis of floating objects (Sayol et al., 2014; Ličer et al., 2020) can successfully help marine SAR operations and oil spill containment. These forecasts depend strongly on the accuracy of the forcing data (i.e. wind, waves and currents, as stated in Sect. 2.1.2) ingested in atmospheric and oceanographic models, where in particular ocean surface currents maps from HFRs can highly improve short-term model outputs due its high resolution and its near-real time nature (Abascal et. al., 2009; Abascal et. al., 2012; Breivik et al. 2013), as described in Sect. 2.1.3. In this context real-time HFR data was accepted as a reliable operational tool for SAR, oil spill and other operational protocols in coastal waters (Roarty et al., 2019).

#### 2.1.1 Search and Rescue

Agencies in charge of SAR operations, marine pollution response and maritime traffic control are among the most significantly targeted users of reliable met-ocean information. Access to multi-platform quality control near real-time met-ocean observations and high-resolution forecasts available for their specific areas of responsibility for marine SAR, assigned by the IMO (International Maritime Organisation), is essential for them to support emergency response missions. Winds, waves and surface currents observations and forecasts are needed to be seamlessly integrated into their SAR emergency response and environmental modelling tools in order to predict oil spill drift and fate or the trajectory of a drifting target aiming to determine the optimal search region. In the sphere of maritime safety, HFRs have the great advantage of providing high spatio-temporal resolution surface currents in wide coastal areas, very close to the coastline when HFR gap-filling methods are applied (listed in Sect. 2.1.3) and where most of the SAR incidents occur (as shown by the Fig. 1), as the review of the location of the SAR





incidents from 5 countries (i.e. Croatia, France, Italy, Slovenia and Spain) along 2019 and from Malta along 2020, clearly shows:

(i) Croatia: 612 SAR interventions were registered in 2019, from which 389 are SAR interventions and 223 are MEDEVAC ones (i.e. actions related to transportation of injured or sick persons). Most of these incidents occur during summer, from June to September and over 98% in inner and territorial waters. Coastal waters from Croatia are operationally monitored by the HFR-SPLIT Wera Radar System, consisting of 2 WERA HFR sites (Ražanj and Stončica) in the eastern part of the eastern mid-Adriatic basin. The HFR-NASCUM system is a historical network located in the eastern part of the Gulf of Venice. All 4

sites were used to build NEURAL project short term predictions (described in Sect. 2.1.3).

(ii) France: accounts with 5 SAR responsibility areas in the continental littoral coordinated by 340 SAR operators from the 5 Maritime Rescue Coordination Centre, including 1 for the French Mediterranean responsibility area. In 2019, a total of 13507 SAR incidents occurred (with 22313 people assisted), being the 51% from June to September, as indicated in the website of the French Ministry of the Sea. Particularly, in the French Mediterranean responsibility area (covering 115000 km$^2$), the

number of SAR incidents accounts a 23% (3110) of the total cases and the 32% (7293) of the people assisted, as included in the 2019 activity report of the French Mediterranean Coordination Center. The 94% of the SAR incidents occur in coastal areas in the first 12 nm (22.2 km) and mostly during summer season (from June to September), being more of the 89% related to recreational boating and sailing. Currently, two HFR networks are operating in the French Mediterranean coastal waters of Nice and Toulon, named HFR-MedTln and HFR-MedNce.

(iii) Italy: SAR operations are under the responsibility of the Italian Coast Guard covering 500000 km$^2$ of sea and 8000 km of coast. In 2019 the Italian coast guard responded for 1875 SAR missions, 226 of them related to human migration. No oil spills were reported as for 2019 by the Italian MISE (Ministero dello sviluppo economico) whereas spills from other sources were not officially reported. SkyTruth, a non-governmental agency reported one spill 60 km south of Genoa in the Ligurian sea. Five HFR networks are currently monitoring the coastal areas of the Tyrrhenian and Ligurian Sea (HFR-TirLig) and the Tuscan

Archipelago (HFR-LaMMA), Gulf of Naples (HFR-GoN), the Malta-Sicily Channel (HFR-CALYPSO), the Gulf of Trieste (HFR-NAdr), while two HFR networks, the Gulf of Manfredonia (HFR-GoM), and the Gulf of Venice (HFR-NASCUM) are historical deployment, being also another new deployment (HFR-SIC) in a planned stage.

(iv) Malta: there is one Maritime Rescue Coordination Centre with around 50 SAR operators, covering one Search and Rescue Region (SRR) of 267 874 km2 with 196.8 km coastline (including Comino & Gozo). During 2020, 429 missions were

coordinated by the MRCC (Maritime Rescue Coordination Centre) in Malta, 26% of which were reported as SAR cases, occurred within the Maltese Territorial Seas. The HFR-CALYPSO monitors the Malta-Sicily channel, which accounts for 7 HFR sites, 4 around the Malta island and 3 more in the Sicilian southern coast. The HFR data together with the outcome of forecasting models are served by the Physical Oceanography Research Group being used for SAR operations.

(v) Slovenia: has 42 km of coastline and a semi-enclosed coastal area. During 2019, the SAR agency has responded to 9 SAR

missions (7 times the rescue boat went out to sea while 2 rescues were of injured people on a moored boat in port). All cases occurred within the 3 nm from the coast (i.e. 3 within 200 meters, 3 around 1 nm and 1 at 3 nm from the coast)





(vi) Spain: the 4 SAR responsibility areas cover 1 500 000 km2 of marine surface (3 times the size of the Spanish national territory) and 8 000 km of coastline. The Spanish Maritime Safety and Rescue Agency (SASEMAR hereinafter) is divided in 19 MRCCs plus 1 National Centre, with more than 370 SAR operators. SASEMAR responded to 5 891 missions in 2019, from
which almost 88% were SAR operations (being the 12% in response to issues related to marine pollution). Fifty percent of the total SAR incidents occurred within 3 km off the Spanish coastlines. From 7 of the HFR networks operating inside their 4 responsibility areas, 3 of them are located in the Western Mediterranean, monitoring the Strait of Gibraltar (HFR-Gibraltar), the Ebro Delta (HFR-Ebro) and the Ibiza Channel (HFR-Ibiza) and all of them are integrated in the SASEMAR Environmental Data Server.

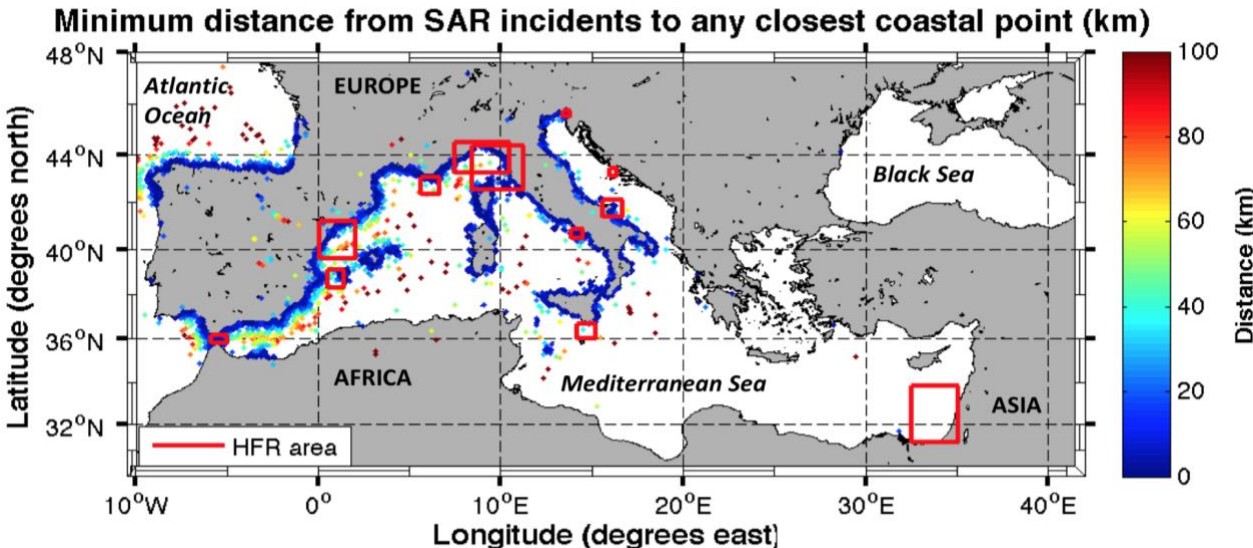


**Figure 1. Map of the Mediterranean showing the HFR bounding boxes and the location of SAR incidents of France, Italy, Slovenia and Spain from 2019 colored based on their distance to the closest coastal point. HFR bounding boxes are shown with red boxes.**

As aforementioned, maritime SAR operations most often depend on leveraging Lagrangian tracking tools using timely and
reliable knowledge of surface circulation, near surface winds and, if applicable, surface gravity waves. Surface circulation is generally provided by numerical circulation models but HFR observations can offer valuable insight into marine conditions over the region of the accident and can - especially when coupled to short term prediction models (see Sect. 2.1.3) - act as a complementary input for Lagrangian predictions, hindcasts or back-tracking simulations. Révelard et al. (2021) evaluated the use of HFR-derived trajectories to complement drifter observations for assessing the performance of different models in
predicting Lagrangian trajectories. They used the Skill Score metric, based on the Normalized Cumulative Lagrangian Separation distance (Liu and Weisberg, 2011), which is a commonly used metric for assessing Lagrangian performance. They have shown that whereas drifters only provide assessment along their drifting paths, HFR allows obtaining a large number of trajectories, improving not only the robustness of the Skill Score statistics but also the spatial and temporal assessment of the





model performance (Fig. 2). Since HFR data are quasi-continuous in time, this method can be applied in near-real-time, which
is a strong advantage for evaluating extremely scenario-dependent models. Indeed, the quality of any numerical model
performance varies with time and can have substantial fluctuations on short temporal and spatial scales even if the model
otherwise exhibits good overall forecasting skills. In cases like these, quality controlled HFR observations represent
particularly valuable short-term inputs for Lagrangian products assisting SAR efforts.

Révelard et al. (2021) also analyzed the Skill Score sensitivity to different forecast horizons and showed that in coastal regions,
where most of the SAR incidents occurred in the Mediterranean Sea, an overly long forecast time (i.e. 72 hours) can lead to
an overestimation of the Skill Score due to the high variability of the surface currents. A forecast time of 6 hours, consistent
with the duration of the search that maximizes survivors in SAR missions, is therefore more appropriate when using HFR as
the observing reference. In addition, they have shown that whereas the original definition of the Skill Score from Liu and
Weisberg (2011) is correct for analyzing its spatiotemporal distribution, it is not adequate if averages are going to be applied
afterward, because of the previous imposition of the negative values to zero. In the aim of estimating the relative average
performance of different datasets over an area and/or period, they introduced the novel Skill Score SS*. The SS* is defined as
in Liu and Weisberg (2011), but without the imposition of the negative values to zero, allowing to obtain a correct average, as
in Fig. 2 where the SS* is temporally averaged over a period of 6 hours. However, as pointed out in Révelard et al. (2021),
only values > 0.5 should be interpreted as a good agreement between HFR surface current observations and model outputs.





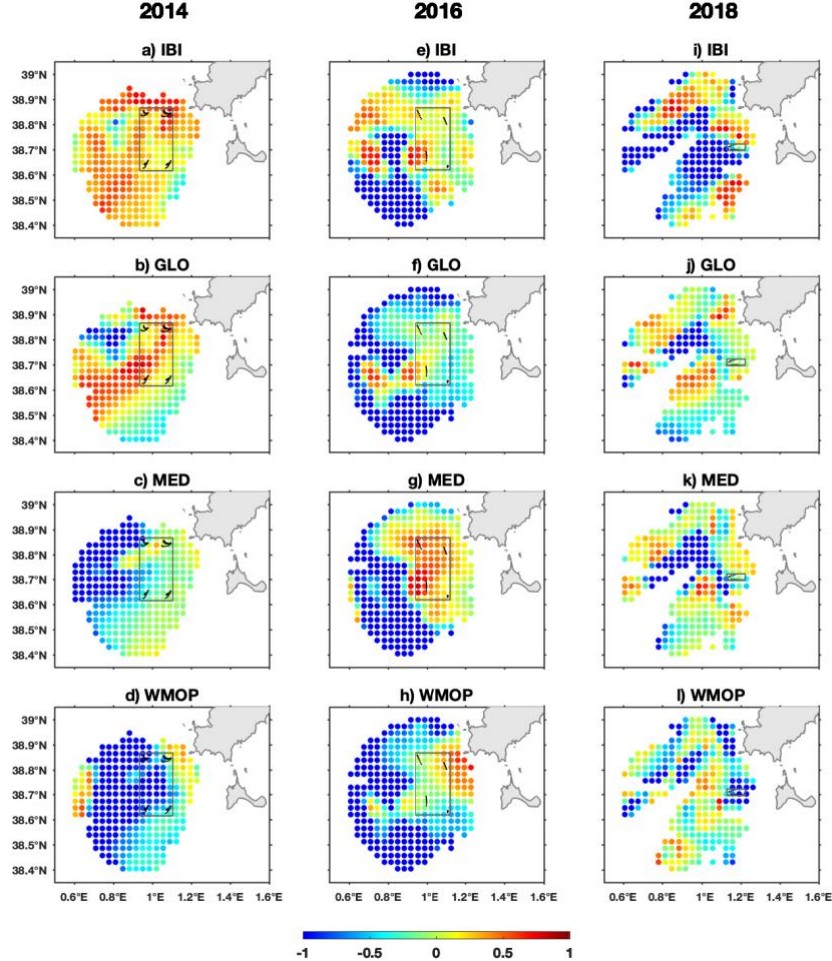


**Figure 2.: Temporally averaged Skill Score SS\* obtained for four models as indicated in the figure title by comparing against the HFR-derived trajectories of the Ibiza Channel during a forecast time of 6 hours. Simulated trajectories are initialized hourly at each grid points on 30-sept-2014 from 13:00 to 16:00 (left panels), on 28-July-2016 from 16:00 to 22:00 (middle panels) and on 15-Nov-2018 from 13:00 to 16:00 (right panels). SS\* values are only obtained in those grid points with data**
**temporal availability equal or higher than 80%. Black lines show the drifter paths available during the same periods, and the boxes indicate the regions where the averages are applied for comparison with the results obtained with drifter observations. Original source: From Révelard et al. (2021).**

A further example of the value of quality controlled HFR observations in SAR operations was the recent case of a person lost at sea in Northern Adriatic during a Scirocco storm on 29 Oct 2018. In this case, HFR-NAdr observations were employed for
hindcasting and survivor's drift trajectory verification (Ličer et al., 2020). Figure 3 depicts Lagrangian drifter dispersal computed from modeled surface winds (the dominant contribution to the drift in this case) and HFR surface currents from the HFR-NAdr network in the Gulf of Trieste after this accident.




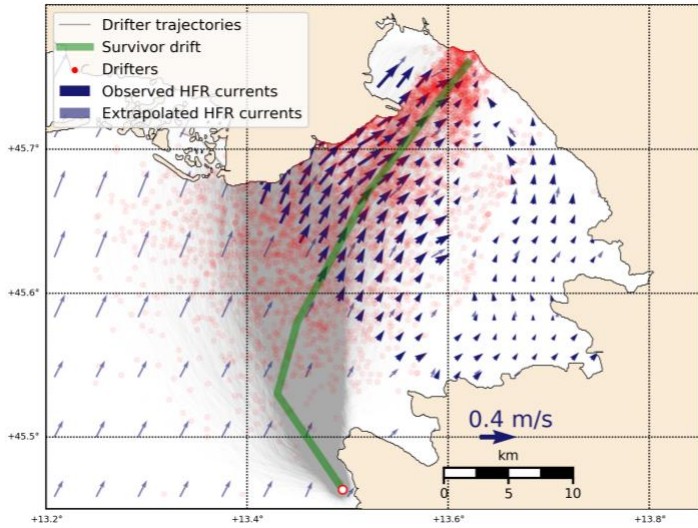

**Figure 3. Using HFR currents for Lagrangian hindcasting of an accident on 29 Oct 2018 in the Gulf of Trieste, details in**
**(Ličer et al., 2020). Blue arrows denote the HFR surface current field on 29 Oct 2018 22UTC. Thinner (but scaled to length)**
**light-blue arrows depict nearest-neighbor extrapolated currents outside the HFR domain (every third point is plotted for**
**clarity). Red dots denote modeled virtual drifter locations after 24 hours of the drift, starting from the accident location (white**
**circle at 13.495 E, 45.4635 N). Green line indicates the survivor's estimate of his drift trajectory. Adapted from Ličer et al.,**
**2020.**

Even though in this case part of the survivor's trajectory outside of the HFR-NAdr domain had to be inferred from extrapolated

currents, such HFR-based nowcasting products would have been valuable during this and similar rescue attempts. However,

since HFR data arrive in near real time, some sort of model-based extension of their prediction horizon is necessary before

they can be used for operational nowcasting. One possible solution is data assimilation of HFR data into a numerical model

(see Sect. 2.1.2.), followed by a forecasting time window. An alternative and numerically less demanding option, gaining

ground in recent years, is the machine learning approach where a neural network model is trained on past data and then used

to create short term predictions of surface currents, addressed in Sect. 2.1.3.

**2.1.2 Model assessment and improvement**

This section addresses one of the main interests and needs of end-users of operational oceanography information: users want

to be able to have confidence in modelled data and they need to know how good they are. Addressing end-user overarching

concerns, model assessment, essentially built upon comparison to observations, is crucial to evaluate the quality of the diversity

of modelling products available in a systematic and long-term routine manner, and to inform users about their usefulness for

a given application.

For this reason, seeking also to strengthen end-user loyalty, the validation of Operational Ocean Forecasting Systems against

independent measurements constitutes a core activity in operational oceanography (Hernández et al., 2015) since it aids: (i) to

infer the relative strengths and weaknesses in the modelling of several key physical processes; (ii) to compare different versions





of the same operational ocean forecasting system and evaluate potential improvements and degradations before a new version is transitioned into operational status; (iii) to compare coarse-resolution "parent" and nested high-resolution "child" systems to quantify the added value of downscaling; (iv) to inform end-users about the consistency and skill of the modeling products disseminated.

Developments in ocean modelling have clearly advanced to address the challenges associated with the increased resolution and its application to coastal areas, also responding to the high demand of providing 4D estimates of multiple oceanic variables at fine-scales (Mourre et al., 2018; Fox-Kemper et al., 2019). Coastal modeling faces numerous challenges and issues as downscaling and representation of open boundary conditions or land-sea/air-sea interactions, for instance (Kourafalou et al.et al. 2015a). Synergies between models and ocean observations are needed to face these challenges and improve ocean processes

representation (Kourafalou et al.2015b; De Mey-Frémaux et al., 2019; Davidson et al., 2019). Additionally, it is worth mentioning the current lack of real-time and historical availability of observations on the coastal areas, which limits the operational capability and reduces the potential of skill assessment operational services aiming to provide synthetic metrics addressing specific user's needs (Révelard et al., 2021).

Within this context, HFR systems play a first-order role thanks to their unique ability to provide fine-resolution maps of the

surface currents over broad coastal areas. This ability of the HFR system makes them particularly appropriate for the validation of numerical models in coastal areas, where other observations are scarce and/or their resolutions (i.e. in space or in time) are not high enough to capture the fine scale. Many HFR systems have therefore been used with this purpose in several regions of the Mediterranean Sea including the Northern Current area off Toulon (Berta et al., 2014a), the Ebro Delta area (Lorente et al., 2016; Ruiz et al., 2020; Aguiar et al., 2020; Lorente et al., 2021; Sotillo et al., 2021), the Northern Adriatic (Vilibić et al.,

2016), the Gulf of Naples (Uttieri et al., 2011), the Ibiza Channel (Mourre et al., 2018; Aguiar et al. 2020; Révelard et al., 2021; Sotillo et al., 2021) and the Strait of Gibraltar (Lorente et al., 2019a; Aguiar et al., 2020).

An example of this added-value of the HFR data was recently shown in the multi-model comparison exercise performed in the Strait of Gibraltar in 2017 (Lorente et al., 2019a). This comparison was made between the coarser CMEMS-IBI model (Sotillo et al., 2015) and their partially nested SAMPA (Sánchez-Garrido et al., 2013) high-resolution coastal forecast system to

elucidate the accuracy of each system characterizing the Atlantic Jet (AJ) inflow dynamics. To this aim, HFR-derived hourly currents at the midpoint of the selected transect (square in Fig. 4,a) were used as a benchmark. The scatter plot of HFR-derived hourly current speed versus direction (taking as reference the north and positive angles clockwise) revealed interesting details (Fig. 4, b): (i) the AJ flowed predominantly eastwards, forming an angle of 78° with respect to the north; (ii) the current velocity, on average, was 1 ms$^{-1}$ and reached peaks of 2.5 ms$^{-1}$. Speeds below 0.5 ms$^{-1}$ were registered along the entire range

of directions; (iii) westwards currents, albeit in the minority, were also observed and tended to predominantly form an angle of 270° (i.e. towards the Atlantic), mostly related to intense easterly winds episodes (Garret, 1983; García-Lafuente et al., 2002; Menemenlis et al., 2007; Péliz et al., 2009; Reyes et al., 2015; Lorente et al., 2019b and 2019b; Bolado-Penagos et al., 2021), as further detailed in Sect. 2.2.1. The scatter plot of SAMPA estimations presented a significant resemblance in terms of prevailing current velocity and direction (Fig. 4,c). Although the time-averaged speed and angle were slightly smaller (0.9





ms$^{-1}$) and greater (88º), respectively, the main features of the AJ were qualitatively reproduced: maximum velocities (up to 2.5

ms$^{-1}$) were associated with an eastward flow and an AJ orientation in the range of 50º–80º. Besides, surface flow reversals to

the west were properly captured. By contrast, noticeable differences emerged in the scatter plot of regional IBI estimations

(Fig. 4, d): surface current velocities below 0.3 ms$^{-1}$ were barely replicated and the AJ inversion was only observed very

occasionally. Despite the fact that IBI appeared to properly portray the mean characteristics of the eastwards flow, the model

tended to favor flow directions between 60º and 180º and to overestimate the current velocity, with averaged and maximum

speeds around 1.17 and 2.80 ms$^{-1}$, respectively.

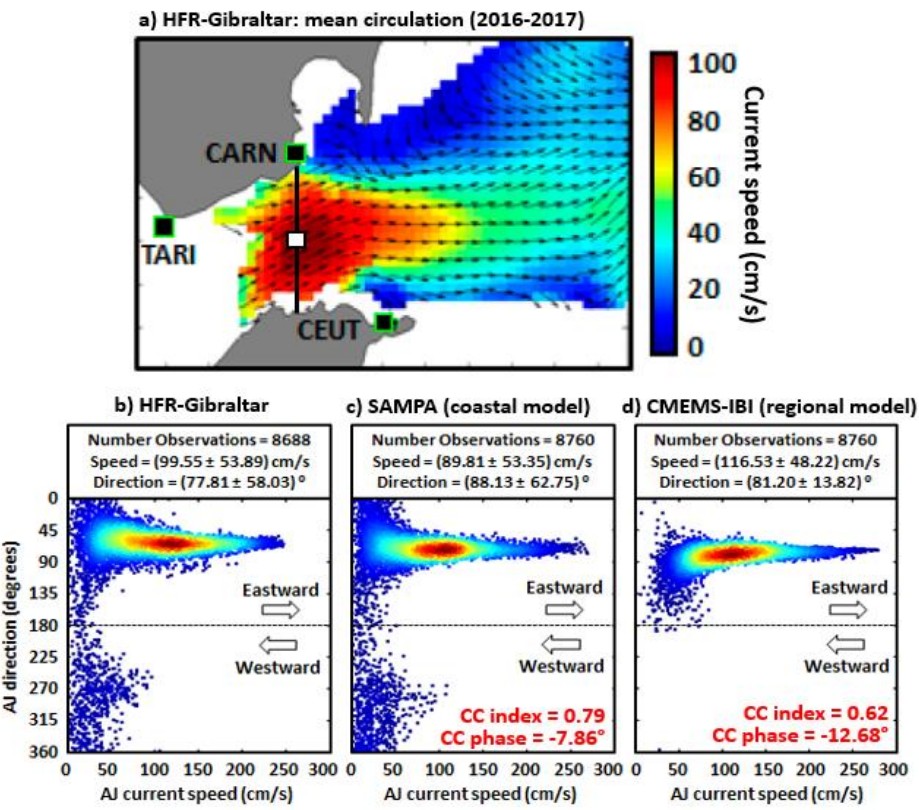

**Figure 4. (a) HFR-derived mean surface circulation pattern in the Strait of Gibraltar for 2016–2017: classical Atlantic Jet**
**inflow into the Mediterranean, with strong surface currents flowing to the NE. Solid black squares represent radar sites.**
**Black line and the related white square indicate the selected transect and its midpoint, respectively. (b-d) Quantitative**
**validation at the selected grid point (5.43ºW, 35.99ºN) within the Strait of Gibraltar: annual (2017) scatter plot of hourly**
**current speed versus direction (angle measured clockwise from the north); estimations provided by HFR-Gibraltar (b),**
**SAMPA high-resolution coastal model (c) and CMEMS-IBI (d), a regional model in which SAMPA is nested into. Mean and**
**standard deviation values of both AJ speed and direction are gathered in black boxes. Complex correlation (CC) index and**
**phase are illustrated in red color. Adapted from: Lorente et al., 2019a.**

In summary, HFR measurements are able to precisely assess the added value of the downscaling performed through the

SAMPA coastal system with respect to the CMEMS-IBI MFC regional solution, in which SAMPA is nested. Overall, a steady





improvement in the Atlantic Jet characterization is evidenced in model performance when zooming from regional to coastal
configurations, highlighting the benefits of the downscaling approach adopted and also the potential relevance of a variety of
factors at local scale, among others: a more refined horizontal resolution, a tailored bathymetry or the higher spatio-temporal
resolution of the atmospheric forcing. Furthermore, SAMPA appeared to better reproduce the reversal events detected with
HFR estimations, demonstrating the added value of imposing accurate meteorologically driven barotropic velocities in the
open boundaries, imported from the NIVMAR storm surge model (Álvarez-Fanjul et al., 2001), in order to consider the remote
effect of the atmospheric forcing over the entire Mediterranean basin, which was only partially included in IBI.

During the next phase of CMEMS, the higher focus will be on coastal downstream applications (e.g., very high-resolution
ocean models integrated with coastal observatories) for specific stakeholders such as harbors and environmental agencies.
Despite the significant progress in the field of coastal modelling, some storm-induced hazards are still not properly resolved
(or even misrepresented) by ocean models due to a variety of factors (e.g. too coarse horizontal resolution, inadequate
meteorological forcing, poor representation of land-sea interactions and the related river freshwater outflows, among others)
as described by Sotillo et al. (2021). Within this framework, HFR might act as a monitoring cornerstone to calibrate and
validate successive, upgraded versions of operational ocean forecasting models with the aim of better capturing extreme events
in terms of strength, extension and timing (Lorente et al., 2021).

Aguiar et al. (2020) used the three HFR systems available in the Western Mediterranean Sea (Gibraltar Strait, Ibiza Channel -
described in Lana et al., 2016 - and Ebro Delta) to evaluate the impact of downscaling on the surface coastal circulation in the
case of the WMOP model (Juza et al., 2016; Mourre et al., 2018). The authors showed that the time-average circulation in the
coastal areas of the Ebro delta and Ibiza Channel were improved through downscaling. In particular, the nested model showed
a better representation of the small-scale coastal flow intensification at the mouth of the Ebro River and a refinement in the
characterization of the circulation in the Ibiza Channel. Notice that HFR-Gibraltar, HFR-Ebro and HFR-Ibiza *versus* model
comparisons are updated daily on SOCIB WMOP webpage -https://socib.es/?seccion=modelling&facility=wmedvalidation-.
Those HFR systems, among others, are also integrated in the IBISAR science-based data downstream service (Reyes et al.,
2020) –freely available under registration in wws.ibisar.es- for visualizing, comparing and evaluating the performance of ocean
current predictions in the Iberian-Biscay-Irish regional seas. IBISAR allows the identification of the most accurate ocean
current dataset in a specific area and period of interest, thus facilitating decision-making to SAR operators and emergency
responders. Additionally, those HFR systems are also being used for the CMEMS IBI-MFC model assessment purposes by
means of the NARVAL multi-parameter and multi-platform validation tool (Lorente et al., 2019c) for the CMEMS IBI-MFC
model validation.

Another added-value of HFR systems is their use to improve model forecast through data assimilation (DA). DA aims at
optimally combining observations and models to provide a better representation of the ocean dynamics. In this sense, HFR
provides very valuable high-resolution observations in areas where satellite observations tend to suffer limitations due to the
vicinity of the coast  (Vignudelli et al., 2019). While  the assimilation of HFR measurements has been applied in many regions
of the world since the first studies from Breivik (2001) and Oke et al. (2002), only a limited number of studies has been





performed in the Mediterranean Sea. Marmain et al. (2014) assimilated radial observations from Toulon HFR system in a regional model in the Gulf of Lion. They showed how HFR observations can be successfully used to correct the wind forcing

used to constrain the model coastal surface circulation. In the Ligurian Sea, Vandenbulcke et al. (2017) were able to correct surface currents and improve the representation of inertial oscillations after the assimilation of all the available hourly radial observations in a regional model of the area. Variational methods were also applied to improve model dynamics through multi-platform data assimilation including HFR in the southern Tyrrhenian Sea (Iermano et al., 2016) and in the Adriatic Sea (Janeković et al., 2020).

More recently, Hernández-Lasheras et al. (2021) specifically assessed the impact of assimilating HFR observations on the surface currents in the Ibiza Channel using the WMOP operational system. They compared the performance of both radial and total daily mean HFR-Ibiza surface currents (Tintoré et al., 2020) for correcting meso and submesoscale circulation using different initialization methods in an operational-like context. An independent Lagrangian validation performed by comparing non-assimilated, assimilated without and with HFR measurements with a set of 14 surface drifters (Tintoré et al., 2014) showed

that the best results were obtained when using HFR total observations along with the traditional observation sources (i.e. satellite altimetry, SST and Argo temperature and salinity profiles). After 48 hours, the mean separation distance between virtual buoys and real drifters was reduced by 53% compared to the simulation without any data assimilation, and by 29% compared with the simulation assimilating traditional observations only (as shown in the Fig. 5).

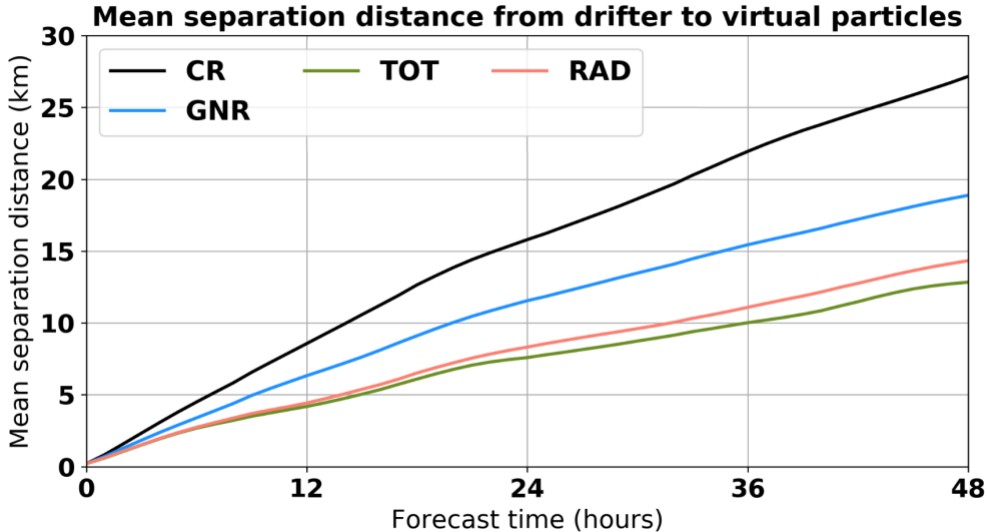

**Figure 5. Mean separation distance between virtual particles and drifters as a function of the forecast horizon. Blue color represents the simulation without DA. Blue stands for GNR (generic), which assimilates data from satellite altimetry, SST and Argo profiles. Green and red lines represent the simulation which assimilates HFR daily mean total and radial observations, respectively, together with generic observation sources. Original source: Hernández-Lasheras et al. (2021)**

To the best of authors´ knowledge, SOCIB WMOP (https://socib.es/?seccion=modelling&facility=forecast) is presently the

only system in the Mediterranean Sea including an assimilation scheme of HFR data in its operational chain.





### 2.1.3 Short Term Predictions

Assimilation of HFR data into models is still computationally expensive and a complex issue, not to mention operational capabilities of such a procedure. Because of these constraints, the availability of real-time high-resolution HFR current fields has led to alternative solutions in order to obtain short term prediction (STP) of surface coastal currents, through the direct use of HFR historical and nowcast observations using different approaches (e.g. Zelenke 2005; Frolov et al. 2012; Barrick et al., 2012; Orfila et al. 2015; Vilibić et al, 2016).

The above-mentioned studies develop and implement different STP approaches (harmonic analysis of the last hours, genetic algorithms, numerical models, etc.) which often require additional data, or long training periods of data without gaps. Hardware failures due to power issues, communications or environmental conditions often result in spatio-temporal gaps within HFR datasets. Spatial gaps can be filled on a real-time basis but the filling of long temporal gaps is not straightforward. Several gap-filling methodologies have been developed for HFR data sets: Open Modal Analysis – OMA - (implemented by Lekien et al., 2004 and further optimized by Kaplan and Lekien, 2007), Data Interpolating EOFs -DINEOF- (Beckers and Rixen, 2003; Alvera-Azcárate et al., 2005; Hernández-Carrasco et al., 2018b, Bourg and Molcard, 2021), and Self-Organizing Maps -SOM- (Kohonen, 1982, 2000, 2001; Hernández-Carrasco et al., 2018b), Reduced Order Optimal Interpolation -ROII- (Kaplan et al., 1997), Optimal Interpolation -OI- (Kim et al., 2008); Artificial Neural Network -ANN- (Ren et al., 2018) , Variational Analysis (Yaremchuk and Sentchev, 2011), Data-Interpolating Variational Analysis, in n-dimensions -DIVAnd- (Barth et al., 2021). HFR derived short term predictions were developed by Zelenke (2005), Frolov et al. (2012), Barrick et al., (2012), Orfila et al. (2015), Solabarrieta et al., (2016), Vilibić et al, (2016), Abascal et al., (2017). More recently, Solabarrieta et al. (2021), developed a Lagrangian-based empirical real-time, Short-Term Prediction (L-STP) system in order to provide short term forecasts of up to 48 hours of ocean currents from HFR data.

Through the NEURAL project (http://www.izor.hr/neural), an innovative neural network-based ocean forecasting system has been developed, providing gridded hourly surface current forecasts in the northernmost part of the Adriatic for the next 72 hours. The forecasting system is using unsupervised neural network algorithm, Self-Organizing Maps (SOM, Kohonen, 1982; Liu et al., 2006), to train joint solutions coming from the HFR measurements and numerical weather prediction model as hourly surface currents and surface winds, respectively. Once the joint SOM solution has been trained, the surface current forecast follows the predicted surface winds being the closest to the specific SOM solution (Fig. 6). Such a system prerequisites a strong relationship between the predictor (here surface winds) and the predictand (here surface currents), which is largely found in coastal regions of the Mediterranean, yet it can be applied for any other combination of predictors and predictands. Also to add, high-frequency processes such as tides are removed from the system as being minor to the wind-driven dynamics, yet the tides can be added to the forecast.

The quoted northern Adriatic forecast system has been trained using 20 SOM solutions (so-called Best Matching Units, Liu et al., 2006) on HFR data measured between February and November 2008 conjoined with 3-hourly surface winds interpolated to 1-hour resolution coming from Aladin/HR operational model run once a day by the Croatian Meteorological and




Hydrological Service (Tudor et al., 2013). The forecasting system performance has been tested in the forecast (hindcast) mode

during 2009 and 2010. Unfortunately, the HFR system had substantial problems since 2010, while the antennas were removed

in the following years, resulting in a relatively short dataset possibly not sufficient to put a strong reliability to the forecasting

system solutions. However, Vilibić et al. (2016) compared this SOM-based forecasting system with the operational ROMS -

Regional Ocean Modelling System- (Shchepetkin and McWilliams 2003, 2005) ocean model for the Adriatic and found that

the SOM-based forecasting system performed better, having lower biases and root-mean-square errors.

As HFR measurements expanded a lot in the last decade, both in space (new HFR systems) and time (longer time series), such

a forecasting system that uses self-learning algorithms might be applied to other Mediterranean locations, in particular to those

having multi-year surface current datasets. For such sites, the stability of SOM-solutions in time may be tested as well, or the

self-learning and training of SOM solutions might change in time to properly reflect long-term changes in oceanographic

conditions in a coastal area.

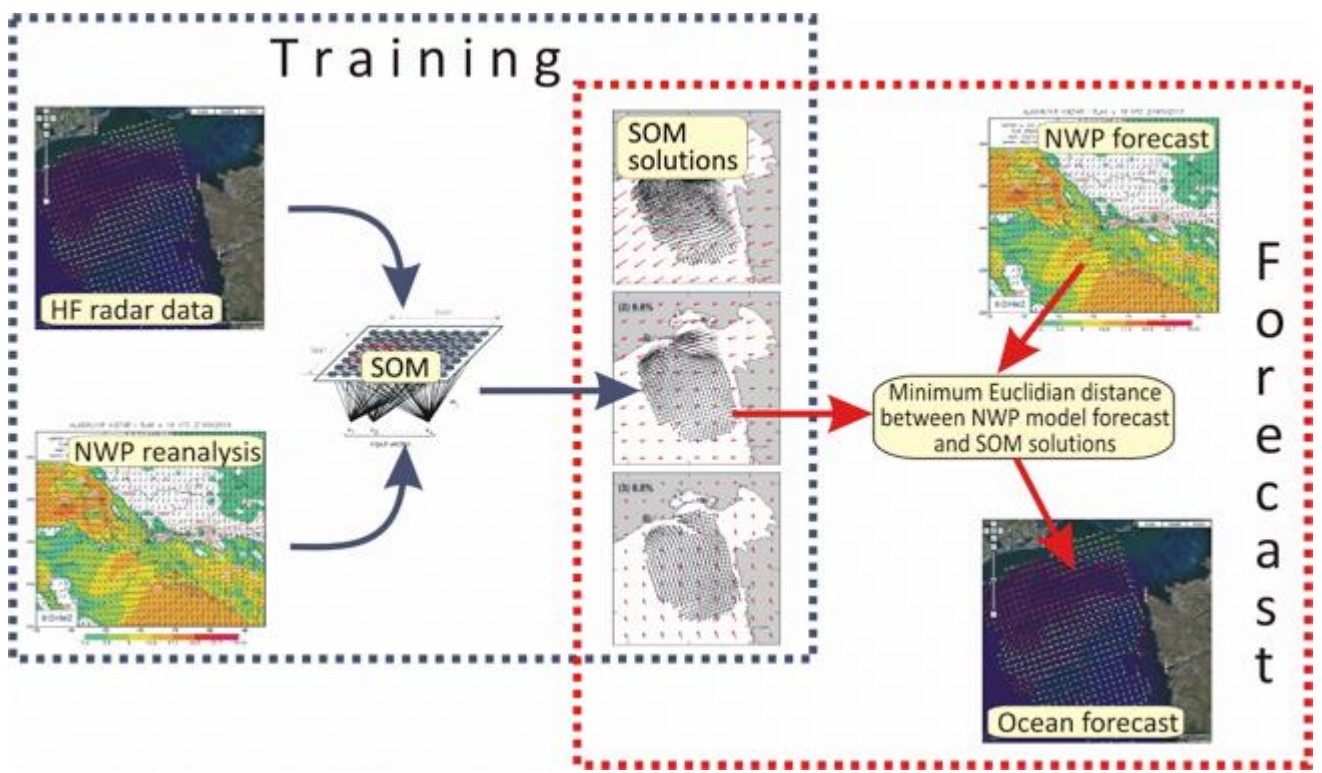


**Figure 6. The architecture of the SOM-based surface currents forecasting system in the northern Adriatic. Original source: from Vilibić et al., 2016.**

**2.2 Extreme hazard coastal monitoring**

Under the current climate-change scenario, no portion of the coastline is safe from the threat of metocean hazards, which are

expected to increase in frequency, duration and virulence during this 21$^{st}$ century (Mitchell et al., 2006; Stott, 2016). HFR





constitute a profitable asset for wise decision-making since it presents a wide range of practical applications, including the effective monitoring in near real time of extreme coastal hazards, such as extreme wind events, severe river discharges, storms and strong flow reversals (see further details in Sect. 2.2.1.), storm surges, tsunamis (more information available in Sect. 2.2.2), typhoons and hurricanes (Barrick and Lipa, 1986; Miles et al., 2017; Lipa et al., 2019).

A detailed characterization of unusual phenomena in coastal areas as well as the increasingly retrieval of waves and winds maps derived from HFRs, is very relevant for the Blue Economy emerging and innovative sectors, including marine renewable energy, among others.

### 2.2.1 Extreme events monitoring

HFRs have been used to investigate the response of coastal submesoscale structures, shaping surface currents and passive
transport, to an extreme wind event in the Ligurian Sea, a sub-basin in the North-Western Mediterranean during October-November 2018, as described in Berta et al. (2020). The analysis was based on estimates of kinematic properties' pattern and magnitude from surface currents measured by the HFR-TirLig network before and after the extreme wind event. In particular this work focused on divergence/convergence and vorticity characterization indicative of the evolution of ocean scales at a few kilometers.

During the storm, sea surface vorticity (Fig. 7, top panels) and divergence (not shown but available in Berta et al., 2020) reach order of the Coriolis parameter $f$, indicating ageostrophic activity typical of submesoscale structures. The evolution of the sea surface structures suggested nonlinear interactions with the wind forcing. Considering the time series of wind speed and sea currents properties (Fig. 7, lower panels), during and right after the storm (around October 29), currents magnitude increased approximately four times while vorticity and divergence associated with the small features almost doubled. Such abrupt
changes in horizontal currents and transport might impact also the vertical properties and in turn the ecosystem.

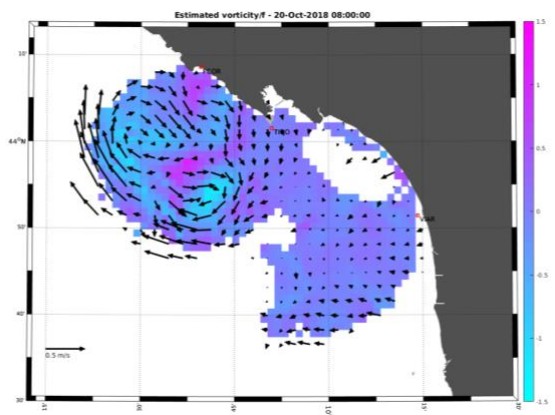

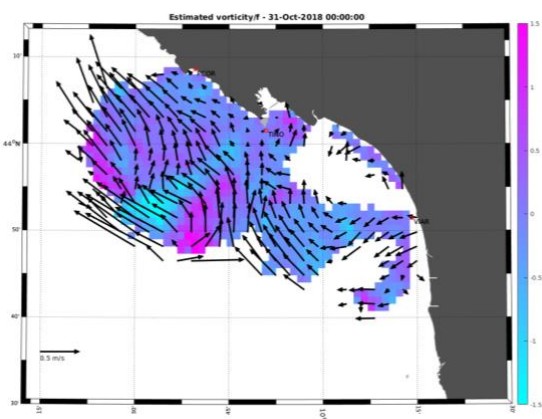





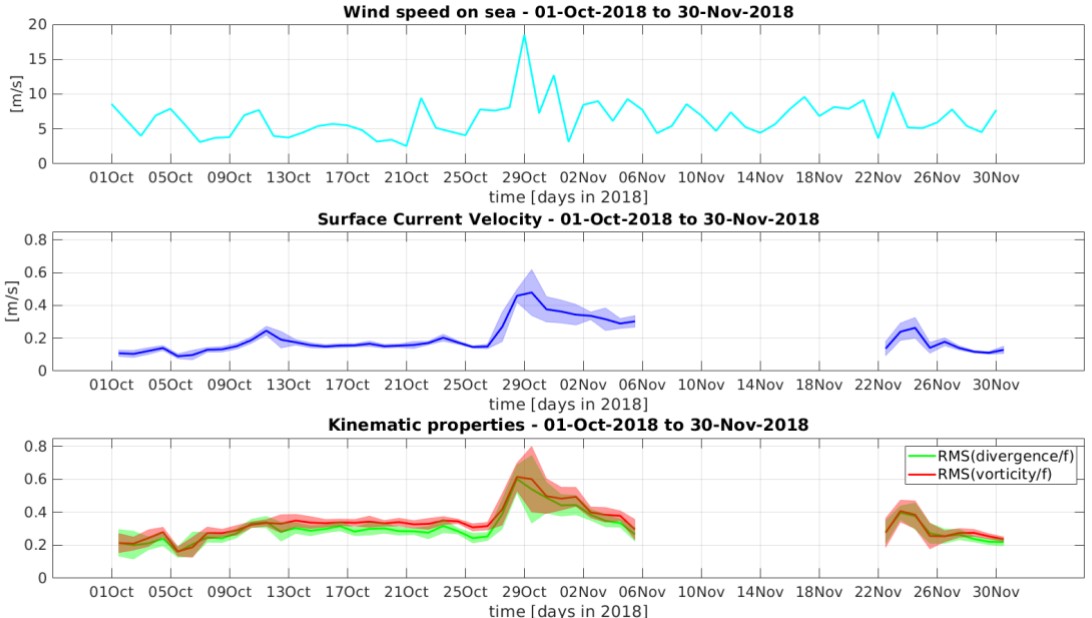

**Figure 7. From top to bottom: example map of normalized vorticity before/during the extreme event. Time series of wind speed, surface currents magnitude, and RMS of normalized vorticity and divergence. Original source: from Berta et al., 2020.**

In the Delta of the Ebro river, also in the North-Western Mediterranean coast, the HFR-Ebro system observations have been crucial to capture the evolution of the most extreme Ebro river freshwater discharge event registered over the last 15 years in

April 2018 and the high impact of the freshwater-pulse discharged on the surface circulation pattern (Ruiz et al., 2020). Results show that the surface circulation pattern is highly influenced by the river discharge, exhibiting a clear correspondence with high concentrations of satellite-derived Chl-a concentration. Hovmöller diagrams of HFR derived meridional and zonal currents indicate an increase of the south-eastward velocity during the period of the extreme river discharge. The proper representation of the basic oceanographic features of the HFR-Ebro, as the Ebro river impulsive-type freshwater discharged,

was previously reported by Lorente et al. (2015).

This same region has been severely impacted by an exceptional storm in January 2020. Particularly for this event, Lorente et al. (2021) have recently assessed the ability of the HFR-Ebro to characterize waves and currents under the record-breaking storm Gloria (19-24 January 2020) and evidenced its remote-effect in adjacent choke-points of the NW-Mediterranean, adding to the analyzing the data from the HFR-Ibiza and HFR-Gibraltar. Furthermore, the effect of Gloria on the particle's dispersion

at the Ebro river's mouth has been established. The main findings are the identification of: (i) the storm Gloria as an extreme event in the NW Mediterranean Sea, considering the persistent surpassing of the 99th percentile for several parameters (i.e. wind speed, significant wave height, wave period and surface current velocity) in relation to the climatology and to a previous storm in the same region (in January 2017); (ii) the remote effects of the Gloria storm in the Ibiza Channel and the Strait of





Gibraltar, which altered the usual water exchanges between adjacent sub-basins. Specifically, the flash pressure drops of about

18 hPa in 27 hours in the Ibiza Channel causes very strong easterly winds of about 11 ms$^{-1}$ by the end of the 19th January.  As a result, surface current speed reached its monthly maximum of 0.6 ms$^{-1}$, showing a clockwise veering from NE, to SW during the storm and to NW afterwards. In the Strait of Gibraltar, the Atlantic Jet speed decreased one third and the HFR registered a uniform inversion of the surface circulation with westwards currents above 0.7 ms$^{-1}$ on the 20th of January. In the HFR-Ebro system the maximum current speed of about 0.87 ms$^{-1}$ was registered on the 21st of January towards the S-SW, surpassing all

values, i.e. the mean value of about 0.16 ms$^{-1}$; the P99 of ~0.34 ms$^{-1}$, being more than twice the mean and the peak value of 0.53 ms$^{-1}$ registered during the severe 2017 event (iii) areas of elevated instantaneous rate of separation (IROS) at Ebro river mouth, indicative of high rates of particle dispersion, which reached a peak by January 21st.

Strategically located at the only natural entrance from the Atlantic Ocean to the Mediterranean Sea the HFR system installed in the Strait of Gibraltar (SoG) can be considered to be an appropriate asset to effectively monitor the Atlantic Jet (AJ) inflow

(Lorente et al., 2019b). The classical picture of the surface circulation is characterized by current pulses often exceeding 2 ms$^{-1}$ and time-averaged north-eastward speeds around 1 ms$^{-1}$ in the narrowest Sect. of the SoG (Fig. 4, a). Complete collapse of the AJ and quasi-permanent inversion of the surface inflow during prevalent intense easterlies is a singular phenomenon that deserved detailed exploration (as previously mentioned in Sect. 2.1.2). Under this temporal premise, a monthly Hovmöller diagram was computed for HFR-derived zonal currents at the selected transect to easily detect a 2-day full reversal episode

during March 2017, represented by black boxes in Fig. 8, a. The event detected consisted of an abrupt interruption of the eastward inflow and complete reversal of the surface stream through the narrowest Sect. of the SoG (Fig. 8, b). The circulation in the easternmost region of the study domain was accelerated up to 0.8 ms$^{-1}$, following clockwise rotation that likely fed the Western Alboran Gyre (WAG), which was out of the picture.

The prevailing atmospheric synoptic conditions were inferred from ECMWF predictions of sea level pressure and zonal wind

at 10 m height (U-10), as shown in Fig. 8, c-d. A significant latitudinal gradient of sea level pressure was observed, with high pressures over the Gulf of Biscay and isobars closely spaced in the SoG, leading to extremely intense easterlies (above 10 ms-1), channeled through the Strait due to its specific geometric configuration. Therefore, high pressures and intense, permanent, and spatially-uniform easterlies prevailed over the entire study domain, inducing a westward outflow through the SoG as revealed by the 2-day averaged HFR circulation maps. Local wind forcing at this scale seemed to play a primary role in

explaining such AJ collapse and the related inflow reversals in agreement with previous studies (Garret, 1983; García-Lafuente et al., 2002; Menemenlis et al., 2007; Péliz et al., 2009; Reyes et al., 2015; Lorente et al., 2019b, 2019b; Bolado-Penagos et al., 2021).



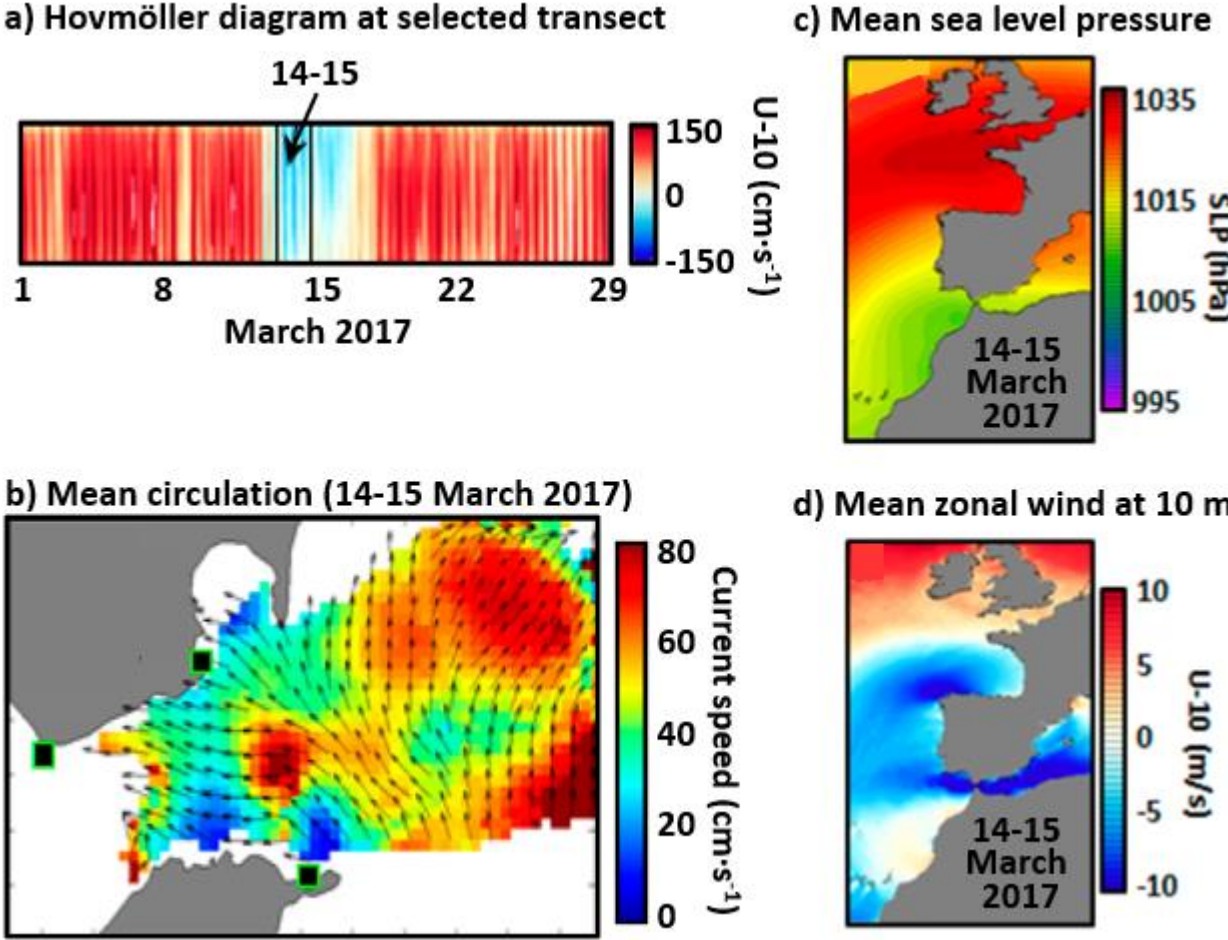

**Figure 8. (a) Monthly Hovmöller diagram of HFR-derived zonal current speed at the selected transect (shown in Figure 4.a). Red (blue) colors represent eastward (westward) surface flow. A 2-day episode of permanent flow reversal is marked (14-15 March). (b) HFR-derived mean surface circulation for 14-15 March 2017: permanent flow reversal. (c-d) 2-day mean sea level pressure and zonal wind at 10 m height (U-10), respectively, as provided by the ECMWF: intense and persistent easterlies were the driver of the flow reversal. Original source: from Lorente et al., 2019b.**

### 2.2.2 Tsunami detection

Tsunami early warning and alert is an emerging and promising application of HFR. The main principle underlying the detection is that the abnormal surface current pattern induced by the orbital velocity of the tsunami wave train can be measured and interpreted in real-time by an appropriate detection algorithm. The idea was first proposed by Barrick (1979), but it was only after the 2004 Indian Ocean disaster that the proof of concept was made on the basis of actual HFR data. It was shown numerically with simulated (e.g. Lipa et al., 2006; Gurgel et al., 2011) and real (e.g. Lipa et al., 2011 & 2012; Dzvonkovskaya et al., 2012) events that the tsunami signature could be clearly seen in the HFR radial currents and some appropriate detection algorithms were proposed. One strong point of HFR tsunami detection is that it is not bound to the nature of the source (seismic





or atmospheric) and can be used as a useful complement to other warning systems in places where those are either not available

or non-effective. Today, more than 20 real tsunamis have been detected 'offline' with the reanalysis of HFR data. In view of

the growing interest for these new capabilities of HFR, some radar manufacturers now provide commercial toolboxes along

with their hardware system for the early detection of tsunamis; such systems have been installed in some places at risk (e.g.

Vancouver Island Canada, Oman, New Jersey USA, SW Portugal). To date, the only real-time detection was issued following

a meteo-tsunami that occurred on 1 October 2016 in Tofino, BC, Canada (Dzvonkovskaya et al., 2017; Guérin et al., 2018).

However, no such HFR tsunami alert system has been yet installed in the Mediterranean Sea. Nevertheless, there is a non-

negligible tsunami hazard in this region, as witnessed by very destructive co-seismic events in recent history (e.g. Messina,

Sicilia, 1908). Some worst-case scenarios with a strong (M7.8) earthquake in the North Algerian margin predict important

tsunami waves with 3-4 meter amplitude on the French-Italian Riviera (BRGM, 2007). Moderate earthquakes such as the M6.9

21 May 2003 Boumerdes-Zemmouri are sufficient to cause 1-3 meter amplitude harbor oscillations within 40-60 minutes in

the Balearic Islands, which would be the most impacted spot by seismic sources in North Algeria (Wang et al., 2005; Sahal et

al., 2009). The impact in the French Mediterranean coast impacted by a seismic source in the North Algerian margin is shown

in Fig. 9.

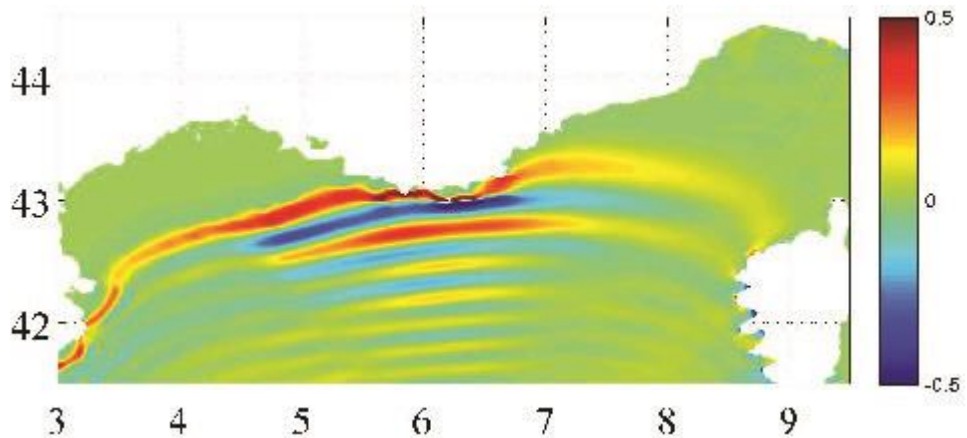

**Figure 9.: Simulated surface elevation (in meter) after 1h10 propagation for a tsunami generated by a M7.8 seismic source in the North Algerian margin. The numerical simulation uses FUNWAVE-TVD software with 3 nested grids in the West Mediterranean basin (courtesy of Stephan Grilli, University of Rhode Island, USA).**

In addition to co-seismic tsunamis, frequent meteo-tsunamis (i.e. tsunamis of meteorological origin) have been reported in the

Balearic Islands, named –"rissaga"- (Jansa, 2007), Adriatic Sea – "šćiga"- (Vilibić and Šepić 2009; Orlić, 2015), Sicily Channel

–"marrobbio"- (Candela et al., 1999; Zemunik et al., 2021), Malta – milgħuba"- (Drago, 2008), northern Persian Gulf

(Kazeminezhad et al., 2021), Black Sea (Vilibić et al., 2021) and Aegean Sea (Papadopoulos, 1993). Even though these events

have limited regional impact, they can cause severe local damages in harbors and bays due particularly to the micro-tidal

regime, resulting in rapid sea level changes (Vilibić et al., 2021). Indeed, the strongest known meteo-tsunami in the



Mediterranean Sea (and likely in the world) was in the Adriatic, the so-called Great Vela Luka Flood (June 1978), with an
amplitude of 6 m and periods of about 20 min, as detailed by Vučetić et al. (2009), closely followed by the event from
Ciutadella Harbor (Menorca island, Spain) in June 2006 (Jansa et al., 2007) with 4-5 m of amplitude. They are caused by
atmospheric disturbances combined with several further possible amplification mechanisms of the induced sea surface wave;
these are mostly the so-called Proudman-, Greenspan-, shelf- resonances in coastal areas which can lead to strong harbor
resonances in semi-closed basins (Orfila et al., 2011). Today, the generation of these meteotsunamis is better understood
(Monserrat et al., 2006; Šepić et al., 2009, 2015; Vilibić and Šepić 2009; Ličer et al., 2017) but their prediction is still a very
challenging task (Denamiel et al., 2019, Romero et al., 2019, Mourre et al., 2020).

When located in the areas affected by the meteotsunamis, HFR-based tsunami early warning systems could be a useful
complement to these forecasting systems, helping issuing specific alerts on the basis of the actual observed surface currents
20-40 km offshore a few minutes before the generation of the extreme sea level oscillations. Note that tsunami early warning
systems only require a software update of existing HFR and could be installed at reduced cost in some places. However, some
strategic spots are not covered yet and would deserve a novel installation to monitor the travel directions of incoming waves
from the most probable sources (North Algerian earthquake, West Corsica submarine mass failure, North Ligurian earthquake,
etc.). Another related issue is extending the range of these HFR, which would imply operating at lower frequency bands (4-5
MHz or 9 MHz) than those usually employed in the Mediterranean region (13, 16 or 25 MHz). Nevertheless, a software for
tsunami detection and warning has been installed in the HFR site located in Sagres (Portugal), which operates at 13 MHz
(Roarty et al., 2019). Such a prototype operating at 4.5 MHz with 200-300 km range was developed by the Diginext Ltd. a few
years ago (i.e. Stradivarius radar) on the French Mediterranean side to monitor a large part of the Gulf of Lion. A proof of
concept was made that such radar could provide early alerts of plausible tsunamigenic sources in the Mediterranean Sea (Grilli
et al., 2015). Such HFR systems can serve the double purpose of warning and characterizing abnormal surface current patterns
arising from tsunami-like waves of seismic or atmospheric origin. As recently suggested, they can also be used as proxies for
the observation of low-pressure fronts of atmospheric gravity waves that could lead to storm surge, if not meteo-tsunamis
(Domps et al., 2020).

**2.3 Environmental Transport Processes**

In the Mediterranean, as elsewhere in the world, the coastal zones serve as the main entry point of nutrients, pollutants and
sediments into the ocean, being the multi-scale coastal ocean dynamics key drivers for their transport, also impacting their
dispersal and retention and the cross-shelf exchanges. HFRs have demonstrated to provide very valuable measurements to
continuously monitor the mesoscale structures and frontal dynamics that organize the coastal surface flow and associated
transport, by the developments in the understanding of Lagrangian dynamics from HFR data (Rubio et al., 2020). The
comprehension of the coastal ocean conditions and variability underlying ocean productivity that correlate with fish stock
abundance, fish recruitment in coastal areas, dispersion and retention of larvae, etc., is critical for a sustainable management



of fisheries resources (Sciascia et al., 2018), being also essential to assist water quality management by tracking source and drift of pollution (e.g. chemical, sewage, oil spills or harmful algal blooms). In the Mediterranean Sea, the applicability of HFRs in this field is particularly relevant since, in addition to its limited exchange with the oceanic basins, its microtidal character and its intense internal meso and submesoscale circulation reduces the potential of dilution and dispersion of

dissolved and particulate wastes, maximizing the impact of one of the most commonly identified threat (e.g. marine litter and contaminants). In addition, despite being considered one of the most oligotrophic areas in the world ocean, it is also one of the world's hot spots for biodiversity (Coll et al. 2010, Gabrié C., et al. 2012), providing vital areas for the reproduction of pelagic species (e.g. Atlantic bluefin tuna, white shark and sea turtles) and hosting sensitive ecosystems in the shallow coastal waters.

### 2.3.1 Pollution and floatables tracking

Coastal areas are very sensitive to pollution impact, in terms of environmental and ecosystem impact as well as economic and societal consequences, and can be both source and target of pollution. Coastal regions in the Mediterranean Sea are heavily inhabited with strong tourist and maritime activity resulting in human and industrial waste intentionally or accidentally dumped into coastal waters. These pollutants evolve according to their chemical transformation over time and are caught in the 3D general circulation and carried offshore and to other distant coastal areas by currents. For example, heavy metals or other

chemical contaminants that may be present in semi-enclosed harbors (Tessier et al., 2011) could have important consequences on the ecosystem, and plastic litter is today a massive and particularly harmful component (e.g. Ryan et al. 2009; Declerck et al., 2019) of the marine pollution in the Mediterranean Sea.

Coastal and littoral areas are also vulnerable target regions for pollution. Detecting, monitoring and cleaning up oil slicks following an offshore spill before they reach the coast is a major challenge. Consequently, monitoring, understanding and

forecasting coastal dynamics is a critical step to develop adequate strategies to mitigate the effects of pollution in marine environments, from or towards coastal areas. However, forecasting coastal dynamics is one of the most challenging issues in geosciences due to their strong space-time variability as well as the complexity of the processes controlling the dynamics that interact simultaneously over a broad range of time–space scales, as previously highlighted in Sect. 2.1.2. Thanks to the fast growth of HFR as a key element of coastal observing systems, coastal currents can nowadays be measured in relatively large

coastal areas providing a regional synoptic view of the surface dynamics with high spatial and temporal resolution.

Therefore, transport properties of the surface flow can be analyzed using continuous HFR observations by means of different diagnostics based on the Lagrangian approach. Recent studies have demonstrated the potential of this land-based remote sensing technology for different applications in the field of tracking oil spills (Abascal et al., 2009), marine litter (Declerck et al., 2019) or phytoplankton (Hernández-Carrasco et al., 2018a). Transport and mixing properties can be studied from the

Eulerian or the Lagrangian approach. While the Eulerian perspective describes the basic characteristics of the velocity field, the Lagrangian approach addresses the effects of this field on drifted particles or tracers. Lagrangian diagnostics have the advantage of exploiting the spatial and temporal variability of a given velocity field. They can even unveil the presence of



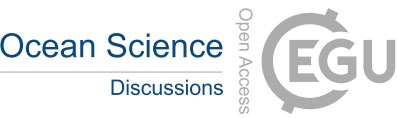

filaments at sub-grid scales generated by the chaotic mixing, providing an improved description of transport phenomena in geophysical flows and complementing the Eulerian metrics. The Lagrangian framework indeed allows to effectively track the

physical processes occurring along the history of the fluid parcels, which clearly is of utmost relevance for biogeochemical and biological dynamics (Hernández-Carrasco et al., 2018a; 2020) or for the tracking of pollutants, such as plastics, or for modeling the interaction between different substances, e.g. biofilm-covered microplastics with marine biogenic particles (Michels et al., 2018).

In this regard, a passive tracer $p$ on the ocean surface is advected by a velocity (Hernández-Carrasco et al., 2011; Sayol et al.,

2014) calculated following Eq. (1):

$$x'_p(x,t) = u_p^{adv} + u_p^{dif} \qquad (1)$$

where the two terms at the right-hand side state for the advective and diffusive components of the total velocity. When chemical pollutants are considered, additional processes should be included. For instance, when simulating oil spill trajectories, the advection term is a linear combination of the surface current velocity (from HFR or Ocean General Circulation Models -

OGCM-), the wind velocity and the stokes drift, and spreading, evaporation and emulsification should be included in the transport model.

Leaving aside the diffusive component that can be parameterized by an empirically determined diffusion coefficient and a random walk model, the two dimensional position of the particle at the ocean surface Eq. (2) can be obtained by integrating the velocity Eq. (1):

$$x_p(t + \delta t) = x_p + \int_t^{t+\delta t} u_p^{adv} dt \qquad (2)$$

Thus, by integrating the measured velocities $u_p^{adv}$ $(lon, lat, t)$ from the HFR, we can track (either forward or backwards in time) the evolution of floating debris at the ocean surface.

Here we provide evidence to support the reliability of the HFR currents for tracking substances at coastal areas. The Lagrangian validation has been performed using data from 8 drifters' trajectories available in the domain of the HFR area of coverage in

the Ibiza channel (HFR operated by SOCIB, Tintoré et al., 2020) during October 2012. Moreover, we use the HFR velocity fields to compute the Lagrangian Coherent Structures (LCS) which are very suitable to provide a template of the fluid flow transport (see Haller, 2015, and references therein), allowing the detection of transport barriers, which are of great relevance for marine dynamics. For example, LCS obtained from ridges of the Finite Size Lyapunov Exponents have been correlated with filaments of remote-sensed Chl-a (Lehahn et al., 2007; Hernández-Carrasco et al., 2014, 2018a, 2020), sea bird foraging

behavior (Tew Kai et al., 2009), with the modelled extension of oxygen minimum zones (Bettencourt et al., 2015) and with wind forcings (Berta et al. 2014b). At coastal scales, the dynamical picture in the Lagrangian frame has been analyzed using data from HFR currents to identify relevant small-scale transport barriers (Lekien et al., 2005; Gildor et al., 2009; Rubio et al., 2018), some of them focusing on coastal areas of the Mediterranean Sea (Haza et al., 2010; Berta et al., 2014b; Hernández-Carrasco et al., 2018a). As seen in Fig. 10, by integration of Eq. (2) for a set of virtual neutrally buoyant particles initially

deployed on the northern and southern flank of a given LCS measured from the HFR-Ibiza in Jan 25th, 17:00 UTC in 2013





(Fig. 10a). Although the location and magnitude of this LCS evolve in time, the LCS persists for several hours manifesting the presence of a coherent transport barrier preventing both sets of particles to be mixed up. A meridional LCS is formed and maintained during the simulated period, limiting water exchanges between the coast and the open ocean (Fig. 10a–c).

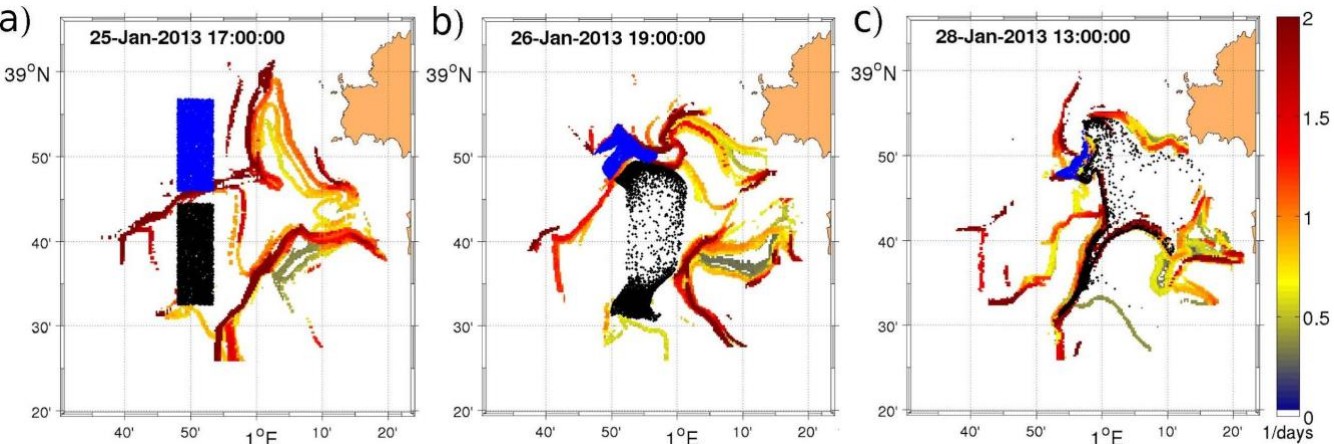

**Figure 10. Evolution of two sets of particles (black and blue) in the area covered by the HFR-Ibiza in January 2013 superimposed on the backward FSLE (colorbar). The virtual particles are initially deployed at both sides of a barrier revealed by a zonal LCS on Jan 25th, 2013 at 17:00 evolving for 68 hours.**

Besides, coastal waters are enriched in nutrients from river outflows, sediment resuspension and coastal upwelling, that can also contain increased quantities of pollutants (e.g. floating marine litter). The physical mechanisms that contribute to the offshore transport of these mesotrophic coastal waters to the oligotrophic offshore areas are critically important for boosting oceanic primary production and sustaining the trophic chain. The mechanisms that can influence the escape times of these waters in a target area can be monitored by means of the Lagrangian properties derived from the HFR by means of the residence times and the escape rate/times (Rubio et al., 2020). Using as input gap-filled HFR velocity fields a Lagrangian Particle-Tracking Model (Eq. 2) provides the particle trajectories. From the Lagrangian model outputs, it is possible to infer the characteristic time-scales for transport processes in the HFR footprint area by means of the escape rate of active particles (Fig. 11). Thus, HFR shows to be an excellent tool to monitor conditions and identify the different scenarios that favor the local retention and dispersal of shelf waters in two study areas under the influence of ocean boundary currents.





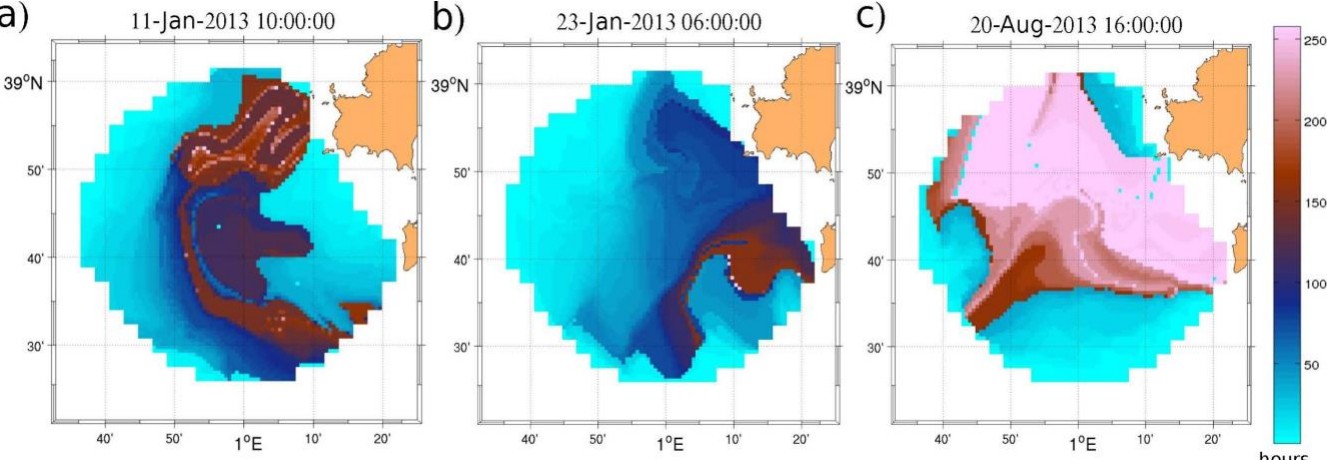

**Figure 11.  Maps of particle residence times (hours) computed for different dates and seasons, from HFR observations in the**
**Ibiza Channel SOCIB (a,b for winter and c for summer conditions). Original source: from Rubio et al., 2020.**

### 2.3.2 Eddy tracking

Ocean eddies are ubiquitous, pervasive flow structures which dominate the ocean velocity field at several scales, from the
meso- to the local scale (Chelton et al., 2011). They play a fundamental role in sea dynamics, being responsible for the energy
transfer among different scales (down to the dissipative range) as well as for their ability to transport nutrients, biomass,
sediments and pollutants. Mesoscale eddies, produced by geostrophic instabilities, are not able to advance the energy transfer,
being constrained by geostrophic and hydrostatic balance (Charney, 1971). When the balance is broken, the downscale may
continue through inertia-gravity waves emitted from currents, ageostrophic instabilities, and bottom boundary layer turbulence,
which are responsible for the formation of submesoscale eddies. At a lower scale, three-dimensional turbulence proceeds
toward the dissipative range (McWilliams, 2019).

The presence of ocean eddies has become more evident in recent years, thanks to the introduction of new oceanographic
measurement techniques, while their exhaustive characterization would require synoptic time series of the velocity field in the
ocean (Robinson, 1983). Such synoptic observations are made available only through satellite data, however, besides being
limited to relatively large scales, and preferably to the open ocean, they do not provide direct measurements of the total velocity
field. Indeed, altimetry data can be used to retrieve the surface geostrophic field, which lacks a possibly important portion of
the dynamics (Rinaldi et al., 2010; Conti et al., 2016). Other ways to observe eddies from satellites consist in the observation
of their presence in the sea surface temperature or in the tracer field patterns, as displayed by ocean color (Robinson, 2010).
Coastal HFRs overcome all the above limitations and provide direct measurements of the total surface velocity field at high
spatial and temporal resolution, thus enabling to detect and follow the time history of surface eddies down to submesoscale, at
the cost of a reduced spatial extent.

Mandal et al. (2019) provide quite an extensive list of recent literature reporting submesoscale features observed by HFRs,
with examples of observations in various coastal areas, from the Atlantic to the Pacific and Indian oceans (Shay et al., 1995,



2000; Kirincich, 2016b; Archer et al., 2017; Lai et al., 2017; Arunraj et al. 2018). It is worth noticing that such features may have a strong vertical signature that HFR data fail to account for, and therefore need to be complemented with further information, spanning from direct measurements of the horizontal and vertical velocity profile to indications indirectly derived

from, e.g., satellite turbidity measurements (see discussion in Uttieri et al., 2011).

A vast HFR network now covers the North-Western Mediterranean coastal areas (see Lorente et al., submitted to this Special Issue), which is characterized by significant mesoscale variability and eddy generation that, in some cases, shows recursive and seasonal patterns. Allou et al. (2010) used HFR to observe and characterize vortex structures, mostly anticyclonic, in the Gulf of Lion. They also argued they were correlated with specific wind patterns. Schaeffer et al. (2011) improved the analysis

and employed both HFR measurements and numerical modeling to analyze the eddy generating mechanism. They found it is primarily influenced by wind forcing and its interaction with topographic constraint (northerly offshore wind), and freshwater input from Rhone river (southerly onshore wind). The combination of HFR and *in-situ* observations, and modelling tools, allowed Guihou et al. (2013) to identify an anticyclonic coastal eddy which was generated in front of Nice by a meander of the Northern Current, and advected downstream toward the Toulon area, interacting with the mean circulation. More recently,

the analysis of the long 2012-2019 HFR time-series in Toulon allowed to identify cyclonic and anticyclonic recurrent eddies mainly generated by wind and boundary current undulations (Bourg and Molcard, 2021).

The development of monitoring networks providing long time series of data, is making automatic eddy detection methods more and more topical and important. Generally speaking, existing eddy detection algorithms can be divided into three families: (i) those that are based on the geometrical features of the velocity field, typically in terms of streamline closeness,

winding angle or vector geometry (Sadarjoen et al. 1998; Heiberg et al. 2003; Ebling and Scheuermann 2003; Nencioli et al. 2010); (ii) those based on dynamical characteristics, such as parameters quantifying the eddy intensity, its vorticity, etc. (Jeong and Hussain 1995; Fang and Morrow 2003; Isern-Fontanet et al. 2003; Morrow et al. 2004); (iii) and hybrid methods, based on the combination of geometric and dynamical criteria (Mkhinini et al. 2014; Conti et al., 2016, see also the extensive review in the paper by Le Vu et al., 2018).

Algorithms specifically devised for HFR data are very few. The methods tested in the Mediterranean Sea (Caldeira et al., 2012) are limited to those by Nencioli et al. (2010), and by Bagaglini et al. (2020). The former, even though developed for HFR data (and for high resolution numerical model outputs), has found a widespread range of applications to observations collected by different platforms, as witnessed by current oceanographic literature (Liu et al., 2012; Dong et al., 2014). It is a method based on the geometry of the velocity vectors. It was conceived for geostrophic or quasi-geostrophic recirculating features, showing

very little divergence. For this reason, it is very suitable to describe mesoscale eddies, but may fail in detecting submesoscale ones, which often are characterized by divergence or convergence and by a high degree of deformation of the velocity field geometry. The YADA (Yet Another eddy Detection Algorithm) algorithm developed by Bagaglini et al. (2020) was conceived specifically to overcome this limitation and be utilized to automatically detect submesoscale eddies, which may exhibit highly non-geostrophic characteristics. It is a hybrid method, which focuses on both the dynamical and geometric features of the

velocity field, first identifying the local extrema of a dynamical field characterizing recirculation (e.g., the local normalized





angular momentum, see Mkhinini et al., 2014; or the Okubo-Weiss parameter, Okubo, 1970; Weiss, 1991), similarly to the first step from AMEDA (Angular Momentum Eddy Detection and tracking Algorithm) defined by Le Vu et al. (2017), and thereafter analyzing the streamline geometry in a neighborhood of the extremum. The YADA (Bagaglini et al., 2020) has been successfully applied to 1 km-resolution HFR data from the Gulf of Naples, showing its ability to identify strongly asymmetric,

convergent or divergent submesoscale eddies. Its application to coastal HFR data from other areas of the Western Mediterranean is presently under way.

The HFR system from LaMMA Consortium (described in Lorente et al., submitted to this Special Issue) covers part of the Ligurian Sea and the Tuscany Archipelago, which is a shallow area separating the Ligurian and Tyrrhenian basins, bordering eastward the Corsica Channel, with complex topography and coastal morphology, also due to the presence of several islands

(Elba, Capraia, Montecristo, Gorgona). Sea dynamics are strongly influenced by seasonality and characterized by the presence of the Tyrrhenian boundary current and its bifurcation, the Eastern Corsica Current (Astraldi and Gasparini, 1992; Millot, 1999; Vignudelli et al., 2000). Through drifters, *in-situ* data, and a numerical model, Poulain et al. (2020) studied the area in the summer season. They found prevailing southward current flowing next to the Italian coast, then turning westward and northward encountering Elba island. A further eastward motion led to the formation of anticyclone centered on Capraia island,

which exhibited variations correlated to wind forcing. The presence of an anticyclone north of Corsica Channel, in summer and autumn seasons was previously documented by Ciuffardi et al. (2016) by means of *in-situ* profiles and altimetric data. Furthermore, they argued that the characteristics of the anticyclone (position and size) may affect the general circulation, by isolating the Tyrrhenian and Ligurian basin, mostly in summer. These hydrodynamics features can influence the concentration of floating marine litter, which was shown to be particularly high in certain periods of the year (Fossi et al 2017).

Here, we show the application of both the YADA and Nencioli et al. (2010) algorithms to the surface currents derived from the HFR system of the LaMMA consortium, during the year 2019.

Figure 12 reports four eddies detected by the YADA algorithm, and the corresponding surface currents in August 2019.



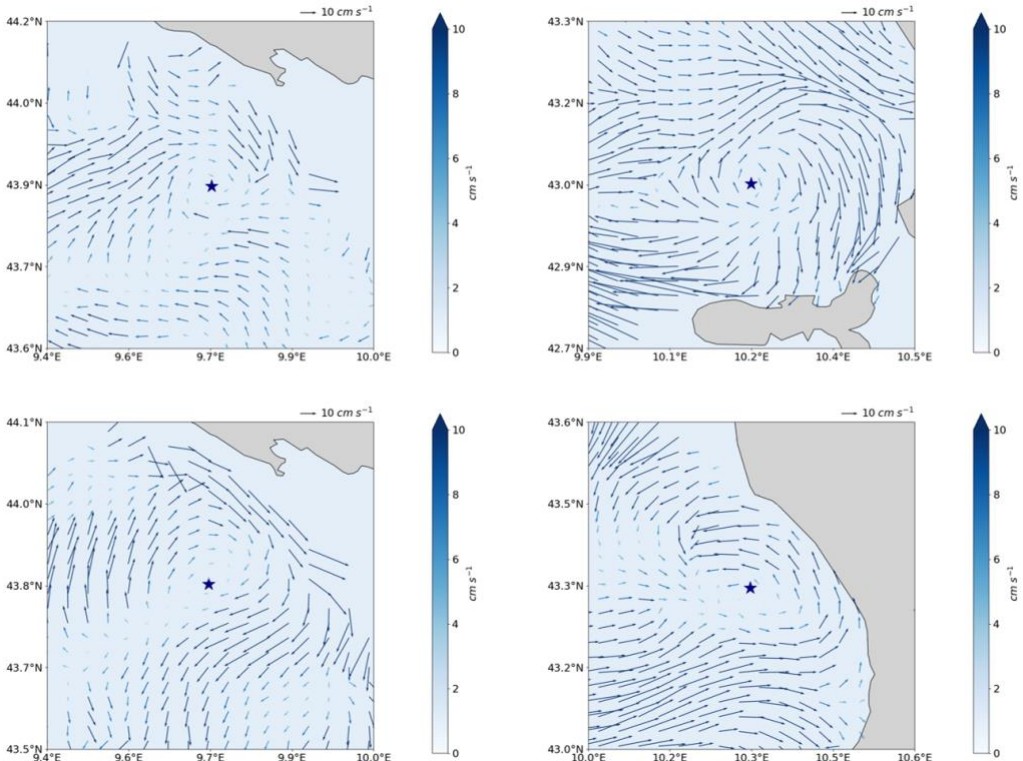

**Figure 12. Maps of the area of eastern Ligurian Sea and Tuscany Archipelago showing the HFR derived surface currents (colored arrows indicating the current speed) and the detection of four eddies in early August 2019 by the YADA algorithm. The blue star marks the eddy center.**

Furthermore, Fig. 13 shows the results of the application of the algorithm by Nencioli et al. (2010) to the whole year 2019, which provided a seasonal census of anticyclonic and cyclonic eddies in the area sampled by the LaMMA HFR system. The HFR coverage was not uniform during the year with generally lower percentages for the warmer seasons (i.e. spring, summer). The area north of Elba island was characterized by the highest eddy activity throughout the year, with a predominance of anticyclonic eddies in colder seasons, and a clustered pattern in summer, showing anticyclonic eddies east of Capraia island and cyclonic ones toward the coast.

The number of detected eddies depended also on the availability of HFR data; indeed, a smaller number of eddies was found in spring (28) with respect to the other months (62 winter, 68 summer, 62 autumn). Median eddy life-span was around 1.125 days for all seasons, even if they were able to survive up to 6 days, except in spring.





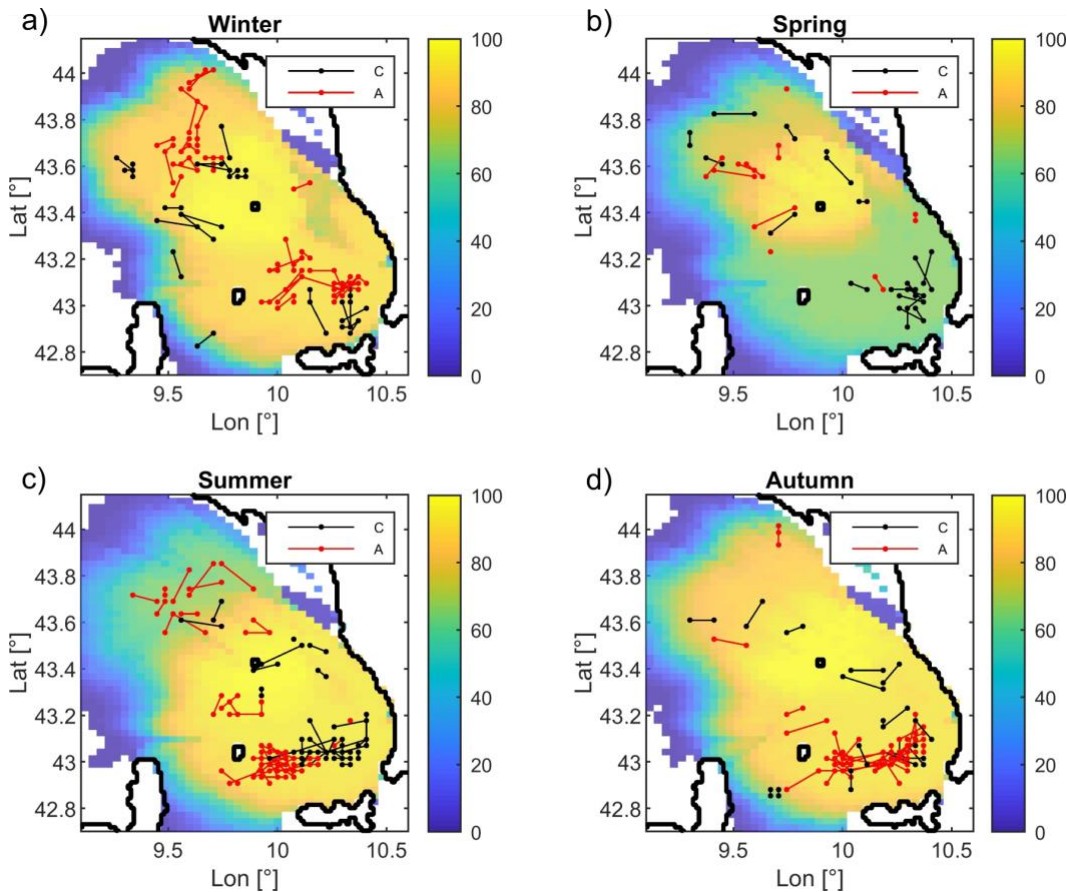

**Figure 13. Maps of the area covered by the LaMMA HFR network showing the percentage of HFR data availability along 2019 for a) winter; b) spring; c) summer and d) autumn. The Tuscany Archipelago (i.e. Elba island) is at the lower-right corner of the figures. Tracked cyclonic (black) and anticyclonic (red) eddies detected in each season are overlaid.**

Although the present work is preliminary, it may lay the basis for a detailed analysis concerning the seasonal features of eddy activity within the Tuscany Archipelago, and its effect on the general circulation. Surface currents from HFR can be combined with the numerical model outputs both to bridge the gap relative to the spatial and temporal coverage of data, and to improve reliability with respect to the actual sea dynamics.

## 2.3.3 Transport of biological quantities and connectivity

The necessity to preserve the marine ecosystem equilibrium and the water quality, has fostered the use of HFR data in supporting the coastal zone management and assessing the variability in the dynamics of marine ecosystems. In particular, HFR data have been used worldwide to address ecological and water quality issues such as: to understand the transport and retention processes of plankton or wastewater discharge in some regions of NW Spain (Ria de Vigo) (Piedracoba et al., 2016) and Western Mediterranean (Hernández-Carrasco et al, 2018a), at coastal upwelling fronts off central California (Bjorksted





and Roughgarden, 1997) and in Monterey Bay (Coulliette et al., 2007); to investigate the enhancement of productivity due to the retain of phytoplankton within the flow in the Santa Barbara Channel (Brzezinski and Washburn, 2011) and the relationship between populations of larval and juvenile fishes and the mesoscale flow field in the California Current System (Nishimoto and Washburn, 2002).

Here we present examples of application of HFR data to investigate ecological questions in three Mediterranean coastal areas: the Gulf of Naples (GoN, Tyrrhenian Sea), the Gulf of Manfredonia (GoM, western Adriatic Sea) and the Malta-Sicily Channel (mid Mediterranean Sea)

The continuous observations of the HFR current fields in the GoN, highlighted several characteristics of the surface circulation and water exchange between the interior of the GoN and the neighboring open Tyrrhenian Sea (Cianelli et al., 2013). An

oscillating plankton population dynamic has been also frequently observed in the GoN, at Long-Term Ecological Research station MareChiara (LTER-MC), where plankton abundance is monitored weekly at the since 1984. A proof of concept study (Cianelli et al., 2017), was thus conducted in order to characterize the spatial scales and the provenance of phytoplankton assemblages detected at LTER-MC and to dissect processes regulating plankton dynamics.

The study focused on a year-long analysis carried out for 2009, which was characterized by a very accurate estimate of the

surface dynamics, with a reduced number of gaps among ecological measurements and HFR data. The approach followed these conceptual steps: (i) Reconstruction of the annual and seasonal regimes of HFR currents detected at the LTER-MC site; (ii) Running Lagrangian backtracking simulations. Virtual phytoplankton patches (VPPs) were released at LTER-MC on the dates of the weekly oceanographic campaigns and tracked backward in time (up to 96 h earlier) to their zone of origin (Fig. 14, a); (iii) Identifying spatial scales of horizontal transport of virtual phytoplankton patches in the GoN (Fig. 14, b); (iv)

Comparison among backtrack Lagrangian reconstruction and ecological analysis based on salinity and chlorophyll a data obtained through weekly sampling at LTER-MC (Fig. 14, c); (v) Identifying different modes of coupled physical and ecological functioning in the GoN as resulting from physical transport and biological processes ('allogenic' and 'autogenic' factors, respectively).



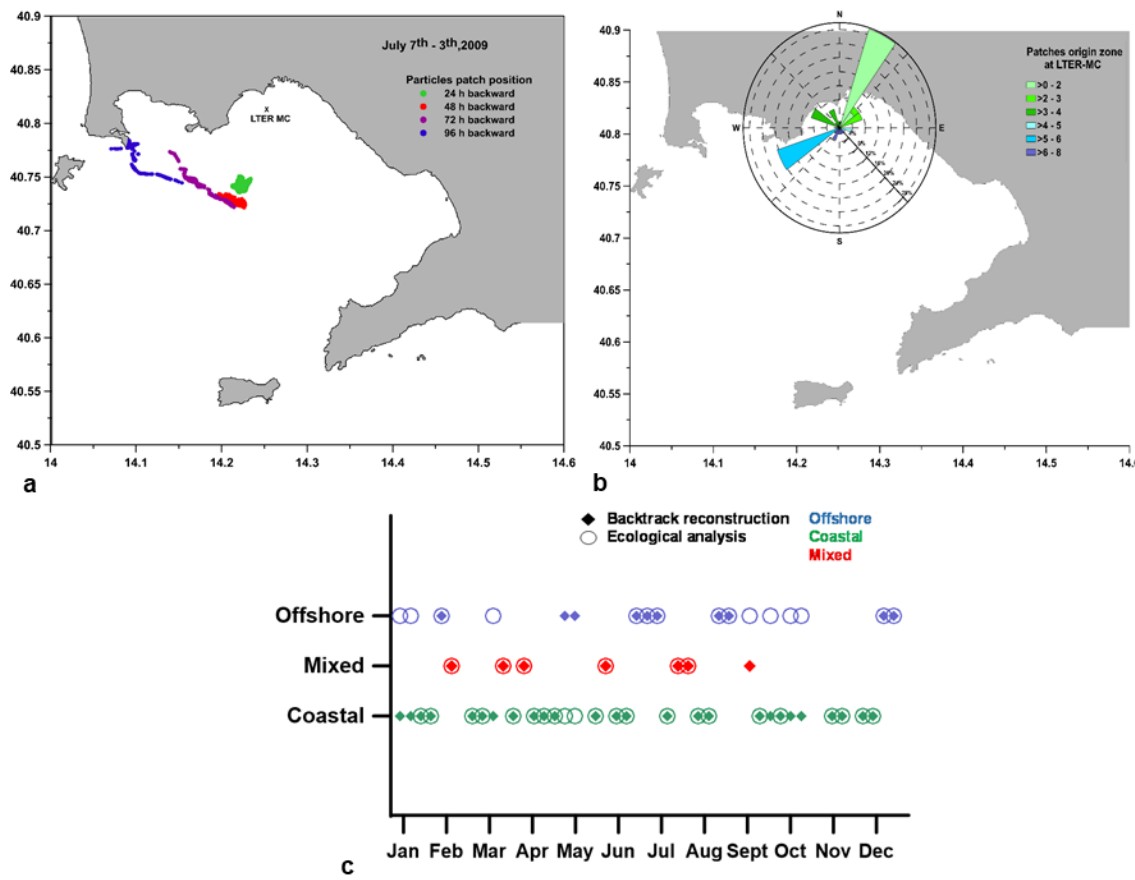

Figure 14.: a) Map of the Gulf of Naples (southern Tyrrhenian Sea) showing the: a) Lagrangian backtracking trajectories of Virtual Phytoplankton Patches (VPPs) in the period 7th- 9th June 2020 and b) main origin sectors of the VPPs at LTER-MC as resulting from backward simulations, index sectors of VPPs are 1-2-3-4-5 for coastal areas and 6-7-8 for offshore areas; c) comparison among backtrack Lagrangian origin and ecological analysis based on salinity and chlorophyll a data weekly measured at LTER-MC. Green = VPPs originating from coastal areas; blue= VPPs originating from offshore areas; red = VPPs originating partly from coastal and partly from offshore areas. Adapted from: Cianelli et al., 2017.

The results showed an alternation in plankton dynamics between phases reflecting the influence of coastal (green) and offshore (blue) circulation patterns on the biological community. The phytoplankton community detected at LTER-MC generally originated from the coast, whereas the offshore inflow marginally changed the main traits of phytoplankton assemblages. Back-tracking simulations and biological data strictly agree, highlighting that the plankton community at LTER-MC during 2009 is affected by the alternation of coastal and offshore influence.

Biological autogenic factors drive the modifications of coastal phytoplankton communities during the coastal 'green' phases, thus suggesting that the GoN tends to retain the same communities via coast-ward circulation, especially during summer. Physical allogenic factors are determinant in driving dilution and species advection of coastal phytoplankton, during the offshore 'blue' phase. This marked alternation between coastal and offshore water masses acts to promote phytoplankton





diversity, because the dilution in the phytoplankton density may decrease the impact of the dominating species over the available resources.

The integration of long-term biological data and high-resolution current fields represents an optimal tool to investigate the role of surface circulation in structuring the marine plankton community, thus confirming the value of HFR systems to analyze the seasonal fluctuations in marine ecosystems dynamics and to unveil the mechanisms of coastal connectivity.

The GoM is a well-known recruitment area in the Adriatic Sea (Sciascia et al., 2018; Corgnati et al., 2019a). In this region, HFRs have been used to understand the role of ocean currents in the recruitment of small pelagic fishes (i.e. European sardines, *Sardina pilchardus*). Fig. 15 shows residence times within the GoM, estimated using trajectories of virtual drifters computed from the surface currents measured by HFRs. Months with high (October)/low (February) residence times are associated with weaker/stronger surface currents in the central area of the Gulf. The relatively short (<12-day) average residence times have

shown that local spawning is less likely to take place than the transport to the Gulf from remote spawning areas through advection pathways. Results agree with otolith measures, suggesting that the arrival of larvae within the Gulf is characterized by repeated pulses from remote spawning areas that are likely to play a fundamental role in maintaining the nursery.

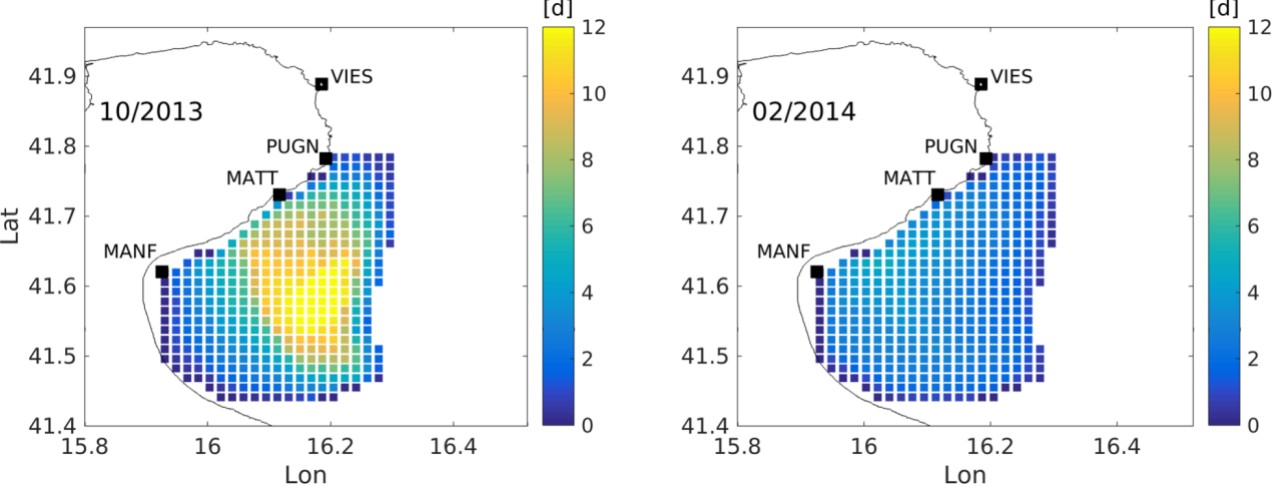

**Figure 15. Maps of the Gulf of Manfredonia showing monthly bootstrap estimates of average residence times (in days) of**
**virtual particles advected in the HFR velocity field and released within the Gulf of Manfredonia for the months of October 2013 (left panel) and February 2014 (right panel). Black squares represent the locations of the four HFR antennas along the Manfredonia gulf. Adapted from: Sciascia et al., 2018.**

The Malta-Sicily Channel is both the most fished area in western Mediterranean Sea and a very important hotspot of biodiversity (Médail & Quézel, 1999). In this region, Capodici et al., (2018) combined the HFR-CALYPSO surface currents
together with satellite images of chlorophyll-a concentration -CHL- and Sea Surface Temperature -SST-, to explain physical driven mechanisms that can help interpret ocean productivity and plan maritime activities in a more adaptive and proactive way. On the one hand, as mentioned by the authors, monitoring water quality data provides valuable insight into the understanding of processes driving spatial and temporal changes in productivity at sea (Behrenfeld et al., 2006). CHL and SST





are generally accepted as proxies for water quality and are very helpful to detect upwelling events, which are frequently
occurring along the Sicilian coast. In particular, sea currents are responsible for dispersion, transport or retention of nutrients
of which CHL is widely used as a proxy variable; moreover, current jets or eddies are often observable as cold or warm areas
in the SST maps, respectively. Even if both CHL and SST maps are usually retrieved by means of satellite data maps,
cloudiness often reduces the satellite data availability; thus, temporal aggregated products (e.g. at 8 or 16 days) are the only
data available. In this framework, the integration of sea surface current data provided by HFRs can fill the gap of knowledge
due to the inadequate temporal (and sometimes spatial) resolution of these water quality maps. Capodici et al., (2018) used the
Principal Component Analysis -PCA- (Preisendorfer, 1988) to firstly extract the dominant spatial patterns of these variables
along 2013 and to secondly quantify the degree of correlation between SST (or CHL) and HFR derived surface current spatio-
temporal patterns. The spatial correlation analysis suggests the importance of current advection in the phytoplankton transport,
being characterized by fringes where very high positive correlation areas are surrounded by very high negative ones and
viceversa.

Moreover, the spatial distributions of time-averaged radar currents and corresponding TKE (Total Kinetic Energy), EKE (Eddy
Kinetic Energy) and the absolute value of the products between temporal fluctuations of the deviations from the time-averaged
zonal and meridional velocity components (ReS), shown swirling areas for the first time at high spatio-temporal resolution.
The trapping zones, characterized by low values of TKE, EKE and ReS at their cores, may help explain why this channel is
particularly rich in pelagic species, highlighting how the continuous high resolution HFR derived surface currents can improve
the decision-making process in spatial management.



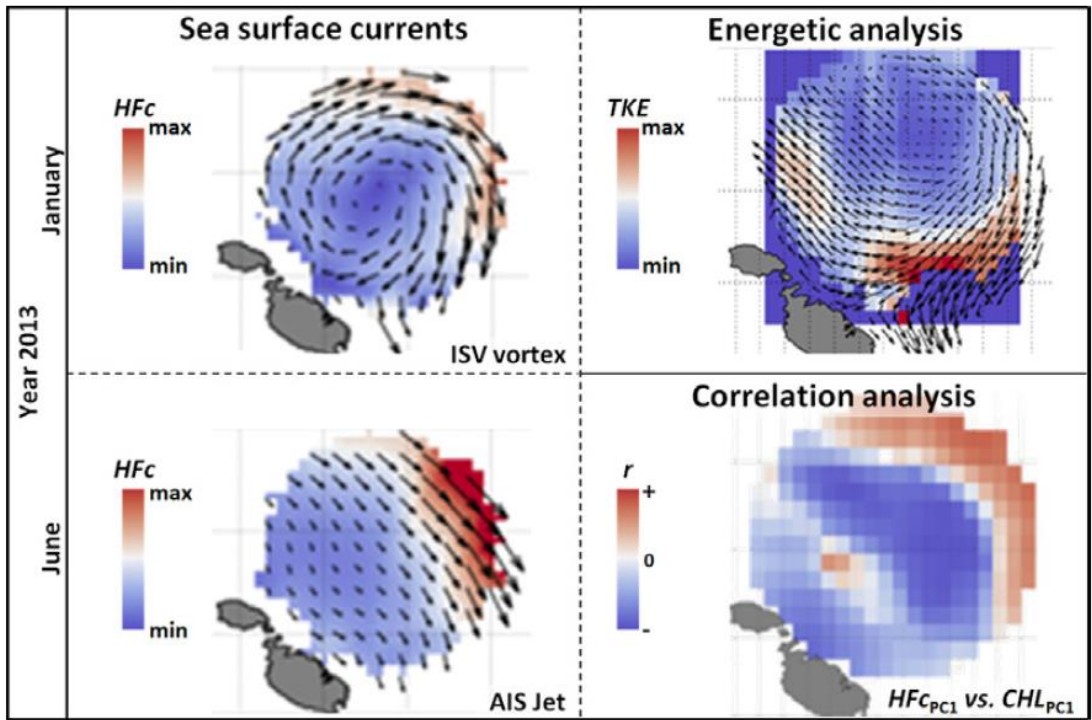

**Figure 16. Map of the Malta-Sicily Channel showing the monthly averaged HFR sea surface currents for January and June**
**2013 (top left and bottom left panels, respectively), spatial distribution of the Total Kinetic Energy (top right panel) and of**
**the correlation coefficient between the first PC of both HF and CHL (bottom right panel). Original from: Capodici et al.,**
**(2018), graphical abstract.**

## 3 Discussion and preliminary assessment of the HFR regional capabilities

This initial inventory of HFR applications implemented in the Mediterranean coastal areas have allowed us to know their
strengths and weaknesses to further contribute to the regional observing system in addressing the regional environmental
threats, the scientific priorities and the society needs. Considering also the current threats and the opportunities ahead along
this decade, we have conducted a SWOT (i.e. strengths, weaknesses, opportunities, and threats) analysis, as schematized in
the Fig. 17, and as discussed in this section for each one of the three challenges: i) maritime safety; ii) extreme hazards and iii)
environmental transport process.

In the sphere of maritime safety, the main strength of HFRs is the provision of high spatio-temporal resolution surface currents
in wide coastal areas, where most of the SAR incidents occur. HFRs complement other scarce observations, helping to assess
and to improve the ocean models, through data assimilation, or being used as alternatives of the models for backtracking
drifting objects in near-coastal risk-prone environments. In addition to that, the machine learning approach where a neural
network model is trained on past data and then used to create short term predictions is gaining ground in recent years. The
steadily growing of the European HFR network (Rubio et al., 2017; Roarty et al., 2019), increasing both the coastal coverage





in many countries and the length of the time series, will allow us to implement these self-learning algorithms in other Mediterranean areas. Nevertheless, there is a strong need for consensus on the methodology to generate these scientific added-value products, and on the definition and further adoption of a common data and metadata model as well as the quality control tests, to operationally distribute standardized HFR gap-filled data and derived STPs, helping to unlock HFR data potential for
their use in several Lagrangian applications. Moreover, the use of the HFR data for data assimilation and model assessment should be boosted through the provision of data uncertainties (Moore et al., 2019).

As a land-based remote sensing technology, HFRs are able to continuously monitor the surface's coastal ocean response to extreme events, by providing near real-time information of the surface's ocean response on a continuing basis. In this context, HFR strength is based on its no deployment requirement, avoiding the risk to be faced by other observing platforms (i.e.
research vessels, ferry boxes or even autonomous instruments), being also unaffected by cloud cover and making also easier their routine maintenance tasks under these severe episodes. Furthermore, it is worth to be highlighted that under severe weather phenomena, near-real time met-ocean information gains value since it is essential to avoid risky situations and to support the emergency response at sea. This capability has allowed us to monitor and deeply investigate the impacts of intense wind episodes, severe river freshwater discharges and record-breaking storms as well as to observe the weakening or even the
reversal of main surface currents and jet streams. Furthermore, their demonstrated improved capacities to detect tsunami-induced currents make it a valuable complement to other warning systems in the Mediterranean. Nevertheless, and despite the existing tsunami risk and the frequent occurrence of meteo-tsunamis in the Mediterranean Sea, no HFR tsunami alert has still been installed in this region. Facing a growing interest in these HFR new capabilities, several challenges must be previously addressed regarding the installation of new systems to monitor the most probable source areas and the extension of the range
by using lower operational frequencies as usual in the Mediterranean Sea to be able to detect the tsunami-induced currents far offshore, to offer early warning.

The recognized capabilities of the continuous HFR observations to analyze the transport properties of the surface flow and to detect and track surface eddies down to submesoscale, have allowed us to understand the phytoplankton distribution, to identify different local retention scenarios, and to investigate the role played by the characteristic mesoscale
variability and eddy generation in the transport of biomass, pollutants and in the recruitment and abundance of small pelagic species in the Mediterranean coastal waters. In this regard, it has fostered the use of HFR data in supporting the coastal zone management and assessing the variability in the dynamics of marine ecosystems, becoming an asset of great value to contribute in the achievement of the Good Environmental Status (GES) of the Mediterranean waters. However, it has been found that the number of detected eddies depended also on the HFR data availability, highlighting the need to combine the HFR with
numerical model outputs both to bridge the spatio-temporal gap and to improve reliability. It should also be noted that the HFR limited coverage reduces the potential of the larger scale applications and connectivity studies, thus requiring their integration with other *in-situ* and satellite observations as well as models, being these applications also benefited from the future expansion of the HFR network.




Considering the high spatio-temporal resolution of HFR derived surface currents being one of its main competitive strengths
when compared versus other observing platforms, it must be recognized that its most serious limitation is that it provides
information at the very near surface layer. In order to fully understand ocean dynamics, the knowledge of the water column
processes is essential as well. A significant number of coastal ocean observatories in the Mediterranean Sea (as described by
Tintoré et al., 2019) encompass a complex multi-platform network including HFRs, aiming to meet the primary but challenging
need to monitor both the surface and the water column. This has motivated the development of techniques able to combine the
information of the processes in the entire water column, in order to provide a three-dimensional picture of the overall dynamics.
The combination of such observations is challenging mainly for two reasons: surface and ocean interior are prone to different
processes and forcings with different spatio-temporal scales and at, the same time, the capabilities to resolve and characterize
the diverse processes may be different for observing platforms at the sea surface and in the water column. Despite the promising
results obtained by Berta et al. (2018) and Guihou et al. (2013) in the Mediterranean Sea, necessary efforts must continue
towards the further development of methodologies to combine HFR data with water column measurements and models.

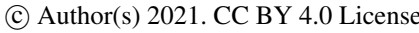

**Figure 17.** SWOT analysis of the HFR capabilities and applications.



## 4. Future prospects for HFR applications and recommendations

Finally, after the description of the current implementation status of the HFR applications in the Mediterranean coastal areas,
920 and based on the results obtained from the SWOT analysis, we present here the prospects for the future and a set of key recommendations. These recommendations aim to ensure that the potential of HFR is fully exploited in the development of operational monitoring systems at the regional level, helping also to derive the added-value achieved by the European HFR network (i.e. centralization of the data management and standard data distribution, new products development, cross-disciplinary emerging applications, training schools, etc). These recommendations should be part of the long-term monitoring
925 strategy for structuring the ocean observation at the regional level, for the development and integration of the COOSs towards addressing key scientific questions and meeting the societal challenges related to the Mediterranean coastal regions, being always aligned with the European strategies to ensure the integration.

Although the future prospects for the HFR data use and applications at the regional level are shared far beyond the geographical borders of the Mediterranean Sea, the regionalization of the recommendations constitutes a benefit: i) to coordinate cross-
930 national efforts; ii) to account for regional specificities (in terms of scientific key priorities, societal needs and existing environmental threats) and iii) to map the existing and potential HFR data end-users, also facilitating the interaction with them. Therefore, we specifically address them in this section, as follow:

**a) Expansion of the Mediterranean HFR network**: although the MONGOOS-HFR network is already representing the 55% of the HFR sites existing in the European inventory (Lorente et al., submitted to this Special Issue), and despite the very
935 recent installations (i.e. in Licata and Portofino -Italy-, in Port de Menton -France- and in Ta' Cenc in Gozo -Malta-), and the several new deployments in an initial planning stage (i.e. in Haifa Bay -Israel-, Sardinia and Sicily islands, Gulf of Genoa and Gulf of Naples, Tuscany Archipelago -Italy-, Gulf of Trieste -Slovenia- and in the Aegean -Greece-), the Mediterranean coastal areas are still under-sampled. In this regard, it should be noted that the spatio-temporal scales currently provided by the HFRs in the Mediterranean allow us to monitor the current environmental threats adequately, but always in limited coastal areas,
940 thus reducing the potential of the larger scale applications (e.g. transport of organic matter and pollutants, connectivity studies, data assimilation into models, etc). Therefore, aiming to improve regionalization of the observatories for a better understanding of region-specific processes towards a fit-for-purpose design, an increased monitoring effort by expanding and improving the HFR network is required, allowing for a covering of a large geographical area on a routine basis. Accordingly, it is needed a previous review of major scientific and social questions, the environmental stressors and their impacts in the Mediterranean
945 waters and blue sectors, identifying the benefit of the new deployments, in coordination with the current monitoring actions (e.g. to identify gaps for monitoring these risks, ensuring cost-effectiveness of observations, etc). To this end, the cross-border coordination activities are key for the involvement of other countries bordering the Mediterranean Sea in the eastern and southern coastlines and to address issues related to frequency sharing for avoiding interferences, as highlighted in Lorente et al., submitted to this Special Issue.





**b)  Reinforcing the Mediterranean's leadership** for continuing to be a major European focus of HFR activity. The Mediterranean institutions are key players in the European and international HFR research effort in multidisciplinary fields and in the development of applications, as mostly covered in the Sect. 2 of this work. In addition to that, HFR experts from the Mediterranean institutions are currently and actively contributing in the definition of the European HFR network roadmap, leading crucial tasks aiming to define the standard model, increase the availability and accuracy of the HFR wave parameters

and to reach a consensus on the methodology for the provision of the HFR data gap-filling products (Rubio et al., 2021). The creation of partnerships between different research groups and institutions in the context of European, regional and national projects (Lorente et al., submitted to this Special Issue) is contributing to move forward both the expansion of the HFR network and the main research areas, also fostering the HFR data interoperability and distribution, helping to unlock the HFR data potential, discovery and usage.

**c)  Keep promoting the HFR data interoperability and distribution at the regional level:** as a result of international and European efforts made in recent years towards the HFR data harmonization and distribution, it has already been defined the common data and metadata model for HFR surface currents (Corgnati et al., 2018, 2019b; Mantovani et al., 2020), the tools for near real-time and historical data processing (as defined in Corgnati et al., 2020 and Corgnati et al., 2019c, respectively), the guidelines (Reyes et al., 2019) and the training activities. Despite that, only 23% of the near real-time and 15% of the

historical data from the Mediterranean HFR sites are integrated to the European HFR node (Lorente et al., submitted to this Special Issue), established in 2018 as a focal point for data management and distribution. In order to reduce the bottlenecks that hinder the HFR data harmonization and their provision as open data (i.e. without requiring authentication or authorization to access them), a data sharing agreement between the parties providing and receiving data should be defined and put in place. It might also partly resolve the difficulties for sharing the HFR data held by companies or compromised to the private sector,

who partially or fully funded the installations. The collaboration with international initiatives will ensure convergence and interoperability. The HFR data harmonization and sharing will ultimately contribute to provide the research community with continuous and more valuable coastal data and to underpin the development of the HFR applications.

**d)  Enhancing data discoverability, access and usability:** a clear stakeholder engagement strategy will allow us to identify, categorize and analyze their needs, thus reinforcing the links and the loyalty with current users and enabling new communities

and sectors to discover and use the HFR data. The EuroGOOS HFR Task Team has already taken the first steps towards the definition of this strategy (Rubio et al., 2021), building a database of current and potential stakeholders, with the input of some Mediterranean institutions. However, a stronger involvement is needed to avoid imbalances between countries and greater efforts are required to move it forward by means of programs that promote networking and coordination. Tight interactions with stakeholders on a fit-for-purpose basis and the enhancement of the societal impact of the HFR data are major elements of

the strategy, especially, towards ensuring long-term sustainability. Boosting the regional involvement in this strategy will result in the spreading of end-user applications at the regional/local level, stimulating also the development of new ones in response to the users' feedback.





**e)** **Further development of emerging HFR applications.** The extension of the HFR surface current to fit for multiple purposes, aiming to address from single to multiple environmental threats, scientific questions and societal needs, requires a multidisciplinary integrative approach and coordinated monitoring of different essential observing variables. In this context, it should be noted that HFRs applications are diverse being able to focus on multiple threat- and different compartment, widening their use at the coastal areas: i) to monitor eutrophication in high-productive coastal waters, combining HFR surface currents with thermistor chains, oxygen and turbidity sensors at various depth increments, and to address physical-biological interactions in coastal basins (Cianelli et al., 2017; Hernández-Carrasco et al., 2018a); ii) to monitor the transport of floating marine litter and other contaminants using surface current fields from HFRs and models (Declerck et al., 2019); iii) for ship-tracking (Dzvonkovskaya et al., 2007; Laws et al., 2016); iv) for early tsunami/meteo-tsunami detection (Lipa et al. 2006; Monserrat et al., 2006; Guèrin et al., 2008 ; Lipa et al. 2011 & 2012; Gurgel et al. 2011, Dzvonkovskaya et al. 2012); v) for freshwater monitoring (Meadows et al., 2013); vi) for extracting new information from the HFR signals aiming to advance the understanding of key processes at the coastal areas, such as  stratification (Shrira and Forget, 2015), air-sea interaction (Berta et al., 2018) and mixing in the upper ocean, near-surface current shear, etc; viii) for promoting the HFR use for supporting marine renewable energy resource assessment (i.e. winds, currents, waves) in the coastal zone (Wyatt, 2012; Basáñez and Pérez-Nuñunzuri, 2021; Mundaca-Moraga et al., 2021), etc. Additionally, since the intensity of the multiple stressors (e.g. climate change effects, habitat loss and degradation, eutrophication, introduction of alien species, fishing practices, etc) is increasing throughout most of the Mediterranean basin, temporal analyses are progressively needed to inform effective current and future marine policies and management actions, also contributing to underpin longer-term scientific objectives.

**f)** **Extension of the HFR time series** that would enable the widespread implementation of novel data science methodologies**:** by guaranteeing the long-term sustainability, it is expected that the HFR measurements will be expanded in the next decade, increasing the availability of multi-year surface current datasets, for the benefit of the HFR derived STPs that uses self-learning algorithms, allowing also to test the stability of SOM-solutions in time. The further development of short-term predictive systems based upon HFR surface current fields and their adaptation to the Mediterranean HFR network by incorporating non-tidal component of current will enhance the STPs' integration into operational maritime safety applications, where it has been demonstrated their capacity to reduce the searching area (Roarty et al., 2010).

**g)** **Fostering the HFR data integration** with other *in-situ* and satellite data: Exploiting the nature of their measurements, HFRs are currently being used for filling the gaps of other sparse or lower spatio-temporal resolution observations in coastal areas, as well as for improving and assessing satellite observations. In this context, it should be considered the opportunity that will offer the launch of the wide-swath Surface Water & Ocean Topography (SWOT) altimeter in 2022, which should be complemented with other remote and *in-situ* sensors to fully resolve the typical Mediterranean mesoscale structures of 10-100 km (Gómez-Navarro et al., 2018). Additional complementarities might be fostered with the monitoring of surface current worldwide using the information from the Automatic Identification System -AIS- data streams (Benaïchoucheet al., 2021), where HFR measurements can be used as a consistent ground-truth dataset for validation purposes and for increasing the spatial resolution of the AIS-reconstructed fields at coastal areas. The integration of HFR measurements with other multi-platform





observations (e.g. ADCP as in Manso-Narvarte et al., 2020, glider data) have already been implemented and tested, particularly under the umbrella of the Jerico-Next project (see Griffa et al., 2019), for a better understanding of the three-dimensional coastal circulation, allowing the broadening of the HFR applications (i.e. below the surface). Data fusion and integration will contribute to increasing the societal and scientific value of all observations, not only HFR ones.

h) **Boosting the HFR data assimilation** for model improvement**:** As shown in the Sect. 2.1.2 of the present work, HFR surface current data assimilation have demonstrated to improve the model performance in many studies (Paduan and Shulman, 2004; Barth et al., 2008; Hernández-Lasheras et al., 2021). Furthermore, the HFR standard data distribution in near real-time throughout the main European marine data portals, facilitating the data access and ensuring the timeliness, make them an ideal dataset for efficient data assimilation in operational modelling (Capet et al., 2020). However, WMOP is the only one regional operational model from the Mediterranean, which is systematically assimilating HFR data (Hernández-Lasheras et al., 2021). As highlighted by Capet et al., (2020), the lack of expertise, training and capacity building is limiting the uptake of assimilation practices.

i) **Expansion of the pool of expertise** by including not only HFR technology, but data management and applications, satellite remote sensing, ocean modelling, data assimilation, training aspects, and exchanging and sharing knowledge, tools, data, know-how between diverse research groups at the European, regional and global levels.

j) **Training of the new generations of HFR technicians and scientists** are needed to ensure the knowledge exchange, the sufficient expertise to allow for a significant expansion of the coverage and the sustainability of the operations and the HFR applications. This is currently being done in the context of periodic workshops and summer schools (such as the recent ISSOR and SICOMAR+ summer schools, etc) in the academia context. Moreover, the development of best practices for demonstrations are not only key to reach satisfactory quality standards but also for fostering the learning process. However, additional technical training courses provided by the manufacturers at the HFR operator level are recommended. Furthermore, the participation of Mediterranean institutions and companies in the creation of an international, intersectoral and interdisciplinary qualified supporting training network will contribute to boost knowledge and know-how exchanges, involving all actors (e.g. academia, operators, manufacturers, private sector, etc).

k) **Strengthening partnerships**: (i) at the regional level, by enhancing inter-institutional collaboration, to exchange expertise, to build consortia and for sharing job opportunities in this field. In this sense, the Mediterranean HFR network is being benefited from the activities carried out in the context of EuroGOOS HFR Task Team, whose main mid-term milestones are summarized in Rubio et al., (2021), and from the ongoing regional joint projects and the existing national HFR coordination structures, as listed in Lorente et al. (submitted to this Special Issue). It is also highly recommended the participation in annual fora led by stakeholders, like the 1st MED-FORUM that brought together the Heads of maritime services or Coast Guards of almost all Mediterranean States, aiming to develop a regional policy in the field of maritime safety and to improve the efficiency of SAR services in the Mediterranean area (Trevisanut et al., 2010); (ii) at the European level, by being aligned with ongoing initiatives and projects, contributing to the EuroGOOS HFR Task Team and the main marine data portals (e.g. Copernicus Marine Service, EMODnet, SeaDataNet), that will avoid duplication of effort; (iii) at the Global level, by





collaborating with the international institutions that are world leaders in HFR and ocean observation as well as with the Global HFR network, to ensure the consistency in the data standards and best practices; (iv) with the private sector, by transferring the knowledge and the applications from the academia to the operational oceanography industry turn them into commercial services, improving the links between research and new technologies. The development of the existing and potential
applications will also assist the HFR manufacturers in marketing their technologies.

**l)   Seeking for funding** is a compulsory task in order to support, together with the stakeholder's engagement and the training of the new generations, the long-term sustainability of the HFR network, their data and applications at national, regional and European levels. To this end, EuroGOOS HFR Task Team has taken early steps to prepare a competence matrix that will facilitate the building of effective, interdisciplinary, intersectoral and well-balanced consortia, grounded in shared research
interests and goals, aiming to prepare competitive bids and applications for funding, taking advantage of the expertise of the team in diverse grant calls (Rubio et al., 2021).

**m) Regional contribution to long-term major effort** towards building a sustained and fit-for-purpose European Ocean Observing System capable to support the UN Decade of Ocean Science for Sustainable Development and the European Green Deal should be two-fold: i) On the one hand, the Mediterranean HFR network outcomes should be scientifically grounded to
further ensure the extension of the science-based added-value products into societal relevant downstream services (Tintoré et al., 2019) and, ii) on the other hand, the Mediterranean HFR community's long-standing cooperation must be further strengthened towards a co-designed and sustained regional network, contributing to and, simultaneously, benefited by the European HFR Task Team endorsement, roadmap and main achievements (as recommended by Lorente et al., submitted to this Special Issue).

**5 Summary and conclusions**

The socio-economically vital and heavily human and environmentally stressed coastal areas of the Mediterranean Sea are ones of the most exposed regions in the world due to the impact of climate change, being also highly vulnerable target regions for maritime safety, oil and marine litter pollution, fish stocks overexploitation and met-ocean hazards. The high spatio-temporal variability of the coastal dynamics requires the monitoring of these (sub)mesoscale processes at the right scale. HFRs are
nowadays the only technology for remotely continuously monitoring surface currents (increasingly waves and winds) at unprecedented high spatio-temporal resolution in the coastal areas and with relatively low cost-effort, when compared with other traditional observing platforms. Their integration in ocean observing has boosted the progress in the research of small-scale features and their interaction with larger scales, also underpinning the further development of applications. In this work, we present a review of the existing advanced and emerging scientific and societal applications using HFR data, developed to
address the major challenges identified in the Mediterranean coastal waters, organized around three main topics: i) maritime safety; ii) extreme hazards and iii) environmental transport processes. In addition to previous studies carried out at global and at the European scale on this topic, this work also provides a list of strengths, weaknesses, opportunities and threats of the



existing HFR applications in the Mediterranean Sea. Finally, we discuss the prospects for the future of the HFR applications and we provide a set of recommendations aiming to maximize the contribution in extending the science-based HFR products into societal relevant downstream services to support the blue growth in the Mediterranean coastal areas, helping to meet the UN's Decade of Ocean Science for Sustainable Development and EU's Green Deal goals.

Considering the capabilities of the existing HFR applications, it can be concluded that HFR technology and its increasingly integration in the operational ocean monitoring systems is playing a key role in the production and use of services for continuous advances in the scientific understanding of coastal ocean dynamics, technological development and in support of the sustainable blue growth in the Mediterranean Sea. However, still major efforts should be done for unlocking the HFR interoperable data access and potential as well as for the further development of HFR scientific and societal applications at the regional level, thus, delivering greater uptake, use and value. Fortunately, the opportunities provided in the framework of the UN Decade of Ocean Science for Sustainable Development and the European Green Deal can help to ensure the full exploitation of this HFR potential. In this sense, the collaboration at regional level is crucial to address region-specific processes towards a fit-for-purpose and coordinated design of the monitoring actions, to identify the environmental threats and their impacts in the environment and in the blue sectors, to easily identify the existing stakeholders, also fostering the interaction with them and to engage the potential users. This will help improve the long-term sustainability together with the training activities to the next generations and the seeking for funding. Certainly, this regionalization should always be aligned with the European strategies to ensure the integration, benefiting from the European HFR roadmap and the availability of near real-time and long-term HFR interoperable data that will boost the research and to underpin the further development of the HFR scientific and societal applications at regional level.

This manuscript constitutes the second part of a double contribution, which complements the comprehensive overview on the current status, achievements, challenges and roadmap of the HFR network in the Mediterranean, provided in the first part (Lorente et al., submitted to this Special Issue).

**Author Contributions**

ER and PL conceived the idea of this manuscript and fostered the collaboration as MONGOOS-HFR co-chairs, being in charge of overall direction and planning. All authors contributed to the writing of the different sections of the manuscript, as follows: ER took the lead in writing the abstract, and Sect. 1, 3, 4 and 5, contributing also to Sect. 2.1. Other authors contributed in the edition of Sect. 1 (CAG, DD, MB, SS) and Sect. 4 (YT, MB, RG). MB also contributed to the summary of HFR capabilities in Sect. 3 and Sect. 5.

CRS, VC, ML, HM, AD, AG, AM, AR and ER (designed the Fig. 1) collected the information from the Maritime Safety and Rescue Agencies for their respective countries.

CRS and ML (designed the Fig. 3) contributed to the introduction of the Sect. 2.1 and, together with VC, AR (designed the Fig. 2), JT and ER they shaped the Sect. 2.1.1.



PL (designed the Fig. 4), BM, JHL (designed the Fig. 5), EA, and ER wrote the Sect. 2.1.2.

IHC, AO, HM (designed Fig. 6), IV, VD took the lead in writing the Sect. 2.1.3.

MB (designed Fig. 7), AG, LC, CM contributed to the data analysis and text drafting in Sect. 2.2.1, in particular for the case study in the Ligurian Sea. In this section, PL (designed Fig. 8) and ER addressed two cases studies in the Delta Ebro and one in the Strait of Gibraltar.

CAG took the lead in writing the Sect. 2.2.2, where BM, ML and MJF also contributed.

AM, ACE, IHC (designed Fig.10 and Fig.11), AO contributed to the writing of the Sect. 2.3.1.

MBen - Consorzio LaMMA - (designed the Fig. 12 and Fig. 13), CB, ST, BD, MU, MF, PF, EZ, HM, IV wrote the Sect. 2.3.2.

MM, DC (designed the Fig. 14), RS (designed the Fig. 15), FC (designed the Fig. 16), CG,  MB, AG, IHC, AO contributed to the Sect. 2.3.3. Particularly, RS and MM focused in the Gulf of Manfredonia;  DC in the Gulf of Naples and FC in the Malta-

Sicily case studies.

AM, IV, CAG, JT and the MONGOOS co-chairs, VC and AO provided critical feedback and helped shape the final version of the manuscript during the internal review process.

All authors have read and agreed to the submission of the manuscript for publication.

**Acknowledgements**

This work has been possible thanks to MONGOOS (Mediterranean Operational Network for the Global Ocean Observing System) collaborative network, aimed toward long-term synergies between multi-disciplinary working groups in the Mediterranean Sea in order to launch strategic initiatives and pursue funding for innovative research projects.

This work was supported by the EuroSea (EU Horizon 2020 research and innovation programme, grant agreement ID 862626). We also thank the partial support of the project CMEMS-INSTAC phase II, which provides the context of the activities for

HFR data harmonization, standardization and distribution, and to the IBISAR CMEMS User-Uptake project (67-UU-DO-CMEMS-DEM4_LOT7). Eva Aguiar and Baptiste Mourre are also very grateful for the MEDCLIC project (LCF/PR/PR14/11090002) supported by La Caixa Foundation, contributing to the development of the WMOP model. The methodology for the extreme events monitoring in the Ligurian Sea (Sect. 2.2.1) has been developed under the framework of the CARTHE III project (Prime Award n. SA 18-14, subcontract agreement SPC-000649) and the CALYPSO Departmental

Research Initiative (Grant number N00014-18-1-2782). The HFR network in the Ligurian Sea has been supported by the IMPACT project (EU funded, PC Interreg VA IFM 2014-2020, Prot. ISMAR n. 0002269). Investigation of Mediterranean Sea dynamics and applications of HFR observations are also supported by the JERICO-S3 project (EU Horizon 2020 research and innovation programme, grant agreement no. 871153). The assimilation of HFR data in WMOP (Sect. 2.1.2) and the biological connectivity application in the Gulf of Manfredonia (Sect. 2.3.3) have been developed under the JERICO-NEXT project (EU

Horizon 2020, grant agreement no. 654410). In the latter case,  the COCONET project (EU FP7, grant agreement no. 287844)





and the Italian national projects SSD-PESCA and RITMARE should also be acknowledged. Recent upgrades in the radar installations of the MIO in Toulon were supported by the SICOMAR-PLUS EU Interreg Marittimo project.

Pier Falco and Enrico Zambianchi acknowledge partial funding from the Italian Ministry for University and Research through the "Exploring the fate of Mediterranean microplastics: from distribution pathways to biological effects" (EMME) PRIN
Project.

We are furthermore grateful for the valuable information about the Maritime Rescue activity for 2019 provided by the Spanish Maritime Search and Rescue Agency, the Italian Coast Guard, the French CROSSMED, the Ministry of Maritime Affairs, Transport and Infrastructure from Croatia, and the Search and Rescue Coordination Centers from the SRR (Search and Rescue Regions) of Slovenia and Malta. We also acknowledge the Spanish Meteorological Agency AEMET for providing
HARMONIE atmospheric fields. The authors would like to thank Stephan Grilli, from the University of Rhode Island (USA) for the design of Fig. 9 (from Sect. 2.2.2) and to Leonardo Bagaglini for processing the data shown in Fig.12 (from Sect. 2.3.2). Finally, authors would like to express their gratitude to the internal reviewers, Anne Molcard, Ivica Vilibić, Charles-Antoine Guèrin and Joaquín Tintoré and to the MONGOOS co-chairs, Vanessa Cardin and Alejandro Orfila, for their careful and meticulous reading of the manuscript. Their detailed and comprehensive reviews have been very helpful to improve the
structure, the reading and to finalize the manuscript.

**Code availability**

Corgnati, L. (2019, December 10). LorenzoCorgnati/HFR_Node__Historical_Data_Processing:

EU_HFR_NODE_Historical_Data_Processing (Version v2.1.1.6). Zenodo. http://doi.org/10.5281/zenodo.3569519

Corgnati, L. (2020, May 26). LorenzoCorgnati/HFR_Node_tools: EU_HFR_NODE_Tools (Version v2.1.2). Zenodo.
http://doi.org/10.5281/zenodo.3855461

**Data availability**

IMO data, mentioned in Sect. 1, is available in the Flow Monitoring Displacement Tracking Matrix website:
https://migration.iom.int/europe?type=arrivals

Eurostat statistics for Maritime transport of goods, mentioned in the Introduction, are available in:
https://ec.europa.eu/eurostat/statistics-explained/index.php?title=Maritime_transport_of_goods_-
_quarterly_data&oldid=485429#EU_ports:_activity

The "Sub-regional Mediterranean Sea Indicators" tool, mentioned in the Sect. 1, is available in the website:
https://apps.socib.es/subregmed-indicators/

The SAR incidences in France, included in Sect. 2.1.1, were obtained from the website of the French Ministry of the Sea:
https://www.mer.gouv.fr/surveillance-et-sauvetage-en-mer



SOCIB HFR-Ibiza data, used in Sect. 2.1.1 (Fig. 2), Sect. 2.1.2 (Fig.5), Sect. 2.3.1. (Fig. 10 and Fig. 11), are available in the https://doi.org/10.25704/17gs-2b59.

SOCIB drifter's data, used in Sect. 2.1.1., are available in  https://doi.org/10.25704/mhbg-q265 for 2014 (they have also been used in Sect. 2.1.2, Fig. 5), https://doi.org/10.25704/bb7m-zv61 for 2016 and https://doi.org/10.25704/84ze-sf42 for 2018.

SOCIB's WMOP simulations are available upon request to info@socib.es

Standardized HFR data is available in the Thredds Server from the European HFR Node for some of the HFR systems mentioned in this research (http://150.145.136.27:8080/thredds/HF_RADAR/HFradar_CMEMS_INSTAC_catalog.html)

MIO's HFR-Toulon data is available http://hfradar.univ-tln.fr/HFRADAR/squel.php?content=accueil and real-time total currents      (hourly      data)      in      standard      format      are      available      for      2020      and      2021      in

https://erddap.osupytheas.fr/erddap/files/cmems_nc_cf0e_c84a_8ead/

HFR-LaMMA data, used in Sect. 2.3.2, can be viewed in http://www.lamma.rete.toscana.it/meteo/osservazioni-e-dati/radar-hf.

HFR      data      for      the      Northern      Adriatic,      used      in      Sect.      2.1.3      (Fig.      6),      can      be      viewed      in http://jadran.izor.hr/hazadr/geoserver_en2.html

HFR-NAdr data, used in Sect. 2.1.1. (Fig. 3), can be viewed in http://www.nib.si/mbp/en/oceanographic-data-and-measurements/other-oceanographic-data/hf-radar-2

**Competing interests**

Authors MF, RG and PL are currently employed at Qualitas Instruments Lda, at HELZEL Messtechnik GmBH and at NOLOGIN Consulting SL, respectively. However, authors have not advertised commercial products and the research has not

been sponsored by any one of the companies.

AO and VC are guest members of the editorial board of the Special Issue from the Journal. The peer-review process was guided and overseen by another member of the editorial board.

The remaining authors declare that there are no relevant financial or non-financial competing interests to report.

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
