# Peer review of "Coastal HF radars in the Mediterranean: Applications in support of science priorities and societal needs"

_Ocean Science, 2021_

## Referee Comment (RC1)

**Review of 'Coastal HF radars in the Mediterranean: Applications in support of science priorities and societal needs', by E. Reyes et al.**

**General Comments**

The paper provides an excellent review of all the huge scientific progress achieved by the Mediterranean oceanographic community using data from HF Radars and clearly shows its increasing potential to develop oceanographic products and services with a high societal impact in the coastal areas.

The analysis of the capabilities of the HFR performed by the authors, including the SWOT analysis, and the derived set of recommendations is a necessary exercise to help the Mediterranean HFR community to further improve and promote the use of HFR data in scientific and societal applications.

The authors provide significant examples on how HFR has helped to advance on scientific questions and how they can improve numerical model performance and forecasts. However, showing some more concrete examples on operational applications could improve the manuscript. Could the authors provide, or highlight, more examples of 'successful stories' of applications of HFR products?

Also, can the authors better explain the concept of 'regionalization' (mentioned in Section 4 and then in the conclusion section)? does it mean that the derived recommendations are particular to the Mediterranean region or general for the global HF community?

Could the authors explain in a clearer way how this manuscript, especially the recommendations and conclusion section, is linked (and complements) the first part of the double contribution?

**Specific Comments**

All along the text, the 'high spatio-temporal scales' and 'wide coastal areas' provided by the HFR are mentioned, but readers not familiarized with HF technology may not know which scales and ranges the authors are referring to.

Section 2.1.1:

- The title refers to SAR incidents, but oil spill incidents are also mentioned in the text.
- A list of incidents for 5 countries and the HFR installations for those countries (except for Slovenia) is given, but there is no clear information on how the HFR data helped to support any of those incidents. Maybe a concrete example (successful story) of use of HFR in one incident could help.  In the example given for the Northern Adriatic it is not clear if the HFR data has been used in the real incident or if it is an academic study.
- Figure 1 showing the location of radars and the areas where SAR incidents took place could be zoomed for the different countries to better see the location of the radars and their proximity to the incident's location. That would also help to know if the area of the incidents were covered by the radars.
- Figure 2 and text in lines 236-239 are not very clear, what is the conclusion of this study? Forecast data is only reliable in areas with red colors in Figure 2?

Section 2.1.2

Section 2.1.2 (model assessment and improvement – including DA) and Section 2.1.3 (STP) are included under 'maritime safety' topic, but they are more general and could also be under topics 2.2 (extreme hazard) and 2.3 (environmental transport.).

Section 2.1.3

- It is not clear from the text if the use case described using SOM is being actually implemented operationally or if it is an academic study.
- Paragraph after line 425 reads more as a discussion on the use of STP algorithms rather than a concrete existing application of HFR data in Short Term Prediction algorithms.

Section 2.2.1

- I would recommend, for ease of reading to make a bullet point of the concrete examples of monitoring of extreme events described by the authors.
- I would recommend shortening the section by focusing more on the added value of the HFR data in detecting and monitoring extreme events response on oceanographic structures, it shouldn't be necessary to reproduce all the results which are already well described in the relevant literature.
- A figure (or a reference to an existing figure) illustrating the emplacement of the different HFR systems mentioned (e.g. HFR-TirLig, etc) would facilitate the reading.

Section 2.3.1

- An depth explanation of the lagrangian approach is included here, but the lagrangian approach has been already used before in the manuscript (f.ex in section 2.1 or 2.1.1). I would recommend avoiding the details of how lagrangian trajectories are computed (i.e. the methodology) and focus more on the added value of HFR in the pollution and floatable tracking and five concrete examples of applications.

Section 3

- The writing of this chapter needs some revision.
- Some important points in figure 17 showing the very detailed SWOT analysis are not reflected in the text in the Section, for example: the important Weakness/Threats detected on 'Difficulties for addressing user engagement challenges' or 'Disconnection between academia with the private sector and policy-makers' or 'limited user uptake' or 'Fail in addressing the user needs' deserve more attention in the discussion.
- Fig17: some bullet points require better redaction, e.g.:
  - 'Risk that the effort will diminish or disappear', which effort?
  - 'Lower HFR operational as usual', what 'as usual' means

Section 4

A set of 13 prospects/recommendations are given by the authors, but it's not clear what are the recommendations and what are the prospects: as it reads now the recommendations are spread all along the 13 points (e.g. 'collaboration with international initiatives' in c) and in m)) or repeated in 2 or

more points (e.g. 'creation of partnerships in points b) and k)). Could the authors perhaps group them by categories and specify what are the concrete recommendations to the HFR Mediterranean community?

The recommendations should also be better linked with the SWOT analysis, especially with the identified weakness and threats. For instance, recommendations on how to improve the long-term sustainability of the HFR network could be better stated (not only the short-term funding in grant calls mentioned in point l).

Section 5

The concept of 'regionalization' mentioned in L1098 is not very clear. What the authors mean here, the same as 'regionalization of the recommendations' (L929) or 'collaboration at regional level'?

**Technical comments**

Some parts of the manuscript need minor some revision. See below some examples.

L115: remove 'as described by several authors'

L117: should read 'installation' instead of 'implementation'

L122: what 'advanced' means here?

L135-137: needs a better formulation of the overall description of Section 4

L137: 'overcome' should read 'achieve'

L138: References to the UN Decade and the European Green deal may help readers not familiar with those initiatives.

L149: is it possible to add a reference to the oil spill incident in 2021 mentioned by the authors?

L150: 'it has once again been demonstrated': explain why 'once again', the reader not necessarily know all the background behind this sentence.

L164: typo in 'fate of the trajectory'

L211: Fig.1: better explain what 'HFR bounding box' is. There is a red square in the eastern Med which doesn't correspond with any country mentioned in the text.

L249: typo in 'Sirocco'

L280: not clear from the text what are the 'challenges associated with resolution'

L337: 'for a wide range of coastal stakeholder including Ports and environmental agencies'

L363: 'assimilated radial velocity observations'

L384/385: what could be the reasons for been the only operational example using HFR DA, is this discussed later in the text?

L440-443: the retrieval of wave and wind map, of high interest for sectors like offshore energy it is not under 'extreme hazard monitoring', please clarify why this sentence here.

L448: 'This work' refers to Berta et al. 2020? Not clear in the text

L468-469: 'adding to the analyzing' needs better redaction

L527: Typo in 'Sicily'

L561-562: it is not clear if the radar in the Gulf of Lyon is operating at 4.5MHz in the Mediterranean, while the example gives before (Sagres, Portugal) operates at 13Mhz. Not clear what is 'Stradivarious radar'? could you please clarify.

L565: 'As recently suggested by Domes et al., 2020...'

L583: can the authors provide some example or reference of 'sensitive ecosystems in the shallow water areas'

L587: 'strong tourist' must read '...strong tourism pressure'

L599: 'fast growth...' must read 'growing importance of HFR as a key...'

L759: 'reliability with respect to the actual sea dynamics', please further explain.

L776: should read 'is monitored weekly since 1984'

L785: must read 'Chlorophyll-a'

L851: should read 'decision-making in fisheries management'?

L859: 'inventory' to be replaced by 'compilation' or 'review'

L859-864: paragraph needs review

L879-891: paragraph needs review

L922-923: sentence is not very clear. What is mean here by 'European HFR network'? is this a stablished Network? In other part of the text 'HFR network' was referred to particular HFR installations in different regions.

L927: the end of the sentence is not clear: 'to ensure the integration.', 'integration' of what?

L928: should read 'HFR data and applications use'

L933: what the authors refer here by 'MONGOOS-HFR' Network? Is the same as the Mediterranean HFR Network?, or the Mediterranean HFR community is already organized around MONGOOS?

L941: what the authors mean here by 'regionalization of the observatories'?

L985: 'essential observing variables' meaning 'essential ocean variables'?

L945: 'Blue sectors' should read 'Blue economy sectors'?

L997: 'etc' should be removed or better placed.

L999: What does 'temporal analysis' mean here?

L1025: should read 'is the only regional model'

1050: should read 'efforts'

L1068: Do the authors refer here to 'European' or 'EuroGOOS' HFR Task Team?

L1077: should read '…ocean observing systems…'

L1079: what does 'advanced' means here? mature applications?

L1087-1090: This important sentence could be carefully reviewed: what are the 'services' referred here? what 'technological development' means here?

---

## Author Comment (AC1)

**RESPONSE TO REVIEWER #1**

- Title: Coastal HF radars in the Mediterranean: Applications in support of science priorities and  societal needs

- Author(s): Emma Reyes et al.

- MS No.: os-2021-115

- MS type: Review article

- Special Issue: Advances in interdisciplinary studies at multiple scales in the Mediterranean Sea

Please, note that all the points raised by the reviewer are in **bold font** below, while the responses to the review are in normal font and the changes made to the revised manuscript are "quoted".

**General Comments**

**The paper provides an excellent review of all the huge scientific progress achieved by the Mediterranean oceanographic community using data from HF Radars and clearly shows its increasing potential to develop oceanographic products and services with a high societal impact in the coastal areas.**

**The analysis of the capabilities of the HFR performed by the authors, including the SWOT analysis, and the derived set of recommendations is a necessary exercise to help the Mediterranean HFR community to further improve and promote the use of HFR data in scientific and societal applications.**

> Thank you for your encouraging comments and the comprehensive comments and suggestions of our manuscript that have helped to significantly improve the quality of the final version. We have incorporated all your suggestions in the revised manuscript. Please, see our responses below.

**The authors provide significant examples on how HFR has helped to advance on scientific questions and how they can improve numerical model performance and forecasts. However, showing some more concrete examples on operational applications could improve the manuscript. Could the authors provide, or highlight, more examples of 'successful stories of application of HFR products?**

> Some examples of operational applications are already provided either in the present manuscript or in the companion paper from Lorente et al., (2021) as follows:

As mentioned in section 2.1.2, the downstream service IBISAR -freely available in www.ibisar.es- is presented as a model skill assessment operational downstream service aiming to provide user-friendly metrics (Révelard et al., 2021). In the companion paper from Lorente et al., (2021), this service is included as a successful example of the long-lasting engagement and collaboration between HFR operators and end users, such as the Spanish Maritime Safety and Rescue Agency.

In parallel with the IBISAR service developments, 3 Spanish HFR networks operating in the Mediterranean coastal areas were ingested by the Environmental Data Server (EDS) managed by SASEMAR.

Also in section 2.1.2, the WMOP model (Juza et al., 2016, Mourre et al., 2018) has been presented as the first and only system in the Mediterranean Sea including an assimilation scheme of HFR data in its operational chain (https://socib.es/?seccion=modelling&facility=forecast). This is also mentioned in L601 and in section 4, issue f)

The companion paper (Lorente et al., 2021) details a recent operational application based on the use of hybrid approach by applying both Beam Forming and Direction Finding techniques to phased-array HFR systems (Dumas et al., 2020)

Also in the companion paper, in section 2.4, the EU HFR Node (Corgnati et al., 2021) is presented as a success story, as it constitutes the European focal point for HFR data management and dissemination fully operational since November 2018.

In this context, the integration of standardized HFR near real-time and reprocessed surface current data in the European marine portals (e.g. Copernicus marine Service, EMODnet, SeaDataNet) is also included in section 2.4 as a success story.

Despite the great advances made and the boost of operational applications and services based on HFR products in the recent years, still major efforts should be done for unlocking the HFR interoperable data access and potential as well as for the further development of HFR scientific and societal applications at the regional level, thus, delivering greater uptake, use and value (as mentioned in the section 5 of this manuscript). Aiming to leverage the HFR data to its fullest potential, a set of recommendations are provided in the section 4.

The publication from Lorente et al., (2021) is a companion paper to this study and cross references have been given in both papers to avoid needless repetition and to complement each other.

**Also, can the authors better explain the concept of 'regionalization' (mentioned in Section 4 and then in the conclusion section)? does it mean that the derived recommendations are particular to the Mediterranean region or general for the global HF community?**

Thank you for pointing out the need for clarification about the concept of 'regionalization'. It means that, although all derived recommendations are aligned and shared with the Global HF radar community and with the roadmap established by the European HF radar network, they have a stronger regional focus to consider the singularities of the Mediterranean Sea.

In particular, section 4 of the manuscript has been upgraded to better clarify:

"Although the future prospects for the HFR data and its applications at the regional level are shared far beyond the geographical borders of the Mediterranean Sea, recommendations have a stronger regional focus: i) to coordinate cross-national efforts (i.e. implementation of technical, financial and management approaches at national and regional level, establishment of cross-border agreements for obtaining dedicated frequency allocation for HFR technology); ii) to account for regional specificities (e.g. north-south unbalance in the monitoring capabilities, prominent use of medium and short range HFR frequencies, existence of ship noise and radio frequency interferences due to high marine traffic density, high restrictions for obtaining the required HFR installation licenses in coastal tourism areas, etc.) and to respond to regional needs (in terms of scientific key priorities, societal needs and existing environmental threats) and iii) to map the existing and potential HFR data regional end-users, also facilitating the interaction with them."

The revised paragraph in section 5 will read as follows:

"Certainly, this regional approach and focus to strengthen collaborations should always be aligned with the global and European strategies to ensure the integration, benefiting from the European HFR roadmap and the availability of near real-time and long-term HFR interoperable data that will boost the research and to underpin the further development of the HFR scientific and societal applications in the Mediterranean coastal areas."

**Could the authors explain in a clearer way how this manuscript, especially the recommendations and conclusion section, is linked (and complements) the first part of the double contribution?**

Since the main goal of the proposed joint work was fairly ambitious and given the relevant number of participants (i.e. 42 co-authors from 7 Mediterranean countries from 22 different institutions), it was agreed with the editors to split the joint contribution.

The first part of the double contribution (hereinafter Part-1) aims to provide a

panoramic overview of the roadmap to transform individual HFR systems into a fully integrated, mature network operated permanently in the Mediterranean Sea.

The second part (hereinafter Part-2) complements Part-1 by focusing on existing advanced and emerging scientific and societal applications using HFR data, developed to address the major challenges identified in the Mediterranean coastal waters (previously defined in Part-1).

In summary, Part-2 benefited from the introductory context offered in Part-1, which provides a comprehensive overview of the current status, the descriptions of the joint efforts made in promoting data standardization, in the implementation of best practices, in the boosting of the data integration, thus ensuring data availability. Part-2 shows how these joint efforts have enhanced the HFR data integration into services and the development of applications contributing to improve data discovery and unlock the interoperable data access and its potential.

Recommendations in Part-1 tackles the main technical, socioeconomic and scientific challenges for the implementation of an integrated HFR regional network, while in Part-2 aim to ensure that the potential of the HFR data is fully exploited in the development of operational monitoring systems at the regional level.

In addition to the appropriate cross-reference between both parts to inform the audience about the existence of the introductory part, a further clarification has been included in the conclusion as follows:

"This manuscript constitutes the second part of a double contribution that supports each other in an integrative way and should be interpreted as a single entity. The first part from Lorente et al., (2021b) provides a comprehensive overview on the current status, achievements, challenges, coordinated efforts and the roadmap to transform individual HFR systems into a fully integrated HFR network in the Mediterranean. Additionally, this work shows how these joint efforts have benefited and boosted the HFR data integration into services and the development of a broad range of multidisciplinary science-based and fit-for-purpose applications, contributing to leverage the HFR data to its fullest potential."

**Specific Comments**

**All along the text, the 'high spatio-temporal scales' and 'wide coastal areas' provided by the HFR are mentioned, but readers not familiarised with HF technology may not know which scales and ranges the authors are referring to.**

Agreed. Further details have been provided in the Introduction as follows:

"The increased capability to address the above-mentioned regional challenges at the required spatio-temporal scales has directly benefited, inter alia, from the key features of the High Frequency Radar (HFR hereinafter) technology, i.e. unprecedented high spatio-temporal resolution (i.e. 0.2-6 km and 15-60 min) over wide coastal areas (up to 200 km offshore, depending on the operational frequency)"

**Section 2.1.1:**

- **The title refers to SAR incidents, but oil spill incidents are also mentioned in the text.**

  The reviewer is completely right. Although maritime safety agencies are also responsible for pollution prevention and response, maritime traffic, operators training, etc., we have here focused on the analysis of SAR operations. Accordingly, we have reworded the sentence referring to oil spills in section 2.1.1.:

  Reworded. "Winds, waves and surface currents observations and forecasts are needed to be seamlessly integrated into their SAR emergency response in order to predict the trajectory of a drifting target for determining the optimal search region."

  Removed. "No oil spills were reported as for 2019 by the Italian MISE (Ministero dello sviluppo economico) whereas spills from other sources were not officially reported."

  Removed. "(the remaining 12% being in response to issues related to marine pollution)"

- **A list of incidents for 5 countries and the HFR installations for those countries (except for Slovenia) is given, but there is no clear information on how the HFR data helped to support any of those incidents.**

The reviewer is right. The HFR sites from Slovenia (i.e. HFR-NAdr) are jointly operated through trans-national collaboration with Italy. This has been clarified in the specific bullet point corresponding to Slovenia, as follows:

"The HFR-NAdr, in the Northern Adriatic Sea and the Gulf of Trieste, is jointly operated through trans-national collaboration with Italy."

We would like to clarify that the objective of providing the list of incidents, also shown in Fig. 1, aims to highlight that most of the SAR incidents occurred in coastal areas, where the HFRs continuously monitor the sea state. In this context, HFRs contribute to fill the gap due to the lack of coastal observations and to address the limitation of other existing observing platforms (i.e. satellite, drifters, fixed platforms) in response to maritime emergencies, helping also to improve the performance of the high-resolution models through data assimilation.

- **Maybe a concrete example (successful story) of use of HFR in one incident could help.**

  In addition to the success stories provided in response to the Reviewer's General comments, particularly to those related to the IBISAR service and the ingestion of the HFR data in the EDS from SASEMAR, we have not documented any other.

- **In the example given for the Northern Adriatic it is not clear if the HFR data has been used in the real incident or if it is an academic study.**

  In the example given for the Northern Adriatic, the HFR-NAdr surface current observations were used to hindcast and verify the survivor's trajectory as mentioned in the text but not during the response stage of the emergency. A clarification about the academic contribution has been added:

  "A further academic study of the value of quality controlled HFR observations in SAR operations was the recent case of a person lost at sea in the Northern Adriatic…"

- **Figure 1 showing the location of radars and the areas where SAR incidents took place could be zoomed for the different countries to better see the location of the radars and their proximity to the incidents location. That would also help to know if the area of the incidents were covered by the radars.**

  Fig. 1 has been modified to improve the visualization of the different HFR systems mean coverage (instead of the geospatial bounding box) over wide coastal areas where SAR incidents have been reported by France, Italy, Slovenia and Spain for 2019.

[Figure]

Zoomed figures for the Strait of Gibraltar, for the North-Western Mediterranean sub-basin and for Italy are also provided below. As duly mentioned in the manuscript [L212-215], it must be considered here that "In the sphere of maritime safety, HFRs have the great advantage of providing high spatio-temporal resolution surface currents in wide coastal areas, very close to the coastline when HFR gap-filling methods are applied (listed in Sect. 2.1.3) and where most of the SAR incidents occur (as shown by the Fig. 1)"

[Figure]

[Figure]

Those maps could be added to Fig. 1 as additional panels, we leave it to the discretion of the anonymous reviewer.

- **Figure 2 and text in lines 236-239 are not very clear, what is the conclusion of this study? Forecast data is only reliable in areas with red colours in Figure 2?**

Red colors in Fig. 2 show the higher model performance, based on the temporally averaged novel Skill Score (SS*) of each model during 6-h periods. This has been specified in the figure caption as follows:

'SS* values, where the red colors represent the higher model performance, are only obtained in those grid…'

The conclusions of this study are that: i) the Skill Score defined by Liu and Weisberg (2011) is sensitive to the forecast time, i.e. the longer the forecast (i.e. 72 hours), the higher the SS value; ii) a short forecast time (e.g. 6 hours) should be applied in coastal areas, characterized by high variability ; iii) the use of the novel Skill Score (SS*) is recommended to assess the average model performance over an area of interest and along a specified period to avoid biased conclusions. The advantage of the SS* is that, despite its similar formulation to the SS given by Liu and Weisberg (2011), SS* does not impose the negative values to zero.

Strictly focusing on the analysis of the HFR data, the conclusions of this study is that the use of HFR derived trajectories improve the robustness of the skill score statistics and the spatio-temporal assessment of the model performance, when compared to the methodology applied using drifter trajectories.

The paragraph highlighted by the reviewer has been reworded as follows:

"Aiming to improve the applicability of this model assessment methodology for SAR operations in coastal areas, Révelard et al. (2021) also analyzed the SS sensitivity to different forecast horizons and showed that, in coastal regions: i) the SS is sensitive to the forecast time: the longer the forecast (i.e. 72 hours), the higher the SS value, due to the high variability of the surface currents; ii) a shorter forecast time (e.g. 6 hours), consistent with the duration of the search that maximizes survivors in SAR missions, is therefore more appropriate. In addition, they have shown that whereas the original definition of the Skill Score from Liu and Weisberg (2011) is correct for analyzing its spatiotemporal distribution, the use of a novel Skill Score (SS*) is recommended to assess the average model performance over an area of interest and along a specified period to avoid biased conclusions. The advantage of the SS* is that, despite its similar formulation to the SS defined by Liu and Weisberg (2011), SS* does not impose the negative values to zero, allowing to obtain a correct average, as in Fig. 2. However, they clarified that only SS* values > 0.5 should be interpreted as a good agreement between HFR surface current observations and model outputs."

**Section 2.1.2**

**Section 2.1.2 (model assessment and improvement –including DA) and Section 2.1.3 (STP) are included under 'maritime safety' topic, but they are more general and could also be under topics 2.2 (extreme hazard) and 2.3 (environmental transport.).**

We agree with the reviewer that the model assessment and improvement through HFR data assimilation and the Short Term Predictions can be considered as added value products to tackle cross-cutting issues from several research areas. It is also true that both products could be placed under topics 2.2 and 2.3. This question was previously and jointly discussed during the internal review of the manuscript. We finally decided to include all under the same section (i.e. "maritime safety") to highlight the HFR strengths for SAR applications through model assessment and improvement, backtracking and short term forecasting. We wanted to emphasize the unique ability of HFRs to provide fine-resolution maps of the surface currents over broad coastal areas, where most of the SAR incidents occur and where other observations are scarce or have limited abilities.

Other options previously discussed were: i) to change the order of the subsections (addressing first the model assessment, then STPs and finally the SAR) and ii) to address the model assessment and STPs in a specific section (i.e. added value products of cross-cutting research areas).

We would prefer to consider the consensual decision and keep all three subsections under "Maritime safety". A specific paragraph has been included in the introduction of the section 2.1, in order to emphasize the wider implementation of the models and the STPs in other cross-cutting areas.

"Concerning the last two sections, it is important to mention the wider implementation of the ocean models and the short-term predictions in other cross-cutting areas from the three addressed main topics. Nevertheless, we have included them in this section to highlight the HFR strengths for SAR applications through model assessment and improvement, backtracking and short term forecasting."

**Section 2.1.3**

- **It is not clear from the text if the use case described using SOM is being actually implemented operationally or if it is an academic study.**

    The innovative neural network-based ocean forecasting system in the context of the NEURAL project is implemented operationally. The surface current forecast is published online (http://jadran.izor.hr/neural/about.htm), being available to numerous potential users. We have pointed this out in L643.

- **Paragraph after line 425 reads more as a discussion on the use of STP algorithms rather than a concrete existing application of HFR data in Short Term Prediction algorithms.**

    Agreed. Since this discussion has already been addressed in section 4, item k), the last paragraph of this section 2.1.3 has been removed, while the item k) has been complemented.

**Section 2.2.1**

- **I would recommend, for ease of reading, to make a bullet point of the concrete examples of monitoring of extreme events described by the authors.**

Thanks for the recommendations. In the introduction of section 2.2 we have enumerated all examples of extreme events addressed in the respective subsections, in a roman numeral ordered list. We preferred to do this way to be consistent with the rest of the subsections, where the bullet points have not been used.

- **I would recommend shortening the section by focusing more on the added value of the HFR data in detecting and monitoring extreme events response on oceanographic structures, it shouldn't be necessary to reproduce all the results which are already well described in the relevant literature.**

Your recommendations are very valuable, thank you. We have shortened the section as much as possible. However, we think that the description of some of the results are needed to facilitate the interpretation of the figures available in the manuscript. For all those other extreme events without a corresponding figure, the reader is referred to the literature for further details.

- **A figure (or a reference to an existing figure) illustrating the emplacement of the different HFR systems mentioned (e.g. HFR-TirLig, etc) would facilitate the reading.**

Agreed. The reference to Fig. 1 of the companion paper has been added in the figure caption of the Fig. 1 of the present manuscript. This figure includes the main geographical features and the name of the different HFR systems.

**Section 2.3.1**

- **An depth explanation of the lagrangian approach is included here, but the lagrangian approach has been already used before in the manuscript (f.ex in section 2.1 or 2.1.1). I would recommend avoiding the details of how lagrangian trajectories are computed (i.e. the methodology) and focus more on the added value of HFR in the pollution and floatable tracking and five concrete examples of applications.**

This paragraph detailing the lagrangian approach has been removed following the recommendation from both anonymous reviewers. Section 2.3.1 has been shortened and the added value of HFR for pollution and floatables tracking has been highlighted in two well-described case studies. Additional examples of HFR applicability in this field have been referenced.

**Section 3**

- **The writing of this chapter needs some revision.**

  The entire section 3 has been revised, as suggested by both anonymous reviewers.

- **Some important points in figure 17 showing the very detailed SWOT analysis are not reflected in the text in the Section, for example: the important Weakness/Threats detected on 'Difficulties for addressing user engagement challenges' or 'Disconnection between academia with the private sector and policy-makers' or 'limited user uptake' or 'Fail in addressing the user needs' deserve more attention in the discussion.**

  The bullet points mentioned by the reviewer have been included in section 3:

  "Additional efforts for unlocking interoperable HFR (basic or added-value) data access will greatly contribute to deliver greater uptake, use and value from the HFR data. Once we are able to turn HFR data into customized information, the further challenge is to extend the science-based added-value products and applications into societal relevant downstream services (Tintoré et al., 2019). To face this challenge, a clear stakeholder engagement strategy is needed to identify, categorize and analyze the user needs, thus reinforcing the links and the loyalty with current users and enabling new communities and sectors to discover and use the HFR data."

  In the Fig. 17, the following items have been rewritten:

  Replace "Difficulties for addressing user engagement challenges" with "Lack of consolidated user engagement strategy"

  Replace "'Disconnection between academia with the private sector and policy-makers' with "Weak ties between the academia with the private sector and the policy-makers"

- **Fig17: some bullet points require better redaction, e.g.:**

  ○ **'Risk that the effort will diminish or disappear', which effort?**
  We are referring here to the coordinated efforts that have significantly increased the prompt distribution, availability, easy access and accuracy of the HFR data and enhanced the creation of a community at the HFR operator level. In order to tackle the recommendations provided in the manuscript to continue boosting and enhancing the HFR data use and user uptake, these joint efforts are needed more than ever.

  We have replaced the bullet point 'Risk that the effort will diminish or disappear' with "Risk that the coordinated efforts needed will decrease."

  ○ **'Lower HFR operational as usual', what 'as usual' means**

It means that the HFR systems of the Mediterranean Sea are commonly medium/short range systems, while lower frequency bands (i.e. long-range) are required to extend the range of the HFRs for implementing HFR-based tsunami early warning systems.

We have replaced the bullet point 'Lower HFR operational as usual' with "Long-range HFR are required for HFR-based tsunami early warning systems in the Mediterranean."

**Section 4**

**A Set of 13 prospects/recommendations are given by the authors, but it's not clear what are the recommendations and what are the prospects: as it reads now the recommendations are spread all along the 13 points (e.g.' collaboration with international initiatives' in c) and in m) or repeated in 2 or more points (e.g. 'creation of partnerships in points b) and k)). Could the authors perhaps group them by categories and specify what are the concrete recommendations to the HFR Mediterranean community?**

Section 4 has been modified to better distinguish the future prospects from the recommendations. The prospects of the future set the basis of the recommendations and will benefit from their progressive implementation.

12 recommendations are provided and are listed in the same alphabetical order as the weaknesses and threats, which are intended to be covered, and the opportunities to be seized by each one of them (as shown in the revised Fig. 17). When the recommendation is specific for the Mediterranean community, it has been adequately highlighted (i.e. Mediterranean, regional, etc.).

**The recommendations should also be better linked with the SWOT analysis, especially with the identified weakness and threats.**

As mentioned above, each recommendation has the same letter as the weakness or the threat that intends to cover or the opportunity to be seized (in the SWOT schema in Fig.17).

**For instance, recommendations on how to improve the long-term sustainability of the HFR network could be better stated (not only the short-term funding in grant calls mentioned in point l).**

We agreed with both anonymous reviewers, since relying on grant funding, particularly where this is focussed on new science, will not be sufficient to improve the long-term sustainability. An additional sentence has been added in Section 4, item l):

"In addition to the research/grant funding, long-term infrastructure funding at national, regional and European government level with financial input from other operational users through regional consortia will be needed towards a truly sustainable infrastructure. For further details about the diverse socioeconomic and technical challenges to be tackled during the implementation of a sustained and integrated HFR regional network, the reader is referred to the Sect. 5.1 of the companion publication from Lorente et al., (2021b)."

**Section 5**

**The concept of 'regionalization' mentioned in L1098 is not very clear. What the authors mean here, the same as 'regionalization of the recommendations' (L929) or 'collaboration at regional level'?**

We have replaced "regionalization" with "regional approach and focus to strengthen collaborations".

The revised paragraph in section 5 will read as follows:

"Certainly, this regional approach and focus to strengthen collaborations should always be aligned with the global and European strategies to ensure the integration, benefiting from the European HFR roadmap and the availability of near real-time and long-term HFR interoperable data that will boost the research and to underpin the further development of the HFR scientific and societal applications in the Mediterranean coastal areas."

**Technical comments**

**Some parts of the manuscript need minor some revision. See below some examples.**

- **L115: remove 'as described by several authors'**

  Removed

- **L117: should read 'installation instead of implementation'**

  Replaced

- **L122: what 'advanced' means here?**

  Advanced means mature applications. 'Mature' is used instead.

- **L135-137: needs a better formulation of the overall description of Section 4**

The reformulated description of Section 4 will read as follows:

Sect. 4 outlines the future prospects for HFR applications along with a set of key recommendations aiming to leverage the HFR data to its fullest extent, thus helping to harness the HFRs potential in the further development of operational monitoring systems at the regional level.

- **L137: 'overcome' should read 'achieve'**

Replaced

- **L138: References to the UN Decade and the European Green deal may help readers not familiar with those initiatives.**

Thanks for the notice. References have been added.

Ref:

Ryabinin, V., Barbière, J., Haugan, P., Kullenberg, G., Smith, N., McLean, C., Troisi, A., Fischer, A., Aricò, S., Aarup, T., Pissierssens, P., Visbeck, M., Enevoldsen, H. O. and Rigaud, J.: The UN Decade of Ocean Science for Sustainable Development , Front. Mar. Sci. , 6 [online] Available from: https://www.frontiersin.org/article/10.3389/fmars.2019.00470, 2019.

Sikora, A.: European Green Deal – legal and financial challenges of the climate change, ERA Forum, 21(4), 681–697, doi:10.1007/s12027-020-00637-3, 2021.

- **L149: is it possible to add a reference to the oil spill incident in 2021 mentioned by the authors?**

A very recent publication (from the 7th of March, 2022) has been included.

Ref:

García-Sánchez, G., Mancho, A. M., Ramos, A. G., Coca, J. and Wiggins, S.: Structured pathways in the turbulence organizing recent oil spill events in the Eastern Mediterranean, Sci. Rep., 12(1), 3662, doi:10.1038/s41598-022-07350-w, 2022.

- **L150: 'it has once again been demonstrated': explain why once again', the reader not necessarily know all the background behind this sentence.**
Paragraph has been reworded as follows:

"In this context, accurate forecasting of oil spill modeling (for this particular event, the model MEDSLIK-II was used, as described in De Dominicis et al., 2013a, 2013b) and Lagrangian trajectory analysis of floating objects (Sayol et al., 2014; Ličer et al., 2020) have demonstrated to successfully help marine SAR operations and oil spill containment. "

- **L164: typo in 'fate of the trajectory'**

  The sentence has been reworded to addressed a previous recommendation of the reviewer:

  "Winds, waves and surface currents observations and forecasts are needed to be seamlessly integrated into their SAR emergency response tools in order to predict the trajectory of a drifting target"

- **L211: Fig.1: better explain what 'HFR bounding box' is. There is a red square in the eastern Med which doesn't correspond with any country mentioned in the text.**

  Figure 1 has been improved, with the red boxes showing the geospatial coverage of each HFR system in the Mediterranean Sea.

  We have replaced "HFR bounding box" with "HFR mean spatial coverage" in the figure caption.

  A reference to the companion manuscript, which describes the Mediterranean HFR networks, has also been included.

- **L249: typo in 'Sirocco'**

  Typo corrected. Thanks a lot!

- **L280: not clear from the text what are the 'challenges associated with resolution'**

  Challenges associated with resolution in Ocean Models include e.g. the lack of well-tested scale-aware and flow-aware parameterizations capable of transitioning between coarse-resolution and fine- resolution to adequately resolve (sub)mesoscale coastal ocean processes (Haidvogel et al., 2017), the increase in strength of the nonhydrostatic effects (as horizontal resolutions increase) that would require a nonhydrostatic capability and the impacts of convergence of the numerical scheme as resolution increases.

  A specific reference from Fox-Kemper et al., 2019 is available in the manuscript for further information about the challenges and prospects in Ocean Circulation Models.

  The first paragraph could be added in the text, although the challenges of ocean models are out of the scope of this manuscript and have been addressed in many other publications already included in the manuscript. On the other hand, the reference from Fox-Kemper et al., 2019 shall be considered adequate for providing additional details about the challenges. We leave it to the discretion of the anonymous reviewer.

- **L337: 'for a wide range of coastal stakeholder including Ports and environmental agencies'**

Replaced. Thanks

- **L363: 'assimilated radial velocity observations'**

Thanks! 'velocity' has been included.

- **L384/385: what could be the reason for been the only operational example using HFR DA, is this discussed later in the text?**

This is a very relevant question that has been addressed in several meetings between in-situ data providers and modelers, since it seems to affect not only HFR data but also in-situ and satellite data in general.

As reported by Capet et al., 2020, only 23% of the 104 European operational model systems analyzed use some form of data assimilation, 8% of which only use it offline, for operational production of reanalysis. The lack of an adequate delivery of observations has been identified as a bottleneck that locks the systematic exploitation of data assimilation.

Specifically, regarding the HFR data assimilation and, despite valuable initiatives carried out in the Mediterranean (Marmain et al., 2014; Iermano et al., 2016; Vandenbulcke et al., 2017; Janeković et al., 2020), we should further strive to develop robust, fully operational assimilation schemes for HFR data, encompassing both radial and total current vectors. Most recently, Hernández-Lasheras et al., 2021 have reported that the HFR high temporal resolution potential has not yet been fully exploited due to new challenges associated with the use of HFR hourly data in the sequential data assimilation scheme. In addition, they concluded that a better knowledge of the observation error covariances, which are unfortunately often unavailable, would also be beneficial to improve the assimilation of HFR surface currents.

In our manuscript, these reasons have been included in section 4, issue f) "boosting of HFR data assimilation", including also the above mentioned references from Capet et al., 2020 and Hernández-Lasheras et al., 2021.

To the best of authors' knowledge, SOCIB WMOP (Juza et al., 2016, Mourre et al., 2018) is presently the only system in the Mediterranean Sea including an assimilation scheme of HFR data in its operational chain (https://socib.es/?seccion=modelling&facility=forecast). This is also mentioned in L465 and in section 4, issue h)

- **L440-443: the retrieval of wave and wind map, of high interest for sectors like offshore energy it is not under 'extreme hazard monitoring', please clarify why this sentence here.**

In the context of the Blue Economy (defined as all economic sectors that have a direct or indirect link to the oceans, such as marine energy, coastal tourism and marine biotechnology), we would like to underline both the importance of winds and waves for characterizing coastal extreme events, and the value of utilizing wind and wave energy from our oceans for sustainable blue growth.

The sentence has been modified as follows:

"In response to the increasingly frequent extreme events associated with climate change, their detailed characterization by means of surface current, waves and wind maps derived from HFRs may aid the Blue Economy development in coastal vulnerable areas of the Mediterranean region. Regardless of this, the increasing retrieval of waves and wind maps derived from HFRs (Lorente et al., 2021b) is very relevant for the development of renewable ocean energy, an emerging and innovative Blue Economy sector."

- **L448: 'This work' refers to Berta et al. 2020? Not clear in the text**

  Yes, it does. Paragraph might be reformulated like this:

  "HFRs have been used to investigate the upper ocean response to an extreme wind event in the Ligurian Sea (NW Mediterranean) during October-November 2018, as described in Berta et al. (2020). This work focused on the analysis of coastal submesoscale structures, shaping surface currents and passive transport. Authors estimate the pattern and magnitude of kinematic properties (e.g. divergence/convergence and vorticity patterns), derived from surface currents measured by the HFR-TirLig network (Fig. 1) to characterize the evolution (before and after the event) of ocean scales at a few kilometers."

- **L468-469: 'adding to the analyzing' needs better redaction**

  Improved '...evidenced Gloria's remote-effect in the Ibiza Channel and the Strait of Gibraltar, altering the usual water exchanges between adjacent sub-basins...'

- **L527: Typo in Sicily'**

  Corrected

- **L561-562: it is not clear if the radar in the Gulf of Lyon is operating at 4.5MHz in the Mediterranean, while the example gives before (Sagres, Portugal) operates at 13Mhz. Not clear what is 'Stradivarious radar'? could you please clarify.**

There are no long-range HFRs in the Mediterranean. The Stradivarius radar was just a prototype. Since this manuscript strictly focused on the Mediterranean HFR systems, we have removed any mention of non-Mediterranean HFR systems (e.g. Sagres) in the paragraph to avoid the confusion:

"Another related issue is extending the range of these HFR, which would imply operating at lower frequency bands (4-5 MHz or 9 MHz) than those usually employed in the Mediterranean region (13, 16 or 25 MHz). A radar prototype operating at 4.5 MHz with 200-300 km range was developed by the Diginext Ltd. a few years ago (i.e. Stradivarius radar) for the Gulf of Lion as a proof of concept, (Grilli et al., 2015). "

- **L565: 'As recently suggested by Domes et al., 2020...'**

Replaced

- **L583: can the authors provide some example or reference of 'sensitive ecosystems in the shallow water areas'**

Yes, of course. Some examples are provided in the text together with the relevant reference as follows:

"...hosting sensitive ecosystems in the shallow coastal waters (e.g. seagrass meadows of the endemic *Posidonia oceanica*, the key intertidal habitat of vermetid reefs built by endemic gastropod *Dendropoma petraeum*, the coralligenous assemblages, etc., as detailed by Coll et al., 2010). "

Reference:

Coll, M., Piroddi, C., Steenbeek, J., Kaschner, K., Ben Rais Lasram, F., Aguzzi, J., Ballesteros, E., Bianchi, C. N., Corbera, J., Dailianis, T., Danovaro, R., Estrada, M., Froglia, C., Galil, B. S., Gasol, J. M., Gertwagen, R., Gil, J., Guilhaumon, F., Kesner-Reyes, K., Kitsos, M.-S., Koukouras, A., Lampadariou, N., Laxamana, E., López-Fé de la Cuadra, C. M., Lotze, H. K., Martin, D., Mouillot, D., Oro, D., Raicevich, S., Rius-Barile, J., Saiz-Salinas, J. I., San Vicente, C., Somot, S., Templado, J., Turon, X., Vafidis, D., Villanueva, R. and Voultsiadou, E.: The Biodiversity of the Mediterranean Sea: Estimates, Patterns, and Threats, PLoS One, 5(8), e11842 [online] Available from: https://doi.org/10.1371/journal.pone.0011842, 2010.

- **L587: 'strong tourist' must read '...strong tourism pressure'**

Replaced

- **L599: 'fast growth...' must read 'growing importance of HFR as a key...'**

Replaced

- **L759: 'reliability with respect to the actual sea dynamics', please further explain.**

  The revised paragraph will read as follows:

  "Surface currents from HFR can be combined with the numerical model outputs both to bridge the gap relative to the spatial and temporal coverage of data, and to improve the representation and forecast of the real sea state: HFR observations enhance numerical simulations by resolving fine-scale processes in intricate regions with complex-geometry configurations. In turn, hydrodynamic models can reciprocally serve as integrative connectors of sparse in situ observations and gappy HFR surface current maps by offering a seamless predictive picture of the three-dimensional ocean state"

- **L776: should read 'is monitored weekly since 1984'**

  OK, thanks

- **L785: must read 'Chlorophyll-a'**

  Replaced

- **L851: should read 'decision-making in fisheries management'?**

  Replace with "Fisheries resource management"

- **L859: 'inventory' to be replaced by 'compilation' or 'review'**

  'Review' has been used instead.

- **L859-864: paragraph needs review**

  The revised paragraph will read as follows:

  "This initial review of scientific and societal applications using HFR data, implemented in the Mediterranean coastal areas, have allowed us to know their potential and limitations to continue contributing to the regional observing system in addressing the regional existing environmental threats, the scientific key priorities and the societal needs. Considering also the current threats and the opportunities ahead along the UN Ocean Decade (2021-2030), we have conducted a SWOT (i.e. strengths, weaknesses, opportunities, and threats) analysis, as schematized in Fig. 17, which includes the key points from this section."

- **L879-891: paragraph needs review**

  The entire section 3 has been reviewed and, in particular, the paragraph mentioned by the reviewer.

- **L922-923: sentence is not very clear. What is mean here by 'European HFR network'? is this a stablished Network?In other part of the text 'HFR network' was referred to particular HFR installations in different regions.**

The European HFR network is an established network that refers to the different HF radar systems operating in Europe.

We briefly provide the context, as it has been included in the companion paper from Lorente et al., (2021). MONGOOS (Mediterranean Operational Network for the Global Ocean Observing System) and EuroGOOS (European Global Ocean Observing System) are the Mediterranean and the European component, respectively, of the Global Ocean Observing System (GOOS).

In 2015, EuroGOOS launched the HFR Task Team with the aim of promoting the implementation of an operational HFR network in Europe based on coordinated data management and integration of basic products into the major platforms for marine data distribution technology (Corgnati et al., 2021, Rubio et al., 2021).

In the context of the EuroGOOS HFR Task Team a survey was conducted to build the inventory of the HF radar systems in Europe (Rubio et al., 2017), which is kept updated in this website: https://eurogoos.eu/high-frequency-radar-task-team/

In addition, each particular HFR system (e.g. HFR-TirLig) can also be referred to as a HFR network, since it is composed of more than one radial site.

The above-mentioned references have been included to provide this context, as follows:

"...to derive the added-value achieved by the European HFR network (Rubio et al., 2017; Rubio et al., 2021; Corgnati et al. 2021; Lorente et al, 2021b), such as the data management centralization and the standard data distribution, the development of new products and cross-disciplinary emerging applications, the provision of training workshops, etc."

References:

Corgnati, L., Mantovani, C., Rubio, A., Reyes, E., Rotllan, P., Novellino, A., Gorringe, P., Solabarrieta, L., Griffa, A. and Mader, J.: The Eurogoos High Frequency radar task team: a success story of collaboration to be kept alive and made growing, in 9th EuroGOOS International conference, pp. 467–474, Brest, France. [online] Available from: https://hal.archives-ouvertes.fr/hal-03328829, 2021.

Lorente, P., Aguiar, E., Bendoni, M., Berta, M., Brandini, C., Cáceres-Euse, A., Capodici, F., Cianelli, D., Ciraolo, G., Corgnati, L., Dadic, V., Doronzo, B., Drago, A., Dumas, D., Falco, P., Fattorini, M., Gauci, A., Gómez, R., Griffa, A., Guérin, C.-A., Hernández-Carrasco, I., Hernández-Lasheras, J., Licer, M., Magaldi, M., Mantovani, C., Mihanovic, H., Molcard, A., Mourre, B., Orfila, A., Révelard, A., Reyes, E., Sanchez, J., Saviano, S., Sciascia, R., Taddei, S., Tintoré, J., Toledo, Y., Ursella, L., Uttieri, M., Vilibic, I., Zambianchi, E. and Card\'\in, V.: Coastal HF radars in the Mediterranean: status of operations and a framework for future development, Ocean Sci. Discuss., 2021, 1–58, doi:10.5194/os-2021-119, 2021.

Rubio, A., Mader, J., Corgnati, L., Mantovani, C., Griffa, A., Novellino, A., Quentin, C., Wyatt, L., Schulz-Stellenfleth, J., Horstmann, J., Lorente, P., Zambianchi, E., Hartnett, M., Fernandes, C., Zervakis, V., Gorringe, P., Melet, A. and Puillat, I.: HF Radar activity in European coastal seas: Next steps toward a Pan-European HF Radar network, Front. Mar. Sci., 4(JAN), 1–17, doi:10.3389/fmars.2017.00008, 2017.

Rubio, A., Reyes, E., Mantovani, C., Corgnati, L., Lorente, P., Solabarrieta, L., Mader, J., Fernandez, V., Pouliquen, S., Novellino, A., Karstensen, J. and Petihakis, G.: European High Frequency Radar network governance, ARRAY(0xd4d7a3c)., 2021.

- **L927: the end of the sentence is not clear: 'to ensure the integration.', 'integration' of what?**

We are referring here to the coordination or blending of the different Mediterranean Coastal Ocean Observatories -COOSs- into a functioning and unified regional entity, which is fully harmonized and aligned with the European roadmap.

The revised paragraph will read as follows:

"These recommendations should be part of the long-term monitoring strategy for: i) the development and the integration of the different COOSs into a robust regional ocean observatory; ii) ensuring that such integration is fully aligned with the European and global roadmaps; and iii) addressing key science priorities and societal challenges of the Mediterranean coastal regions. "

- **L928: should read 'HFR data and applications use'**

Replaced with "HFR data and its applications "

- **L933: what the authors refer here by 'MONGOOS-HFR' Network? Is the same as the Mediterranean HFR Network?, or the Mediterranean HFR community is already organized around MONGOOS?**

The reviewer is right. Although both terms ("Mediterranean HFR Network" and "MONGOOS HFR Network") tend to be used indistinctly, they are not equivalent. Furthermore, we must not refer to the MONGOOS HFR Network but to the MONGOOS HFR Task Team. In short:

The Mediterranean HFR network comprises all the HFR systems deployed in this regional sea and the respective operators.

The MONGOOS HFR Task Team is a specific component of the Observing Working Group from MONGOOS, that coordinates joint actions and research contributions from the entire network. So thus, it constitutes a sub-entity where almost all members of the Mediterranean HFR network are involved, but not all of them.

In this case (section 4, item i) we are referring to the Mediterranean HFR network:

"Expansion of the Mediterranean HFR network: although this network is already representing the 55% of the HFR sites existing in the European inventory (Lorente et al., 2021b) "

- **L941: what the authors mean here by 'regionalization of the observatories'?**

We are referring here to the strategy of the ocean observatories to respond to regional needs. This has been clarified in the text:

"Therefore, aiming to improve the strategy of the ocean observatories to respond to regional needs, aiming to a better understanding of region-specific processes towards a fit-for-purpose design, an increased monitoring effort by expanding and improving the HFR network is required, allowing for a covering of a large geographical area on a routine basis. "

- **L945: 'Blue sectors' should read 'Blue economy sectors'?**

Replaced.

- **L985: 'essential observing variables' meaning 'essential ocean variables'?**

Replaced.

- **L997:'etc' should be removed or better placed.**

Removed.

- **L999: What does 'temporal analysis' mean here?**

We refer to trend analysis. The paragraph has been reviewed:

"Additionally, since the intensity of the multiple stressors (e.g. climate change effects, habitat loss and degradation, eutrophication, introduction of alien species, fishing practices) is increasing throughout most of the Mediterranean basin, trend analysis is an essential process in assessing the state of the ocean of a region. This will contribute to effectively informing current and future marine policies and management actions as well as to underpin longer-term scientific objectives. "

- **L1025: should read 'is the only regional model'**

Replaced.

- **L1050: should read 'efforts'**

Replaced.

- **L1068: Do the authors refer here to 'European 'or 'EuroGOOS' HFR Task Team?**

We are referring here to the EuroGOOS HFR Task Team (Corgnati et al., 2021, Rubio et al., 2021), which promotes the implementation of an operational HFR network in Europe based on coordinated data management and integration of basic products into the major platforms for marine data distribution technology.

References have been included.

- **L1077: should read'...ocean observing systems...'**

'Observing platforms' has been replaced with 'Ocean observing platforms'

- **L1079: what does 'advanced' means here? Mature applications?**

"Advanced" has been replaced with "Mature"

- **L1087-1090: This important sentence could be carefully reviewed: what are the 'services' referred here? what 'technological development' means here?**

By "services" we mean "fit-for-purpose science-based data downstream services", to access the data and metadata (i.e. data services) or to access near real-time information derived from the data (i.e. operational services) for marine policy-makers and society.

By "technological development" we mean the active and expanding field of investigation of the HF radar technology, which is being demonstrated by the broad coverage of the research topics and applications addressed.

However, aiming to clarify and improve the reading, this sentence has been reformulated as follows:

"Considering the steadily growing integration of HFRs in the COOSs of the Mediterranean Sea and once their capabilities as an active and expanding field of investigation has been demonstrated by a wide range of practical applications, we can conclude that this consolidated land-based remote sensing technology plays a key role in the development of fit-for-purpose services for marine and maritime end-users."

---

## Author Comment (AC2)

- Title: Coastal HF radars in the Mediterranean: Applications in support of science priorities and  societal needs
- Author(s): Emma Reyes et al.
- MS No.: os-2021-115
- MS type: Review article
- Special Issue: Advances in interdisciplinary studies at multiple scales in the Mediterranean Sea

Please, note that all the points raised by the reviewer are in **bold font** below, while the responses to the review are in normal font and the changes made to the revised manuscript are "quoted".

**General comments**

**This paper presents an extensive review of many different and important applications of HF radar in the Mediterranean. Other reviews of this type have been published, and are referenced in this paper, but I think this is the first one that focusses on this particular region. As a review it is not intending to present new science but it brings together in an interesting way many different scientific contributions and developments, as well as some operational applications, all well-referenced and acknowledged. I did occasionally get lost in some of the detail and felt that there was some unnecessary repetition of key HFR advantages (high temporal and spatial resolution for example) when addressing each of the different applications.**

We sincerely appreciate the reviewer's comments and the time invested on that. Below are all of her/his thoughtful suggestions and concerns followed by our responses.

In short, repetitions have been avoided, unclear concepts and/or paragraphs have been modified, the quality of the figures have been improved and the figure captions further detailed, thus contributing to ensure our joint work is communicated clearly and accurately.

**Although I am providing a long list of suggested modifications to the English, on the whole the paper is well-written and will be of interest to many, and not just to those working in the Mediterranean region.**

Authors are very grateful to the reviewer for her/his careful, detailed and extensive revision for language that will help to improve the reading and comprehension of the manuscript. Following her/his recommendations, we have polished this manuscript after carefully correcting any errors in spelling, grammar and word choice to guarantee the publication quality.

**Specific comments**

1. **To make it easier to navigate through the paper it might be helpful in section 2 to list the subheadings of the three main topics in an introduction to the section. The key features of HFR that support these applications could perhaps be included in this introduction so they do not need to be repeated elsewhere.**

   Agreed. Thanks for your comments.

   We have listed the subheadings of the three main topics in the introduction in section 2.

   "This section presents the existing advanced and emerging scientific and societal applications using HFR data, aiming to address science priorities and societal needs identified in the Mediterranean coastal waters, organized around three main topics: (i) maritime safety; (ii) extreme hazards and (iii) environmental transport processes."

   We have included the key feature of the HFR technology in the Introduction (section 1) to avoid needless repetition along the manuscript.

   "The increased capability to address the above-mentioned regional challenges at the required spatio-temporal scales has directly benefited, *inter alia*, from the key features of the High Frequency Radar (HFR hereinafter) technology, i.e. unprecedented high spatio-temporal resolution (i.e. 0.2-6 km and 15-60 min) over wide coastal areas (up to 200 km offshore, depending on the operational frequency)."

2. **Fig 14 and associated discussion. I couldn't relate what I can see in (a) and (b) to the discussion on page 31. Can the link between them be made more clearly?**

   The reviewer is completely right, since an additional step was missing in the discussion on page 31. Thanks for the notice. The paragraph has been revised:

   "(i) Reconstruction of the annual and seasonal regimes of HFR currents detected at the LTER-MC site; (ii) Running Lagrangian backtracking simulations advecting virtual phytoplankton patches (VPPs) in the HFR field (Fig. 14, a). VPPs were released at LTER-MC site on the dates of the weekly oceanographic campaigns and tracked backward, allowing thus the estimation of the positions of the VPPs up to 4 days (i.e. 96 h) prior to its arrival at LTER-MC; (iii) Identifying the prevailing directions from which the VPPs arrive at LTER-MC site, as resulting from backtracking simulations (Fig.14, b), also allowing the definition of the spatial distribution of the VPPs origin zones (not shown) in the GoN; "

3. **I am not sure why the mathematical underpinning of lagrangian transport is included here (equations 1 and 2) when such details for other methods reviewed in the paper are not – and do not need to be.**

Agreed. Thanks for your suggestion. Equations have been removed and the paragraph has been reworded following your recommendation and aligned with the anonymous reviewer #1 comment.

4. **There is no specific discussion in here about the different types of HFR and their relative merits. Perhaps that is in the companion paper which I haven't seen yet. For most of the applications which involve currents this may not be necessary but when it comes to mapping waves and winds the type of radar is more important so a few words where these measurements are referred to would be useful.**

The reviewer makes an excellent point here about the importance of discussing the type of HFR, particularly in the case of wave and wind monitoring. Authors therefore have included a detailed discussion in this regard in the companion paper, in section 2.1 (entitled "Fundamentals of the HFR technology") and in the SWOT analysis, highlighting the "Non-mature operational stage for waves and winds" as well as the "lack of HFR-derived waves and winds data standards".

The companion paper (Lorente et al., 2021), which has already been accepted, has been cited in L. 566 and in section 4 item b, as follows:

"In addition to that, HFR experts from the Mediterranean institutions are currently and actively contributing in the definition of the European HFR network roadmap detailed in Rubio et al., (2021), leading crucial tasks. One of these tasks aims to define the standard model and to increase the availability and accuracy of the HFR wave parameters, which are weaknesses identified in Lorente et al., (2021). Another important task focuses on reaching a consensus on the methodology for the provision of the HFR data gap-filling products."

Ref:

Lorente, P., Aguiar, E., Bendoni, M., Berta, M., Brandini, C. Cáceres-Euse, A., Capodici, F., Cianelli, D., Ciraolo, G. Corgnati, L., Dadić, V., Doronzo, B., Drago, A., Dumas, D., Falco, P., Fattorini, M., Gauci, A., Gómez, R., Griffa, A., Guérin, C-A., Hernández-Carrasco, I. Hernández-Lasheras, J., Ličer, M., Magaldi, M., Mantovani, C., Mihanović, H., Molcard, A., Mourre, B., Orfila, A., Révelard, A., Reyes, E., Sánchez, J., Saviano, S., Sciascia, R., Taddei, S., Tintoré, J., Toledo, Y., Ursella, L., Uttieri, M., Vilibić, I., Zambianchi, E., Cardin, V. (accepted). Coastal HF radars in the Mediterranean: Status of operations and a framework for future development. Ocean Sci. Discuss., 2021, 1–58, doi:10.5194/os-2021-119, 2021.

5. **In connection with section 4 item I on funding. If European HFR networks are to be truly sustainable there needs to be long term infrastructure funding at national, regional and European government level with financial input from other operational users through regional consortia. Relying on grant funding, particularly where this is focussed on new science, will not be sufficient.**

We do agree with the reviewer. The following sentence has been included in section 4) :

"In addition to the research/grant funding, long-term infrastructure funding at national, regional and European government level with financial input from other operational users through regional consortia will be needed towards a truly sustainable infrastructure. "

**Minor corrections/suggestions**

- **Abstract li35-36. I would suggest a slight rewording for clarity 'The Mediterranean Sea is a prominent climate change hot spot, with many socio-economically vital coastal areas being the most vulnerable targets for maritime safety, diverse met-ocean hazards and marine pollution.' Or something like that.**

  Thank you! Reworded!

- **P2 li 40. Remove 'the' in 'in the Coastal Ocean'.**

  Done!

- **P2 li 46. Remove 'finally'.**

  Done!

- **P2 li 47. Replace 'societal' with 'societally'.**

  Done!

- **P2 li 56. Remove 'the' after 'covering'.**

  Done!

- **P2 li 58. 'Both not really needed.**

  Removed!

- **P4 li 116. Remove 'et al' after Wyatt. Reference lists shows just one author.**

  Removed!

- **P4 li 124. Replace 'It is worth to highlight' by 'It is worth highlighting'.**

  Replaced

- **P4 li 130. Rearrange 'providing the first one a' to 'the first one providing a'.**

  Rearranged

- **P4 li 135. Remove 'the' before 'Sect.4'.**

Removed!

- **P5 li 155. Replace 'highly' with 'greatly'. Replace 'its high' by 'their high' and 'its near' by 'near'.**

  All replaced

- **P5 li 160. Replace 'control' by 'controlled'.**

  Replaced

- **P6 li 170. Replace 'from which' with 'of which'.**

  Replaced

- **P6 li 176. 'accounts with'?? do you mean 'has'**

  Yes, I do. Replaced

- **P6 li 178. Replace 'being the 51% from' with 'of which 51% were from'.**

  Replaced

- **P6 li 179-180. Rewrite as 'In particular, the number of SAR incidents in the French Mediterranean responsibility area accounts for 23% (3110) of the total number of cases and 32% (7293) ….'.**

  Rewritten

- **P6 li 181-182. Remove 'The' before '94%' and replace 'being more of' with 'with more of'.**

  Removed and replaced

- **P6 li 186. Replace 'for 1875' with 'to 1875'.**

  Replaced

- **P6 li 192. Replace 'being also' with 'with'.**

  Replaced

- **P6 li 194. Remove gap in '267 874' for consistency with large number notation elsewhere (or perhaps include a hyphen if that was missing?).**

  Gap removed

- **P6 li 195-196. Rewrite as '…reported as SAR cases occurring within Maltese Territorial Seas'.**

  Rewritten

- **P6 li 198. Rewrite end of sentence from ' are served by..' I am not sure what is meant here.**

Agreed. We have modified the sentence:

"HFR data are combined with forecast model outputs to get the best representation of the sea state during SAR operations"

- **P7 li 204. Replace 'from' with 'of'.**

  Replaced

- **P7 li 205. Rewrite as '(the remaining 12% being in response to …'.**

  Rewritten

- **P7 li 206. Replace 'From 7 of the HFR' by 'Of the 7 HFR '**

  Replaced

- **P7 li 222. Replace 'obtaining' with 'for'**

  Replaced

- **P11 li 282. Replace 'issues as' by 'issues such as'/**

  Replaced

- **P11 li 283. 'for instance' not really needed.**

  Removed

- **P12 Fig 4 caption. Is CC index the same as complex correlation coefficient?**

  Yes, it is. A better description has been included in the caption of Figure 4: "Magnitudes of the complex correlation (i.e. CC index) and phase between HFR and model-predicted currents are provided in red font color."

- **P13 li 362. 'only a limited number of studies have been'.**

  Replaced

- **P14 Fig 5. I can't see a definition of 'CR'?.**

  You are completely right. Thanks for the notice. The proper definition of CR (i.e. control run) has been included in the Figure caption.

- **P15 li 408. Insert 'an' between 'using' and unsupervised'.**

  Inserted

- **P15 li 411. I don't think prerequisite is a verb. Suggest 'requires' or 'assumes' instead.**

  'Requires' is used instead.

- **P16 li 420. Replace with 'the HFR systems has had substantial problems since**

**2010 and the antennas were eventually removed.'**

Replaced

- **P17 li 446-447. Perhaps rewrite as 'The analysis was based on pattern and magnitude estimates of kinematic properties from surface currents'**

Rewritten

- **P17 Fig7. The whole figure should be on one page rather than spilt up. The font size on the maps is too small to read clearly.**

Agreed. We have modified Figure 7 to make the font sizes bigger and to display it on one page.

- **P18 li 469. Replace ' analyzing' with 'analysis of'.**

Replaced

- **P19 ;l 474. Replace 'the flash pressure drops' by the sudden pressure drop'.**

Replaced

- **P19 li 485-486. Unit split across lines.**

Solved

- **P21 li 542. Replace 'damages' with damage'.**

Replaced

- **P22 li 559. Remove 'a' before 'software'.**

Removed

- **P22 li 570. I suggest 'into the ocean, multi-scale coastal ocean dynamics being the key drivers…'.**

Suggestion accepted, thanks!

- **P22 li 571. 'HFRs have demonstrated a capacity to provide very ….'.**

Rewritten

- **P23 li 580. Replace 'threat' with 'threats'.**

Replaced

- **P24 li 617. Replace 'state for' with 'are' and define the x' variable at the end of the sentence. Although see also**

This paragraph has been removed following the recommendation from the anonymous Reviewer 1 aiming to avoid the details on how Lagrangian trajectories are computed since it is being considered as part of the methodology.

- **P24 li 630. 'Moreover' not needed.**

  Removed

- **P25 li 648. 'Besides' not needed.**

  Removed

- **P29 ;l 743. 'Furthermore' not needed.**

  Removed

- **P29 li 751. I suggest rewrite as 'for all seasons except spring, although some were able to ….'.**

  Rewritten

- **P31 li 776. Remove 'at the'..**

  Removed

- **P32 Fig 14. Map font sizes are too small.**

  Agreed. We have modified Figure 14 to make the font sizes bigger

- **P32 Fig 14 caption li 797. Delete 'as'**

  Removed. In addition, the date has been modified according to the period of analysis as mentioned in the text.

- **P32 li 808. Replace 'determinant' with perhaps 'important' or 'most important' or …**

  'Important' is used instead.

- **P33 li 830-831. Either 'physically driven' or 'physical driving'.**

   'physical driving' used instead.

- **P34 li 842. Replace 'along' with 'in'.**

  Replaced

- **P34 Fig 16. I can't find a link to Fig 16 in the text although presumably it should be on this page.**

  Thanks for the notice. Cross-reference to Fig. 16 has been included three times in this page.

- **P35 li 863. Remove 'the' before 'Fig'.**

  Removed

- **P36 li 880-881. Suggest rewriting as '….'and making routine maintenance tasks easier under these severe…'**

  Suggestion accepted. Thanks a lot!

- **P36 li 881. Rewrite as 'it is worth highlighting that under…'**

  Rewritten

- **P36 li 887. Replace 'still' by 'yet'.**

  Replaced

- **P36 li 890. Remove 'as usual'.**

  Removed

- **P36 li 902-903. I wasn't sure what the phrase beginning ' models, being these..' was getting at. Perhaps replace with 'models, which in turn will benefit from the future expansion of the HFR network'.**

  Replaced

- **P37 Fig 17. The fact that HFR is a surface measurement and many applications require currents at depth (as mentioned in the text) could perhaps be added as a weakness.**

  Yes. The fact that the HFR observations are limited to the very near surface has been considered as a weakness in the SWOT analysis of the companion paper (Lorente et al, 2021, accepted). The reference has been added in this manuscript.

  **And I was not sure what 'Lock (should this be lack?) of the HFR data potential' means?**

  The 'lock of the HFR data potential' refers to the limitations in the HFR interoperable data access that do not allow the leverage of the data in a systematic and dynamic way. This restricts the possibilities for delivering greater uptake, use and value from the collected data to its fullest potential. The unlocking data access and potential is aligned with so-called "democratization of data" (Buck et al., 2019), aiming to turn data into information, facing the further challenge of extending the science-based added-value products into societal relevant downstream services (Tintoré et al., 2019), impacting in how we use, manage and sustain our coastal oceans.

  We have modified the sentence in the Figure 17:

  "Lock of the interoperable data access", followed by:

"Limited data uptake, data use and value"

**I'm not sure why 'lower HFR frequencies than is usual in the Med' is a threat. HFRs operating at lower frequencies are available. Perhaps a weakness of the current network?**

In order to provide early alerts of plausible tsunamigenic sources in the Mediterranean Sea, the extension of the HFR range is required, which imply operating at lower frequency bands than those usually employed in the Mediterranean. As detailed in the companion paper, the Mediterranean HFR network includes 15 different systems, which cover a small portion of the entire coastal domain (Fig. 1). The limited spatial coverage is not only due to the reduced number of HFR deployed but also to the predominant use of medium (13.5 MHz) and short (above 20 MHz) range systems. While these HFRs present a maximum range of 80 km, long range systems (which operate below 5 MHz and are typically deployed in the Atlantic European waters) can map the surface circulation over broader areas for distances up to 200 km offshore.

Long-range HFR systems are not deployed in the Mediterranean since they present some technical limitations in this semi-enclosed sea that seriously handicap the full coverage of coastal waters. On one hand, they provide surface circulation maps with coarser horizontal grid resolution (above 5 km), which are not convenient to adequately resolve some submesoscale ocean processes (i.e., eddies, instabilities, etc.) that commonly characterize the Mediterranean sea state. On the other hand, they cannot accurately monitor the wave field under low sea states as the second-order spectrum is closer to the noise floor (and more likely to be contaminated with spurious contributions) than in the case of short and medium range HFR systems. As the Mediterranean wave climate is not as intense as the Atlantic one, the use of long-range systems would result in limited precision and reduced temporal continuity in wave measurements (Lipa and Nyden, 2005).

- **P38 li 919. 'Finally' not needed'**

  Removed

- **P38 li 935. Remove 'the' at the end of the line.**

  Removed

- **P 38 li 943-944. Suggest rewriting as ' Accordingly a review of major scientific and social questions is needed including the environmental….'**

  Rewritten

- **P39 li 969. I suggest something like ' held by companies or controlled by arrangements with the private sector'. I'm not sure what 'compromised to' means.**

Suggestion accepted. Many thanks

- **P40 li 986. I am stumped. What does 'different compartment' mean?**

  Agreed. We have modified the text to make this clearer:

  "In this context, it is worth mentioning that the HFRs multi-parameter monitoring of the sea state allows the development of diverse applications to tackle a wide range of coastal threats"

- **P40 li 996. A recent paper on applications to offshore wind power could perhaps be included in this list [http://dx.doi.org/10.21926/jept.2101005](http://dx.doi.org/10.21926/jept.2101005).**

  Reference added. Thanks!

- **P40 li 1003. Suggest a full stop after 'sustainability' followed by 'It'.**

  Suggestion accepted. Thank you very much

- **P40 li 1011. Suggest moving 'will offer' to after 2022.**

  Suggestion accepted.

- **P42 li 1071. Replace 'ones' by 'some'.**

  Replaced

- **P43 li 1087. Replace 'increasingly' with 'increasing'.**

  Replaced

---

## Author Comment (AC3)

[revised manuscript text omitted]


occurring within Maltese Territorial Seas. The HFR-CALYPSO monitors the Malta-Sicily channel, accounting for 7 HFR sites and the HFR-CALYPSO-SOUTH, is composed by two HFR sites located in the South of Malta. HFR data are combined with forecast model outputs to get the best representation of the sea state during SAR operations.

(v) Slovenia: has 42 km of coastline and a semi-enclosed coastal area. During 2019, the SAR agency has responded to 9 SAR missions (7 times the rescue boat went out to sea while 2 rescues were of injured people on a moored boat in port). All cases occurred within the 3 nm from the coast (i.e. 3 within 200 meters, 3 around 1 nm and 1 at 3 nm from the coast). The HFR-NAdr, in the Northern Adriatic Sea and the Gulf of Trieste, is jointly operated through trans-national collaboration with Italy.

(vi) Spain: the 4 SAR responsibility areas cover 1 500 000 km2 of marine surface (3 times the size of the Spanish national territory) and 8 000 km of coastline. The Spanish Maritime Safety and Rescue Agency (SASEMAR hereinafter) is divided in 19 MRCCs plus 1 National Centre, with more than 370 SAR operators. SASEMAR responded to 5 891 missions in 2019, of which almost 88% were SAR operations. Fifty percent of the total SAR incidents occurred within 3 km off the Spanish coastlines. Of the 7 HFR networks operating inside their 4 responsibility areas, 3 of them are located in the Western Mediterranean, monitoring the Strait of Gibraltar (HFR-Gibraltar), the Ebro Delta (HFR-Ebro) and the Ibiza Channel (HFR-Ibiza) and all of them are integrated in the SASEMAR Environmental Data Server.

[Figure]

**Figure 1. Map of the Mediterranean showing the HFR mean spatial coverage (pink contours) and the location of SAR incidents of France, Italy, Slovenia and Spain from 2019 colored based their distance to the closest coastal point. For further details about the operational status and the names of the HFR systems the reader is referred to Lorente et al, (2021b).**

As aforementioned, maritime SAR operations most often depend on leveraging Lagrangian tracking tools using timely and reliable knowledge of surface circulation, near surface winds and, if applicable, surface gravity waves. Surface circulation is generally provided by numerical circulation models but HFR observations can offer valuable insight into marine conditions


[Figure]


[revised manuscript text omitted]

[Figure]

[Figure]

[revised manuscript text omitted]

**STRENGTHS**
- HFRs has boosted the research at the coastal areas.
- HFRs help to overcome the scarceness of observations and the ocean model limitations in coastal areas.
- HFRs are a great asset for NRT model operational assessment and model improvement.
- HFRs are very resilient platforms, even under extreme events.
- HFRs can detect tsunami-induced currents.
- HFRs can track oil spill and marine litter.
- HFRs can detect and track small-eddies.
- Increasingly capability to develop HFR Short Term Predictions.

**OPPORTUNITIES**
- i) Steadily growing European HFR network.
- j) Growing interest of HFR new capabilities.
- k) Extending the coverage and the timeseries.
- l) UN's Decade raises ocean's awareness.
- Fostering of the HFR data in supporting the CZM.
- HFRs can greatly contribute in the achievement of the GES of the Mediterranean water.
- Existence of an European HFR stakeholder engagement strategy to facilitate co-production.
- Integration of HFRs in tsunami warning systems.
- HFR combination with other observing platforms.
- Key role of NGOs and citizen science in monitoring.

**WEAKNESSES**
- a) Limited adoption of common data and metadata models for HFR surface currents.
- b) Lack of HFR-derived waves and winds & added-value operational distribution in standardized format.
- Lack of consensus on the methodology to generate some basic and added value HFR data products.
- c) Lack of consolidated user engagement strategy.
  - Need to reinforce the user's loyalty.
- d) Weak ties between the academia with the private sector and the policy-makers.
- e) Limited development of methodologies for data combination and integration.

**THREATS**
- f) Limited HFR data assimilation in models.
  - Lock of the interoperable data access.
  - Limited user uptake, data use and value.
  - Fail in addressing the user needs.
  - Lack of HFR tsunami alert system available in the Mediterranean Sea, despite the existing risk.
- Long-range HFR are required for HFR-based tsunami early warning systems in the Mediterranean.
- g) Different capabilities to resolve and characterize the processes for diverse observing platforms and models.
- Risk that the coordinated efforts needed will decrease.
- h) Limited training to HFR technicians and scientists.

445 **Figure 17. SWOT analysis of the HFR capabilities and applications. Dashed-line boxes around the text highlight those weaknesses, threats and opportunities that have been addressed in the recommendations.**

**4. Future prospects for HFR applications and recommendations**

After the description of the current implementation status of the HFR applications in the Mediterranean coastal areas, and based on the results obtained from the previous SWOT analysis, we present here the prospects for the future and a set of key

450 recommendations. Future prospects for HFR applications will benefit from the progressive implementation of the defined recommendations: (i) to ensure that the potential of the HFR data is fully exploited in the development of operational monitoring systems at the regional level; (ii) to help to derive the added-value achieved by the European HFR network (Rubio et al., 2017; Rubio et al., 2021; Corgnati et al. 2021; Lorente et al 2021b), such as the data management centralization and the standard data distribution, the development of new products and cross-disciplinary emerging applications, the provision of

455 training workshops, etc.; (iii) to include the recommendations in the long-term monitoring strategy. This last point is crucial for: (i) the development and the integration of the different COOSs into a robust regional ocean observatory; (ii) ensuring that



[revised manuscript text omitted]

**Reinforcing the Mediterranean's leadership** for continuing to be a major European focus of HFR activity. The Mediterranean institutions are key players in the European and international HFR research effort in multidisciplinary fields and in the development of applications, as mostly covered in the Sect. 2 of this work. In addition to that, HFR experts from the Mediterranean institutions are currently and actively contributing in the definition of the European HFR network roadmap, leading crucial tasks aiming to define the standard model, increase the availability and accuracy of the HFR wave parameters and to reach a consensus on the methodology for the provision of the HFR data gap-filling products (Rubio et al., 2021). The creation of partnerships between different research groups and institutions in the context of European, regional and national projects (Lorente et al., submitted to this Special Issue) is contributing to move forward both the expansion of the HFR network and the main research areas, also fostering the HFR data interoperability [6]

Movido hacia arriba[2]: Enhancing data discoverability,

[revised manuscript text omitted]

| Página 19: [1] Eliminado | Emma Reyes | 25/3/22 7:10:00 |
| Página 20: [2] Eliminado | Emma Reyes | 3/4/22 18:49:00 |
| Página 24: [3] Eliminado | Emma Reyes | 3/4/22 19:50:00 |
| Página 24: [4] Eliminado | Emma Reyes | 8/3/22 13:05:00 |
| Página 38: [5] Eliminado | Emma Reyes | 4/4/22 12:51:00 |
| Página 42: [6] Eliminado | Emma Reyes | 4/4/22 3:40:00 |
| Página 42: [7] Eliminado | Emma Reyes | 25/3/22 5:43:00 |

a)

| Página 42: [8] Eliminado | Emma Reyes | 25/3/22 5:47:00 |

b)

| Página 43: [9] Eliminado | Emma Reyes | 4/4/22 3:23:00 |
| Página 43: [10] Eliminado | Emma Reyes | 4/4/22 3:16:00 |